# Unraveling the role of vaporization momentum in self-jumping dynamics of freezing supercooled droplets at reduced pressures

Xiao Yan [1,2,3,6] ✉, Samuel C. Y. Au [1,6], Sui Cheong Chan[1], Ying Lung Chan[1], Ngai Chun Leung[1], Wa Yat Wu[1], Dixon T. Sin [1], Guanlei Zhao[4], Casper H. Y. Chung [1], Mei Mei[1], Yinchuang Yang[1], Huihe Qiu [1] & Shuhuai Yao [1,5] ✉

Supercooling of water complicates phase change dynamics, the understanding of which remains limited yet vital to energy-related and aerospace processes. Here, we investigate the freezing and jumping dynamics of supercooled water droplets on superhydrophobic surfaces, induced by a remarkable vaporization momentum, in a low-pressure environment. The vaporization momentum arises from the vaporization at droplet's free surface, progressed and intensified by recalescence, subsequently inducing droplet compression and finally self-jumping. By incorporating liquid-gas-solid phase changes involving vaporization, freezing recalescence, and liquid-solid interactions, we resolve the vaporization momentum and droplet dynamics, revealing a size-scaled jumping velocity and a nucleation-governed jumping direction. A droplet-size-defined regime map is established, distinguishing the vaporization-momentum-dominated self-jumping from evaporative drying and overpressure-initiated levitation, all induced by depressurization and vaporization. Our findings illuminate the role of supercooling and low-pressure mediated phase change in shaping fluid transport dynamics, with implications for passive anti-icing, advanced cooling, and climate physics.

Water droplet freezing on solid surfaces poses safety and economic threats to transportation infrastructure, power generation/transmission systems, and telecommunication facilities[1–4]. A classic example in the aerospace industry is the in-flight icing of supercooled droplets that has been recognized as a significant aviation hazard. More recently, the global transition from fossil energy to renewable energy has exacerbated the issue of ice accretion on wind turbine blades and photovoltaic panels[2], which causes significant power loss and serious safety concerns. Freezing of a water droplet on solid surfaces starts from ice nucleation[5] followed by crystallization propagation accompanied by the release of latent heat (recalescence)[6]. Extensive studies during the past decades have shed

[1]Department of Mechanical and Aerospace Engineering, Hong Kong University of Science and Technology, Hong Kong, China. [2]Key Laboratory of Low-grade Energy Utilization Technologies and Systems, Chongqing University, Ministry of Education, Chongqing 400030, China. [3]Institute of Engineering Thermophysics, Chongqing University, Chongqing 400030, China. [4]State Key Laboratory of Automotive Safety and Energy, School of Vehicle and Mobility, Tsinghua University, Beijing 100084, China. [5]HKUST Shenzhen-Hong Kong Collaborative Innovation Research Institute, Futian, Shenzhen, China. [6]These authors contributed equally: Xiao Yan, Samuel C. Y. Au. ✉e-mail: yanx23@cqu.edu.cn; meshyao@ust.hk

light on the solidification physics governing sessile droplet freezing[4,7–10], condensation frosting[11–13], and impact droplet freezing[14–16]. The established understanding contributes to the development of various icephobic surfaces such as super-hydrophobic surfaces[17–23], slippery surfaces[24,25], and gel-type or polymer-based low-toughness surfaces[26–29] that can regulate ice accretion by removing supercooled droplets before freezing, delaying ice nucleation, or reducing ice adhesion.

Despite the growing knowledge in freezing physics and icephobic surfaces, most of the studies confined themselves to atmospheric conditions with little attention to the environmental effects. Different from atmospheric freezing, droplets in a low-pressure environment have enhanced vaporization flux over the droplet interface, leading to substantial supercooling that enables fast freezing from the free surface of the droplet[6,30,31]. The instant freezing subsequently leads to droplet self-launching from superhydrophobic surfaces[6,32]. To account for the remarkable self-launching of freezing droplets under low-pressure, an overpressure theory[6] has been proposed. The overpressure theory postulates that the Cassie-Baxter droplet experiences an overpressure underneath it as a result of liquid vaporization and confinement of the released vapor within the surface structures. This overpressure generates a force that propels the droplet to lift off the superhydrophobic surface and jump. However, the overpressure is predicted to decay sharply with increasing surface structure scales, and it becomes insignificant as the length scale of surface structures reaches ~100 μm[6]. In fact, freezing droplets have been observed to jump off superhydrophobic surfaces that possess large-scale structures within this range[32].

The unexplored driving force other than overpressure for droplet self-launching has prompted researchers to examine the role of vaporization during progressive recalescence. Recalescence has been demonstrated to elevate the surface temperature, introduce intensive vaporization, and exert forces on the droplet[33]. Recent endeavor has identified the recalescence-induced reaction force, referred to as the propulsion force[34] or recalescence force[32], which acts on the freezing droplets and plays a crucial role in shaping their behaviors. This is exemplified by the remarkable Cassie-Baxter-to-Wenzel wetting transition of freezing droplets on textured superhydrophobic surfaces[32,35], as well as the intriguing acceleration exhibited by airborne droplets[34,36]. For a sessile droplet, the recalescence force typically develops in a progressive top-down manner, from the droplet's free surface to the substrate, and tends to induce droplet impalement into sparsely spaced substrate structures[32,35]. This opposes the localized overpressure that typically separates the droplet from the substrate, leading to self-trampolining of vaporizing droplets[6]. Rational substrate

structuring promotes self-expulsion of freezing supercooled droplets[32], potentially due to rebound behavior[35]. However, a more comprehensive understanding of the top-down recalescence force's role in rebound and self-expulsion dynamics is needed. The diverse outcomes (impalement, jumping, and trampolining) emphasize the necessity for a thorough comprehension of the intricate dynamics in multiple phase change processes, as well as the challenges in predicting the droplet freezing and jumping phenomena.

In this study, we investigate the self-jumping dynamics of supercooled, freezing droplets on superhydrophobic surfaces at low pressures (~100 Pa). We reveal a significant counteractive momentum, termed vaporization momentum, resulting from the progressive recalescence of the droplets. We demonstrate that this vaporization momentum surpasses the overpressure generated underneath the droplet and leads to compressive deformation followed by jumping. In contrast to the previous work on self-launching[6,32], wetting transition[32,35], and self-propulsion of freezing droplets[34,36], the focus of this study is to unveil the fundamental role played by vaporization momentum in shaping the self-jumping dynamics, considering their intricate interplay with droplet size, freezing dynamics, droplet deformation, and substrate-droplet interactions. Particularly, a regime map is developed to demonstrate the findings of droplet size-defined regimes of depressurization-induced transport phenomena observed in supercooled droplets.

## Results

Figure 1a shows the experimental setup used to investigate the freezing and departure dynamics of supercooled droplets at a low-pressure environment (see also Supplementary Fig. 1 and Fig. 2, Section S1 of Supplementary Information for more details on experiments). A deionized water droplet was initially deposited on a horizontal superhydrophobic substrate. The superhydrophobic substrate has an apparent advancing contact angle $\theta_a^{app} \approx 160°$ and an apparent receding contact angle $\theta_r^{app} \approx 156°$. The vacuum chamber was connected to a mechanical pump that rapidly depressurized the chamber from the atmospheric pressure (101 kPa) to a relatively stable and low-pressure (~100 Pa) within 15–20 s before the droplet froze (Fig. 1b). High-speed optical and thermal imaging were performed to capture the droplet freezing and jumping dynamics. See Methods for more details on experimental setup and procedures.

Due to the intensive evaporative cooling at the liquid-gas interface (with a heat flux of up to ≈100 kW/m², see Supplementary Fig. 1d, Section S1 of Supplementary Information), the droplet surface temperature rapidly decreased to a supercooled temperature (Fig. 1b) until ice nucleation initiated from the free surface of the droplet (Fig. 2a, see

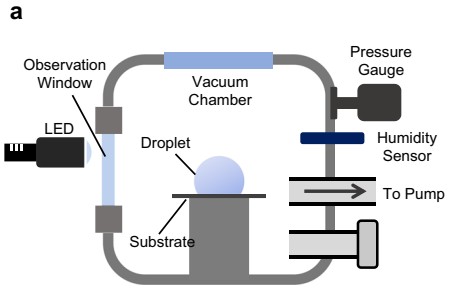
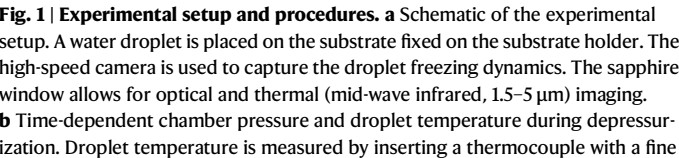
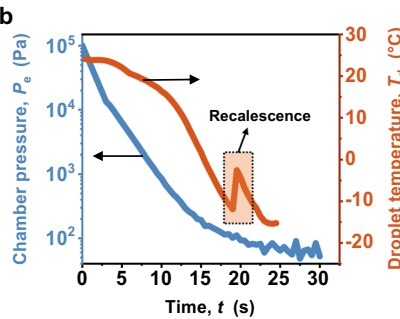

**Fig. 1 | Experimental setup and procedures. a** Schematic of the experimental setup. A water droplet is placed on the substrate fixed on the substrate holder. The high-speed camera is used to capture the droplet freezing dynamics. The sapphire window allows for optical and thermal (mid-wave infrared, 1.5–5 μm) imaging. **b** Time-dependent chamber pressure and droplet temperature during depressurization. Droplet temperature is measured by inserting a thermocouple with a fine

tip into the droplet residing on the substrate. The environmental pressure is measured by the pressure gauge. As the chamber pressure decreases from 101 kPa to ≈100 Pa, the droplet is supercooled to reach an internal temperature below −12 °C before experiencing sudden heating to ≈0 °C due to recalescence (marked by the dotted frame). See Methods and Section S1, Supplementary Information for details on experimental setup and procedures.

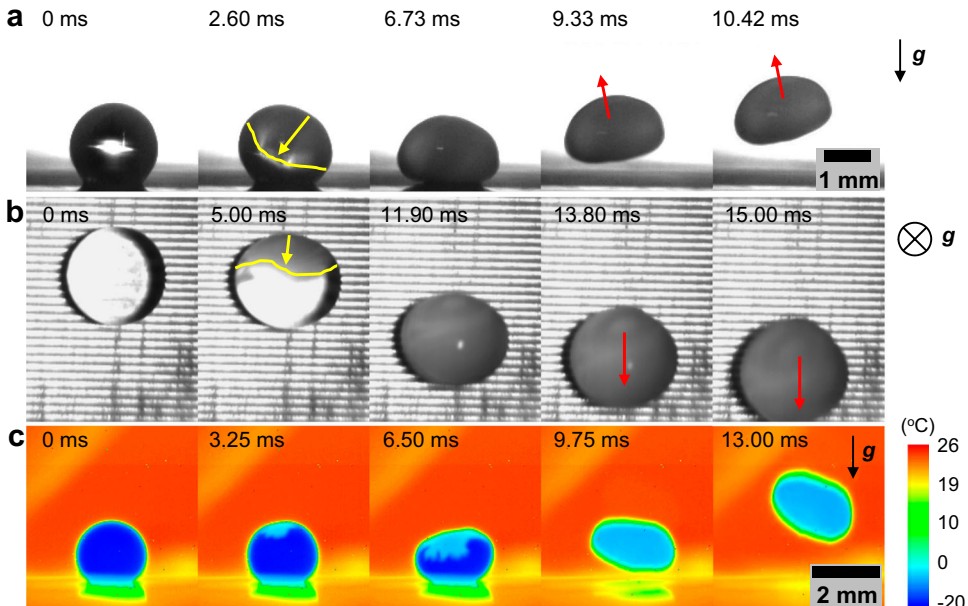

**Fig. 2 | Visualization of freezing droplets. a** Side-view and (**b**) top-view high-speed time-lapse images of a water droplet (with a volume of ≈10 μL) freezing and jumping on micro/nanostructured superhydrophobic surfaces at reduced pressure. The freezing front and the freezing propagation direction are indicated by yellow lines and arrows, respectively, while the jumping direction of the freezing droplet is indicated by red arrows. **c** Side-view thermography of a freezing and jumping droplet, showing the ice (in light blue) nucleation and propagation over the free surface of the droplet (in dark blue). The direction of gravity is indicated by the arrows in (**a**) and (**c**) or the circled cross in (**b**) (with gravity pointing to the paper plane). See Supplementary Fig. 2, Section S1, Supplementary Information for a detailed characterization of the droplet temperature. See Supplementary Fig. 3, Section S2 of Supplementary Information for detailed characterization of the superhydrophobic surfaces. See Supplementary Movies 1, 2.

Supplementary Movie 1). Upon ice nucleation, the freezing front (marked by yellow lines, $t = 2.6$ ms in Fig. 2a and $t = 5$ ms in Fig. 2b) propagated rapidly along the droplet surface. The freezing process was accompanied by a sudden release of latent heat of solidification, which led to recalescence and heating of the ice-water slurry from the supercooled state to the equilibrium freezing temperature (≈0 °C), as indicated by the infrared (IR) imaging (Fig. 2c, see Supplementary Movie 2). As the ice shell formed to enclose the droplet, the droplet underwent compressive deformation from a sphere to an ellipsoid ($t = 6.73$ ms, Fig. 2a), followed by the self-detachment from the substrate ($t = 9.33$ ms, Fig. 2a). In addition to out-of-plane jumping, the freezing droplet may experience in-plane movement depending on the progressive freezing, as confirmed by the top-view visualization (Fig. 2b, see Supplementary Movie 1).

In contrast to previously reported droplet levitation induced by local overpressure[32], the observed droplet self-detaching in this study is characterized by the droplet's compressive deformation and freezing propagation-dependent jumping. The unique droplet freezing and departure dynamics indicate the presence of a counteractive momentum, which we term vaporization momentum, that acts on the freezing surface. We hypothesize that the vaporization momentum results from the counteraction of the outward vaporization flux that leaves the droplet surface (Fig. 3a, b). To confirm the vaporization momentum, we performed droplet freezing on a smooth, slippery surface with a low contact angle hysteresis ($\theta_a^{app}$-$\theta_r^{app} < 8.5°$)[37]. The use of slippery surfaces allowed us to eliminate the effects of overpressure and surface structures[6]. Upon depressurization-induced freezing, the droplet was pushed to deform and slide on the surface in the direction of freezing propagation (Supplementary Fig. 12, Section S6, Supplementary Information), suggesting the universality of the vaporization momentum of freezing droplets at low pressures. Similar to the vaporization momentum mechanism proposed here, evaporation momentum has also been identified in boiling, where fast evaporation of heated liquid leads to the deformation of growing bubbles[38].

Furthermore, the vaporization momentum is consistent with the recalescence force identified in a recent study on freezing droplets[32]. These observations support our hypothesis of the vaporization momentum resulting from intensive interfacial phase change.

To further understand how vaporization momentum influences freezing droplet dynamics, we quantified the timescale of droplet detachment $\triangle t_d$, defined as the time duration starting from ice nucleation to droplet-substrate separation. Visualization of droplet freezing suggests that the ice shell almost encloses the droplet at the detachment moment, with a small portion at the droplet bottom remaining unfrozen upon detachment (Fig. 2a, c, see also Supplementary Fig. 20 and Supplementary Movie 3). This indicates that droplet detachment occurs within the recalescence stage (-10 ms, Fig. 2c). Regardless of surface microstructures (Fig. 3c), $\triangle t_d$ is shown to scale with $R_d^{1.5}$ ($\triangle t_d \approx 260R_d^{1.5}$, Fig. 3d), where $R_d$ is droplet's spherical radius immediately before freezing. In addition to the ice shell formation governed by freezing kinetics, the fluid dynamics of the unfrozen portion of the droplet likely play a significant role in determining the timescale of droplet deformation and detachment. High temporal resolution visualization revealed a liquid bulb at the bottom of the droplet, which underwent deformation and contraction when interacting with the substrate (see Supplementary Movie 3 and Supplementary Fig. 20, Supplementary Information). Furthermore, energy and heat transfer analysis shows that the average ice mass fraction within the ice-water slurry is adequately low (≈10%) and the liquid volume within the freezing droplet occupies a large portion at the droplet detaching moment ($t = \triangle t_d$) (see Supplementary Fig. 6, Fig. 10, Section S5, Supplementary Information for the transient heat transfer modeling and the calculation of the ice-water distribution[39,40] within the droplet upon the completion of recalescence). Even in the presence of an ice shell, $\triangle t_d$ is found to be comparable with the characteristic timescale of a freely oscillating droplet[41] and the contact time of a low-deformation impact droplet on superhydrophobic surfaces (see Section S11, Supplementary Information for a discussion of timescale)[42].

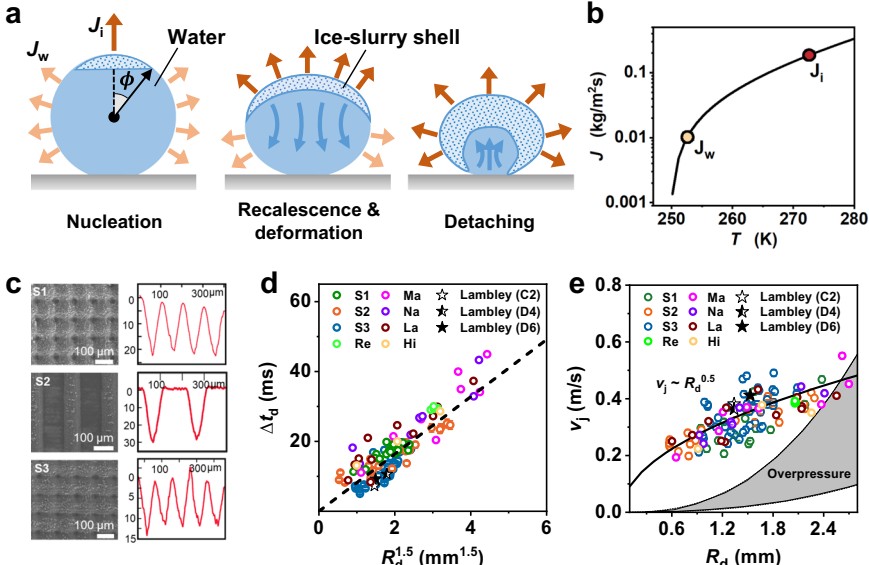

**Fig. 3 | Droplet freezing and self-jumping dynamics. a** Schematic showing the cross-section of a freezing droplet. As the ice shell (hatched) develops over the droplet's free surface, the droplet is deformed due to the momentum induced by asymmetric vaporization. The substrate counteraction towards the liquid core redirects the vaporization momentum to enable droplet detaching. $\phi$ indicates the angular position of the ice front. $J_i$ (orange arrows) and $J_w$ (light orange arrows) indicate the vaporization fluxes at the recalescing surface and supercooled water surface, respectively. The flow field within the enclosed liquid core (light blue) is indicated by dark blue arrows. **b** Calculated vaporization flux as a function of surface temperature. $J_i$ (at 273.15 K) and $J_w$ (at 253 K) are marked. See Section S3, Supplementary Information for more details on the calculations of vaporization flux. **c** Scanning electron microscopy (SEM) of micro/nanostructured substrates, labeled S1, S2, and S3, with their cross-section profiles on the right side. See Section S2, Supplementary Information for more details on the substrates. **d** Detachment time $\triangle t_d$ as a function of droplet radius $R_d^{1.5}$ ($R_d$ measured immediately before

freezing) on substrates with differing structures, including boehmite nanoblades (labeled Na), Al microlattice with nanostructures (labeled La), Al macrogrooves with nanostructures (labeled Ma), hierarchical CuO nanowires (labeled Hi), and re-entrant Si microgrids (labeled Re), in addition to samples S1–S3. See Supplementary Fig. 4 and Table 1 for detailed characterization of the surfaces. **e** Droplet jumping velocity $v_j$ as a function of droplet radius $R_d$ for different substrates. The solid line and gray band represent the prediction by the vaporization momentum ($v_j \sim R_d^{0.5}$) and overpressure theory ($v_j \sim R_d^{2.5}$), respectively. The predicted velocity by the overpressure theory reflects the variation of microstructures of samples S1-S3. See Section S10, Supplementary Information for detailed modeling of over-pressure. Independent experimental data from literature (Lambley, H. et al., ref. 32) which used different structured surfaces (samples C2, D4, D6, star symbols) also follow the same scaling law in (**d**) and (**e**). Source data are provided as a Source Data file. The errors of $\triangle t_d$, $R_d$, and $v_j$ associated with imaging processing and measurement are estimated to be <5%[49].

We then quantified the vaporization flux resulting from recalescence during droplet freezing. The interfacial vaporization flux ($J$) is driven by vapor pressure differences and can be described by the Schrage equation, $J \sim \left(P_d/\sqrt{T_d} - P_e/\sqrt{T_e}\right)$[43,44], where $P_d$ and $T_d$ are the local vapor pressure and temperature at the droplet surface (ice or water, see Eq. S7, Section S3, Supplementary Information), respectively; $P_e$ and $T_e$ are the environmental pressure and temperature, respectively, both of which were stabilized before droplet freezing (Supplementary Fig. 1). Here, the vapor pressure in the far field is assumed to be equal to $P_e$ due to the fast depressurization that minimizes the content of NCGs. During the short-time recalescence ($\triangle t_d$<20ms), the freezing surface remains at the equilibrium freezing temperature (≈0 °C), and the temperature of the supercooled water remains almost constant (Fig. 2c). Calculations show that the vaporization flux at the freezing surface ($J_i$) is up to 10X larger than that at the supercooled water surface ($J_w$), i.e., $J_i \sim 10 J_w$ (Fig. 3b, see also Supplementary Fig. 4, Section S3, Supplementary Information). When calculating $J_i$ for the ice-water mixture, we used the vapor pressure of ice at 0°C, which is slightly (≈6%) lower than that of water, and thus the calculated $J_i$ is a conservative estimation of the vaporization flux at the freezing surface (see Eqns. S5 and S6, Section S3, Supplementary Information). Corresponding to the vaporization flux, the ejecting velocity of vapor was calculated to be ≈144 m/s at the freezing surface and ≈65 m/s at the water surface, both of which are lower than the sonic velocity (≈430 m/s, see Section S8, Supplementary Information). Thus, the shock wave effects were not considered in our study. The non-uniform vaporization results from the temperature and vapor pressure difference at the freezing and supercooled liquid

water surfaces[33], and the vaporization flux contrast represents a pulse of vapor flow as the freezing front proceeds (Fig. 3a). Indeed, a strong vapor pulse during recalescence was detected by a flow indicator placed close to the freezing droplet (Supplementary Fig. 6, Section S4, Supplementary Information), consistent with previous observations[33].

Given that the central core inside the freezing droplet during recalescence remains as liquid (see Section S5, Supplementary Information), we assume the jumping of the freezing droplet to be a liquid-like bouncing process[42], where the majority of the vaporization momentum, which pushes the droplet to deform, is largely transferred to the droplet's bouncing momentum when the liquid core interacts with the superhydrophobic substrate. This process is different from the classic droplet impact and rebounding on a superhydrophobic surface[42], where a droplet experiences spreading and retraction before bouncing under the capillary and inertia forces. In the freezing droplet case, the droplet deforms as the ice shell builds up, and the retraction of the liquid core is partially restricted by the ice-slurry shell. Mediated by the liquid core-substrate interaction, the vaporization momentum is accumulated and eventually redirected to propel out-of-plane droplet jumping. Such a vaporization-bouncing assumption allows us to estimate the maximum possible contribution of vaporization momentum to droplet jumping through the momentum conservation during droplet freezing and jumping:

$$\int_0^{\triangle t_d} F_t dt \approx m_d v_j \tag{1}$$

The left term of Eq. (1) represents the vaporization momentum accumulated during progressive recalescence with intensified vaporization. The right term represents the momentum of the freezing droplet at the detaching moment. $F_t$ is the total net vapor propulsion force at time $t$, resulting from the intensive release of vapor from the droplet's free surface. $F_t$ is surface area-dependent and can be obtained by integrating the vapor propulsion force over the freezing surface, and $F_t = \pi R_d^2 J_i v_r \sin^2 \phi$ (see Eq. S17, Supplementary Information), where $J_i$ is the vaporization mass flux at the recalescing surface (see Eq. S7, Supplementary Information). $v_r$ is the vapor velocity determined by $v_r = J_i/\rho_v$, where $\rho_v$ is the vapor density. $\phi$ is the angular position of the freezing front with respect to the initial nucleation site (Fig. 3a). $m_d$ and $v_j$ are the mass and translational velocity of the jumping droplet, respectively. Given the limited mass loss due to vaporization during the short freezing time, $m_d \approx \rho_w V_d$, where $\rho_w$ and $V_d$ ($\sim R_d^3$) are the density and volume of the initial water droplet, respectively. For the droplet volume ($V_d < 70 \mu L$) used in our experiments, the gravitational force is neglected in Eq. 1 due to the small Bond numbers of droplets (Bo = 0.05–0.945 < 1), defined as the ratio of the gravitational force to capillary force of the droplet. The substrate-droplet adhesion was also negligible given the superhydrophobicity. Furthermore, the propulsion force contributed by the vaporization of supercooled liquid was not accounted for in $F_t$, since its vaporization mass flux is significantly lower than that at the freezing surface ($J_i/J_w$, Fig. 3b, see also Supplementary Fig. 4, Supplementary Information). The detachment time $\triangle t_d$ is used to characterize the timescale for momentum transport in Eq. (1), during which the substrate interacts with the droplet to re-direct the vaporization momentum for out-of-plane jumping. Recalescence is assumed to complete upon droplet detachment given the comparable time scales of freezing and detachment (see Section S11, Supplementary Information for a detailed discussion on the time scales of detachment and freezing). Noting that $\triangle t_d \sim R_d^{1.5}$, and $J_i$ is almost constant for a given equilibrium freezing temperature ($\approx 0\,°C$) and environmental pressure ($\approx 100$ Pa), Eq. (1) can be simplified as (see Section S8, Supplementary Information for detailed derivation):

$$v_j \sim R_d^{0.5} \tag{2}$$

We note that the scaling relationship represents the upper limit of the jumping velocity contributed by vaporization momentum, and the freezing is assumed not to reduce the bouncing momentum.

To quantify droplet jumping velocity as a function of droplet size, we performed experiments using droplets of 6–70 μL on superhydrophobic surfaces having varying microstructures (labeled as samples S1, S2, and S3 in Fig. 3c). The microstructures are characterized by a solid fraction ($\varphi_m$) ranging from 0.223 to 0.447 and a height ranging from 15 μm to 30 μm. All microstructures are conformally covered by nanostructures with a solid fraction ($\varphi_n$) estimated to be 0.108 and a characteristic height of 100 nm (see Section S2, Supplementary Information)[45]. Surfaces having differing microstructures were specially selected to vary the potential local overpressure underneath the freezing droplet, as shown by previous studies to govern freezing droplet self-jumping at a higher ambient pressure (~1000 Pa)[6]. More importantly, these surfaces represent the commonly used superhydrophobic surfaces with robust water repellency against liquid impalement, condensation, frosting, and ice accretion[45–48]. The robust superhydrophobicity of our surfaces eliminates the potential interference by surface adhesion and droplet impalement[32], thus allowing us to focus on the droplet jumping dynamics and to develop a general framework of vaporization-momentum-dominated droplet dynamics. The high-speed droplet jumping process was captured to extract the jumping velocity and direction at the detaching moment by tracing the droplet trajectory (see Supplementary Fig. 13, Section S7, Supplementary Information)[49,50]. Fig. 3e displays the measured

jumping velocity as a function of the square root of the droplet radius, demonstrating consistency with the scaling analysis ($v_j \sim R_d^{0.5}$). It was shown that a larger droplet size leads to a higher droplet jumping velocity (for Bo ≤ 0.945). To validate the scaling relationship, we conducted additional experiments using a greater variety of surfaces having differing surface structures and materials. The length scales of surface structures span a wider range from nanoscale (~100 nm) and microscale (~10 μm to ~100 μm) to macroscale (~mm). The morphology of the structures varies from randomly distributed boehmite nanoblades (labeled Na), Al microlattice with nanostructures (labeled La), Al macrogrooves with nanostructures (labeled Ma), hierarchical CuO nanowires (labeled Hi), to re-entrant Si microgrids (labeled Re) (see Supplementary Fig. 4 and Table 1, Supplementary Information for detailed characterization). In addition, we analyzed recently reported droplet jumping on polymer surfaces with well-defined microstructures (labeled C2, D4, and D6, respectively)[32]. All results were in good agreement with the scaling (Eq. (2)) regardless of the surface structures. The scatter of $v_j$ in experiments (Fig. 3e) may be attributed to variations in the extent of droplet deformation, whereas our calculations of the vaporization momentum (Eq. (1)) assume a perfect sphere. A deformed droplet has a larger surface area, resulting in a higher vaporization momentum. Indeed, we measured an increase in jumping velocity with increasing droplet deformation in terms of surface area (see Supplementary Fig. 15, Section S9, Supplementary Information).

The vaporization momentum analysis above, which accounts for the intensive vaporization at the freezing surface, can well explain our experimental results even without considering the role of substrate-mediated local overpressure in droplet jumping. To evaluate the role of local overpressure, we examined the overpressure underneath the freezing droplet by utilizing the overpressure theory[6]. The overpressure correlation reveals that for a given surface structure, the overpressure force $F_{op} \sim R_d^4$ (see Eq. S29, Section S10, Supplementary Information). Considering the timescale of the development of overpressure ($\triangle t_{op} \approx 0.3\tau_c$)[6] and using the momentum equation having a similar form to Eq. (1), the overpressure momentum yields a jumping velocity $v_j \sim R_d^{2.5}$ (see Section S10, Supplementary Information for derivation). We found that the best-fitted power index to our experimental data is 0.5 rather than 2.5 (Supplementary Fig. 16, see Section S10, Supplementary Information). Moreover, the overpressure theory significantly underestimates the droplet jumping velocity (Fig. 3e, see also Eq. S30, Supplementary Information). More importantly, the local overpressure is anticipated to be sensitive to substrate structure length scales[6], while we observed little dependence of jumping velocity on surface patterns (nanosheets, pillars, grooves, lattice, and re-entrance) and solid fractions ($\varphi_m$ ranging from 0.223 to 0.447, $\varphi_n$ ranging from 0.108 to 0.196 (Fig. 3e, see also Supplementary Fig. 17, Section 10, Supplementary Information). The discrepancy between the overpressure theory and our experiments suggests that droplet jumping is not dominated by the local overpressure in our experiments. However, at higher environmental pressure with a significant amount of NCGs, we anticipate the vaporization momentum to decrease due to diffusion-limited interfacial vaporization (see Supplementary Fig. 18, Section S10 in the Supplementary Information). In such conditions, the local overpressure mechanism mediated by small-scale surface structures may play a more important role in droplet behavior[6]. The overpressure effects can be also amplified by the droplet sizes since the overpressure force (as well as the resulting velocity, Fig. 3e) increases dramatically with droplet radius. As such, droplet levitation in the absence of freezing can be observed for extremely large droplets (~100 μL) as we will demonstrate later.

The vaporization momentum is further shown to govern the jumping direction. Since the initial nucleation site represents the origin of progressive recalescence, it determines the spatiotemporal evolution of vaporization fluxes over the droplet surface and thus the

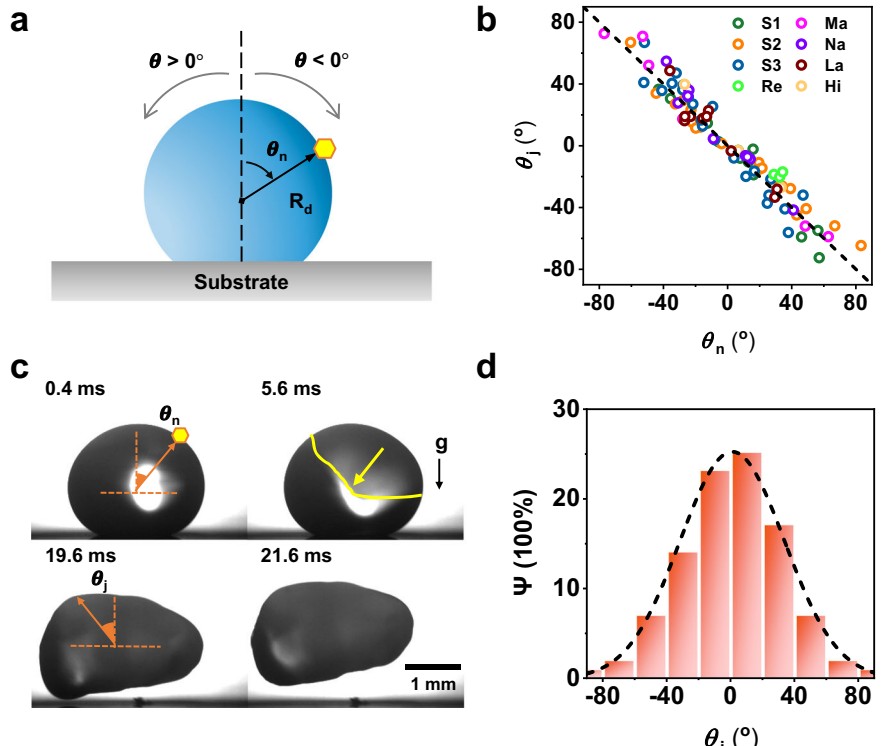

**Fig. 4 | Nucleation-mediated directional jumping. a** Schematic showing the angular position of the nucleation site ($\theta_n$) with respect to the droplet's sphere center and jumping angle ($\theta_j$) with respect to the substrate's normal line. The normal line with respect to the substrate represents $\theta_j = 0°$ or $\theta_n = 0°$. An angle towards to right is defined to be positive and vice versa. **b** Jumping angle ($\theta_j$) as a function of the angular position of the ice nucleation site ($\theta_n$) for differing surfaces (labeled S1, S2, S3, Ma, Na, La, Hi, and Re), showing a linear trend, i.e., $\theta_j = -\theta_n$. See Supplementary Fig. 4 and Table 1 for surface characterization. **c** An example

demonstrating the relationship of jumping angle ($\theta_j$) and angular position of the nucleation site ($\theta_n$). The nucleation site is marked by a hexagon. **d** The relative frequency ($\Psi$) of the jumping angle ($\theta_j$), fitted using a Gaussian function (dotted curve), $\Psi = \frac{1}{43.3\sqrt{2\pi}}\exp[-0.5\left(\theta_j/43\right)^2]$ with a mean of $\approx 0°$ and a standard deviation of $\approx 43°$. The errors of $\theta_j$ associated with imaging processing are estimated to be $<5°$[49], and the errors of $\theta_n$ associated with measurement are estimated to be $<2°$. The total number of samples for evaluating $\Psi$ is 101.

jumping direction. We define the jumping direction by the jumping angle ($\theta_j$) with respect to the substrate's normal line at the detaching moment, which is obtained by jumping trajectory analysis (see Section S7, Supplementary Information)[49]. The angular position ($\theta_n$) of the initial ice nucleation site with respect to the droplet's sphere center is used to characterize the spatial distribution of ice nucleation (Fig. 4a). It was shown that $\theta_j$ is linearly correlated with $\theta_n$, independent of droplet sizes and substrates, i.e., $\theta_j = -\theta_n$ (Fig. 4b). Figure 4c demonstrates a droplet with an ice nucleation position on the right-top interface of the droplet ($\theta_n = -34.2°$), while the freezing droplet jumps toward left-top (e.g., $\theta_j = 30.3°$). The vaporization momentum (Eq. (1)) explains the directional jumping in relation to the ice nucleation position. When nucleation occurs away from the central line of the droplet, the vaporization momentum gains an in-plane component during the progressive recalescence process, while simultaneously, the component facing the substrate is redirected to detach the droplet from the surface. The jumping directionality observed here, which cannot be explained by the overpressure mechanism[32], further highlights the role of vaporization momentum in self-jumping of freezing droplets.

Because the ice nucleation position and the jumping direction observed appear probabilistic, we then statistically sampled the data from repeated experiments, and found that the distribution of jumping direction (and ice nucleation position), quantified by the relative frequency $\Psi$, follows the Gaussian density function (Fig. 4d). The Gaussian distribution reveals that the droplet is more likely to jump perpendicularly to the substrate ($\theta_j \approx 0°$) and less probable to jump in plane ($\theta_j \approx \pm 90°$). This is because ice nucleation tends to initiate at the

top of the supercooled droplet, where the temperature is lower due to evaporative heat transfer in a sessile droplet (see Supplementary Fig. 24 for the simulated temperature profile). Furthermore, the symmetry of the jumping direction distribution suggests that the freezing droplet holds an equal probability of jumping leftwards and rightwards, indicative of the minimal effects of experimental conditions (e.g., directional airflow during vacuuming) on droplet freezing and jumping.

The resolved dynamics of jumping droplets highlight the significant role of vaporization momentum, which, however, may be diminished or inhibited for significantly small or large droplets. To determine the regime where vaporization momentum governs droplet dynamics, we extensively expanded the range of droplet sizes in our experiments. This led to the identification of distinct regimes, defined by droplet size, in the depressurization-induced phenomena of sessile water droplets. These regimes include evaporative drying for small droplets, vaporization-momentum-dominated droplet freezing and self-jumping for medium-sized droplets, and overpressure-initiated Leidenfrost effects for large droplets (Fig. 5a). Specifically, there exist a lower ($R_{d,l}$) and upper ($R_{d,u}$) boundaries in terms of equivalent droplet radius ($R_d$) that delineate the regime of droplet freezing and self-jumping. For $R_d < R_{d,l}$, the droplet dries up before freezing (Fig. 5b, Supplementary Movie 4). The value of $R_{d,l}$ is determined when the evaporation lifespan ($t_{eva}$) of a droplet with an initial $R_d$ is shorter than the ice incubation time ($t_{inc}$) (Fig. 5c). While for $R_d > R_{d,u}$, the droplet experiences a marginal and dynamic levitation rather than jumping, facilitated by a vapor layer underneath, as well as remarkable fluctuations at the interface (Fig. 5d, Supplementary Movie 5), both of which

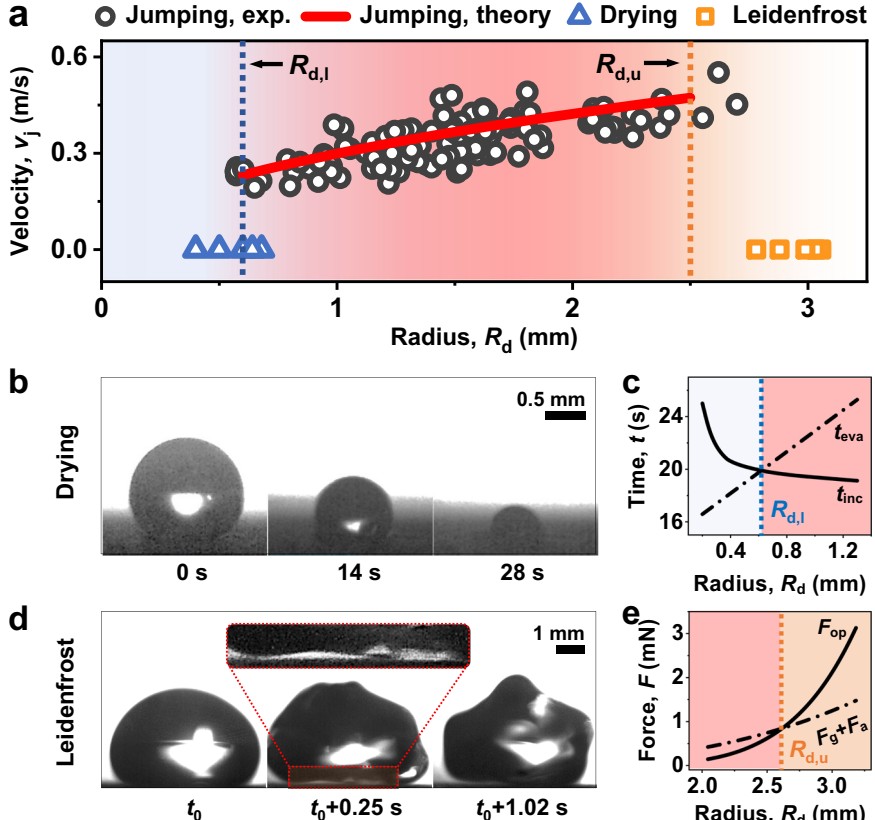

**Fig. 5 | Size-defined droplet dynamics upon depressurization. a** Regime map illustrating droplet transport phenomena on superhydrophobic substrates in a low-pressure (-100 Pa) environment, depicting the relationship between droplet jumping velocity ($v_j$) and droplet size ($R_d$). The vaporization-momentum-governed self-jumping regime (jumping) is delimited by the evaporative drying regime (drying) for small droplets and the overpressure-initiated Leidenfrost effect regime (Leidenfrost) for large droplets. Theoretical boundaries are represented by dashed lines. Note, for the evaporative drying regime, $R_d$ is defined as the initial droplet radius immediately before pumping down. For the self-jumping regime, $R_d$ is measured as the droplet radius immediately before freezing. For the Leidenfrost regime, $R_d$ is defined as the equivalent droplet radius of a spherical droplet having an identical volume immediately before droplet levitation. **b** Optical images showing the evaporative drying regime. $t = 0s$ corresponds to the start of pumping down. See Supplementary Movie 4. **c** Theoretical determination of the lower

boundary ($R_{d,l}$) by comparing the evaporation lifespan ($t_{eva}$) and ice incubation time ($t_{inc}$) as a function of the initial droplet radius ($R_d$). $t_{eva} < t_{inc}$ represents the droplet dries up before freezing. **d** Optical images showing the Leidenfrost effect regime. The overpressure-induced droplet-substrate separation is highlighted and zoomed in in the frame. $t_0 \approx 30s$ represents the pumping down time before the initiation of droplet levitation. See Supplementary Movie 5. **e** Theoretical determination of the upper boundary ($R_{d,u}$) by comparing the overpressure force ($F_{op}$) and the sum of gravitational force and adhesion force ($F_g + F_a$) as a function of equivalent droplet radius ($R_d$). $F_{op} > F_g + F_a$ represents that the overpressure overcomes the resistance forces and induces droplet-substrate separation. Here, gravitational and adhesion forces are incorporated since the droplet has a significantly large size (Bo>1) and a non-spherical shape with an increased droplet surface contact area. Sample S1 with a microstructure height of 15 μm was used in the experiments and modeling.

indice the characteristics of the Leidenfrost effect[51–53]. The Leidenfrost effect significantly delays freezing due to enhanced convection caused by the remarkable droplet fluctuations, counteracting the temperature decrease at the droplet interface. $R_{d,u}$ is reached when the overpressure force $F_{op}$ overcomes the resistance forces (gravitational force $F_g$ and adhesion force $F_a$), causing droplet-substrate separation (Fig. 5e). Here, gravitational and adhesion forces are incorporated as the droplet has a significantly large size (Bo > 1) and a non-spherical shape with increased droplet surface contact area. The theoretical boundaries are in good agreement with our experiments (see Section S13, Supplementary Information for detailed analyses of the regime boundaries).

## Discussion

In conclusion, we have demonstrated the underlying physics for the freezing and self-jumping of supercooled water droplets from superhydrophobic surfaces at reduced pressures. Although observed at low pressures (-100 Pa), the self-jumping phenomena indicate a universal mechanism with respect to droplet surface interaction coupled with vaporization momentum. Different from the previously reported local

overpressure underneath the Cassie-Baxter droplet that lifts the droplet, the vaporization momentum arises from the recalescence-intensified progressive vaporization and imparts a counteractive force to the freezing surface, leading to the compressive deformation of the droplet as freezing front propagates. We quantified the droplet jumping dynamics and revealed a scaling law of jumping velocity, which did not depend on surface structures. To identify the droplet size range where the vaporization momentum governs the droplet dynamics, we constructed a regime map delineating the regime of freezing and self-jumping for medium-sized droplets, in contrast to the regimes of evaporative drying for small droplets and overpressure-initiated Leidenfrost effects for significantly large droplets. While previous studies have delicately elucidated the effects of surface structures in droplet dynamics (self-trampolining, suffusion, wetting transition, and expulsion)[6,32,35], current work focuses on droplet self-jumping from impalement-resisting superhydrophobic surfaces, and highlights the crucial role of droplet size rather than surfaces structures in jumping velocity and the size-defined fate of droplets at reduced pressures (evaporative drying, self-jumping, and overpressure-initiated levitation).

In the future, the fluid dynamics within the liquid core confined by a growing ice shell, the effects of evaporative cooling rate on droplet freezing and detaching dynamics, and the coupling between the top-down vaporization momentum and bottom-up over-pressure, demands further investigation. Furthermore, while our current study demonstrates that the jumping dynamics are primarily governed by freezing dynamics and show minimal dependence on the tested surface structures, surfaces with heterogeneous adhesion and structures may interfere with droplet-substrate interaction and impact the jumping dynamics (see Section S12, Supplementary Information). Moreover, the absence of wetting transition[32], an opposite phenomenon induced by recalescence force, suggests the robustness of self-jumping of freezing droplets on a variety of superhydrophobic surfaces. However, the criteria of wetting transition[32,35] need to be incorporated into the rational design of surface structures, especially single-tier structures, to regulate vaporization-mediated droplet dynamics. Our study provides insights into the phase change of supercooled water droplets, which will facilitate the design of environment-controlled water surface interaction systems for advanced cooling, passive anti-icing, as well as cloud freezing and climate physics[36,54].

## Methods

### Surface fabrication and characterization

We conducted experiments using surfaces with diverse structures and materials. These surfaces include: (i) Aluminum (Al) surfaces featuring micropillars/grooves coated with boehmite nanoblades (refer to as samples S1, S2, and S3 depending on the spacing or height of the microstructures), fabricated through direct laser writing using a picosecond laser (Edgewave PX100, Germany), followed by chemical etching[45]; (ii) Al surfaces featuring conformal boehmite nanoblades (sample Na), exclusively achieved by chemical etching[45]; (iii) Al surfaces featuring microlattice coated with nanostructures (sample La), fabricated by laser ablation using a nanosecond fiber laser source (HM20, Han's Laser, China), followed by chemical etching[46]; (iv) Al surfaces featuring macrogrooves coated with nanostructures (sample Ma), fabricated by laser ablation using a nanosecond fiber laser source (HM20, Han's Laser, China), followed by chemical etching[46]; (v) Copper (Cu) surfaces with hierarchical copper oxide (CuO) nanowires (sample Hi), fabricated by direct laser writing using a picosecond laser (Edgewave PX100, Germany), followed by thermal oxidation[55]; (vi) Silicon (Si) surfaces with re-entrant microgrids (sample Re), fabricated via photolithography and etching[16]. All surfaces, except for samples Hi and Re, were modified with heptadecafluorodecyltrimethoxy silane (TCI America, CAS #: 83048-65-1) via vapor phase deposition to achieve superhydrophobicity[49]. Sample Hi attained superhydrophobicity after an approximately 2-week-long atmospheric exposure due to adsorption of volatile organic compounds (VOCs)[55]. Sample Re, characterized by T-shaped re-entrant microstructures, exhibited apparent hydrophobicity[16]. The surface morphologies were imaged by scanning electron microscopy (SEM, JEOL JSM-6490) and optical microscope (Nikon ECLIPSE Ci), showing a wide range of structure length scales (~100 nm to ~1 mm). All surfaces displayed water repellency with an apparent advancing contact angle $\theta_a^{app} > 160°$ and an apparent receding contact angle $\theta_r^{app} > 140°$, as measured by a goniometer (Biolin Scientific Theta Lite, with backlighting). See Supplementary Fig. 4, Table 1, and Section S2 of Supplementary Information for detailed surface fabrication and characterization.

### Depressurization setup and process

We employed a customized vacuum chamber (Fig. 1a and Supplementary Fig. 1a) for conducting low-pressure droplet freezing experiments. Initially, a deionized water droplet was dispensed onto a horizontal superhydrophobic substrate within the chamber using a pipette (Thermo Fisher Scientific). Following a 2-min equilibration period at standard room temperatures and pressures, the mechanical pump (Agilent DS 602 Rotary Vane Pump) was activated to evacuate the chamber from atmospheric pressure (101 kPa) to approximately 100 Pa within 15–20 s (Fig. 1b, pressure measured by a manometer, Baratron MKS 627H), after which the droplet started to freeze. The high depressurization rate (-10 kPa/s) effectively reduces the amount of non-condensable gases (NCGs) during droplet freezing, thereby minimizing their potential impact on vaporization. For further information regarding the chamber setup and depressurization procedure, please refer to Section S1 of the Supplementary Information.

### Visualization of droplet dynamics

Droplet freezing and jumping dynamics were recorded using high-speed cameras (PCO.DIMAX CS3 and Photron Fastcam Nova S12) coupled with a microlens (up to 12X magnification) through optical windows. The capture rate reached up to 20000 fps. For visualization purposes, back, front, and top lighting (Thorlabs M470L5-C1) were employed as required. Thermal imaging was conducted using an infrared camera (Teledyne FLIR SC7700BB) through a side-view sapphire window, with a capture rate of up to 500 fps. The infrared camera was equipped with a 1X lens. The jumping velocity and angle at the detachment moment were determined by analyzing the center-of-mass trajectory of the droplets using a custom imaging processing algorithm (MATLAB script, MathWorks) previously presented in our published studies[49]. Further details regarding droplet dynamics imaging and data processing can be found in Section S1 and Section S7 of the Supplementary Information.

## Data availability

Source data are provided with this paper. The video data supporting the manuscript experimental figures have been deposited in a figshare database under https://doi.org/10.6084/m9.figshare.25150778[56]. The data that support the findings of this study are also available from the corresponding author upon request.

## Code availability

The codes that support the findings of this study are available from the corresponding authors upon request.

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

## Acknowledgements

This work was financially supported by the Research Grants Council of Hong Kong under the General Research Fund (16213721, S.Y.), the Project of Hetao Shenzhen-Hong Kong Science and Technology Innovation Cooperation Zone (HZQB-KCZYB-2020083, S.Y.), and the International Science and Technology projects of Huangpu District of Guangzhou City (2021GH03, S.Y.). The authors appreciate Prof. Weihong Li at City University of Hong Kong for his initial support in the experimental setup. Special thanks to Dr. Wei Ma, Dr. Ali Safari, Dr. Frank Liu, Ms. Yin Zhang, Mr. Chen Xu, and Mr. Zexi Li at Hong Kong University of Science and Technology for their kind support in surface fabrication, chamber setup,

and visualization. The authors thank Prof. Fuqiang Chu at the University of Science and Technology Beijing and Prof. Shinan Chang at Beihang University for the helpful discussion. X.Y. thanks Shujuan Tan for her understanding and support during this work. X.Y. and S.Y. especially appreciate Prof. Nenad Miljkovic at the University of Illinois at Urbana-Champaign for his generous support.

## Author contributions

X.Y. and S.A. contributed equally to this work. X.Y., S.A. and S.Y. conceived the idea for the work. X.Y. and S.Y. designed and guided the experiments. S.A. and X.Y. performed experiments with the help of S.C., Y.C., N.L., W.W., C.C., and Y.Y. G.Z., X.Y. and M.M. fabricated samples for experiments. D.S. and C.C. performed surface characterization. X.Y. and S.A. performed the modeling. S.A. and X.Y. processed and analyzed the data. X.Y. and S.A. wrote the manuscript. S.Y. and H.Q. edited the manuscript. S.Y. supervised the work.

## Competing interests

The authors declare no competing interests.
