## [Peer Review File · Nature Communications]

REVIEWER COMMENTS

Reviewer #1 (Remarks to the Author):

The authors investigate experimentally the effect of vaporization on droplet motion from a freezing supercooled droplet in a low pressure environment. They attribute the momentum transfer from the vapor to the droplet due to recalescence-intensified vaporization at the droplet free surface. Such vaporization can be strong enough to deform the droplet and promote self-propelled jumping from non-wetting superhydrophobic surfaces. They investigate the effects of vaporization, freezing recalescence, and liquid-solid interaction on vaporization momentum and jumping velocity. The experiments are clear—top and side view high speed imaging with optical and infrared cameras—and they compare the detachment time scale to the capillary time and they analyze the effect of droplet radii on the jumping droplet velocity and find a square root dependence. They also look at the effect of nucleation freezing location on the jumping angle of the droplet.

For context, which the authors note in their introduction, there has been previous work done on sessile droplets and supercooled freezing droplets that self-jump on superhydrophobic surfaces in low pressure environments, which is driven by vaporization, and an analysis on the effects of surface texture on the overpressure build up beneath the droplet (DOI: 10.1038/nature15738). In the present paper under review here, the authors state that the mechanism they observe is distinct from that which was previously reported. Specifically, the authors show that vaporization alone is strong enough to deform the droplet interface and also overcome surface tension and gravitational forces. Very recent experimental work, which the authors also note, has been published where they analyzed the effect of supercooling on the pressure change near the interface of a supercooled freezing droplet in a low pressure environment (~100 Pa to 1000 Pa; DOI: 10.1038/s41567-023-01946-3), modelled it, and proposed superhydrophobic surface designs to prevent such strong vaporization from driving wetting transitions on non-wetting surfaces and also identify surface textures that promote what is termed here as “self-jumping”.

I believe that the paper could benefit from more clearly stating what was novel—in my opinion the analysis on jumping height and direction, which to the best of my knowledge has been observed but never explored with such depth—along with some different surface geometries on their influence on jump angle would lead to more universal understanding, as is typical for this journal. My comments below reflect this and are hopefully helpful.

- Clarification: The modeling in the abovementioned paper (DOI: 10.1038/s41567-023-01946-3) wasn't limited to an overpressure analysis, but was rather based on evaporation into a vacuum environment containing non-condensable gases accounting for both kinetic and diffusive terms. I

think it would be important to emphasize the main findings here in the abstract/introduction, namely, that an analysis was done especially on the dynamics of droplet self-jumping, i.e., angle, velocity, and not on the magnitude of the pressure build up (overpressure) or the magnitude of momentum transfer from vaporization of a supercooled freezing droplet.

- Figure 4: Was the effect of nucleation location (θ_n) and jumping angle (θ_j) only done for one surface type? Is this universal to different surface types? The surface in panel (c) also doesn't look like a superhydrophobic surface.

- Comparing the internal capillary time scale to the freezing time scale implies that droplet oscillation, or conversion of kinetic to potential (surface) energy is important in the jumping process. The problem I see with this though is that when the droplet partially solidifies, you do not rectify kinetic energy back from the surface deformation. I can't follow that analysis. Can the authors clarify through further theoretical analysis or experiments the mechanism they envision here? I can imagine a clear trend in detachment time and "capillary time", which depends only the droplet size and has nothing to do with capillary forces.

- I would expect that the jumping velocity increases initially with droplet size (radius R), the driving force being proportional to area (R^2), the resisting term being proportional to the contact line length (R), however, at a certain point, the droplet weight should prevent further increase in velocity as the gravitational force scales as R^3 . Can the author clarify where the cut-off should happen? I don't believe that v_j should scale like $R^{0.5}$ forever. I'm also curious to know if a "sticker" the superhydrophobic surface plays a role in preventing such ballistic transport, for example, by local dissipation of energy through contact line/meniscus deformation. I don't see why such a narrow range of Bond numbers was investigated.

- Other work has investigated the role of the inherently asymmetric recalescence freezing process— nucleation only happens at one point and freezing progresses towards supercooled water—itsself on momentum transfer without a substrate (DOI: 10.1103/PhysRevFluids.8.L021601). I would recommend discussing this work in the context of the present manuscript.

- I had a bit of trouble following the work as the abstract states that the present finding was: "This momentum is attributed to the counteraction of recalescence-intensified interfacial vaporization over the droplet's free surface, which is distinct from the previously identified overpressure underneath the droplet." While in the introduction they write, "While the recalescence force is a significant factor, the mechanism by which the recalescence force gives rise to the droplet's out-of-plane jumping remains elusive." To me this implies either that the novelty is focusing on either

interfacial vaporization without a substrate nearby (overpressure is not important) or overpressure is important. I would recommend trying to write this a bit clearer for the readers.

Minor comments:

- Reference 34 is missing a journal name.

Reviewer #2 (Remarks to the Author):

The authors show that a mismatch in vaporization momentum during the recalescent (i.e., stage one) freezing of a droplet on a superhydrophobic surface can result in spontaneous jumping when the system is within a vacuum chamber pumped down to ~ 100 Pa. While previous works have shown the spontaneous jumping of droplets from superhydrophobic surfaces in vapor chambers, this was a separate “overpressure” mechanism where the vapor pressure was building up in the underlying surface structure due to evaporation. In contrast, here the vaporization force is top-down and is due to the recalesced fraction of the droplet being warmer (and thus evaporating faster); further, the direction of jumping depends on the direction of the recalescent freeze front.

I especially appreciated the clever control experiments to ensure that the jumping mechanism was the recalescence force, rather than overpressure. Namely, showing that the droplets dramatically deform and slide even on a liquid-impregnated surface (which removes the air pockets causing overpressure), and that the jumping kinetics did not change with varying the underlying surface structure. Showing that the data matches well with the scaling for the recalescence force ($v \sim R^{0.5}$) and poorly for the overpressure force ($v \sim R^{.25}$) is also a nice touch.

However, all this being said, I cannot recommend this report for consideration in Nature Communications. For the sole reason that another report, published only months ago in Nature Physics by another group, has already discovered, measured, and theoretically analyzed this recalescence force for droplets on a SHPB surface in a vacuum chamber (Ref. 33). Further, this existing report also reported on the possibility of two other types of wetting transitions during recalescent freezing (impalement and suffusion) depending on the surface structure, which this present manuscript does not focus on. While I do think there are some creative approaches to looking at the recalescent force in this present manuscript, especially the aforementioned control cases to rule out overpressure, scaling analysis of jumping speed, and focus on jumping

directionality, these are incremental and not revolutionary given that the majority of the paper is covering similar ground to Ref. 33. Given that Ref. 33 was published only months ago, it is obvious that the research projects on discovering the recalescence force were prepared in parallel, and it is bad luck that Ref. 33 got there first. Given the fresh perspective of the present work on the recalescence force, it would still be suitable for publication but in a lower-impact journal.

1) I don't understand how the high depressurization rate is related to better removing NCGs. To my knowledge, the amount of NCGs in the chamber is purely related to the final pressure, not the rate of depressurization. Please elaborate.

2) The scaling analysis is interesting and should have stand-alone equations that are more clearly developed. It was hard to compare the graphs to the scaling theory without this.

3) When submitted to a different journal, the authors should devote at least an entire paragraph of discussion contrasting their contributions to Ref. 33. Here, they do so very briefly in a couple of sentences in the introduction, but it's not sufficient. Perhaps a discussion paragraph after the results have been presented would be the most fruitful way to compare and contrast the two similar works.

4) The authors vary the surface structure to show it has no effect on the jumping speed, but according to Ref. 33, aren't there certain choices of surface structure that would result in impalement upon recalescence rather than jumping? Did the authors ever see this?

Reviewer #3 (Remarks to the Author):

Comments to "Unraveling the Role of Vaporization Momentum in the Self-Jumping of Freezing Supercooled Droplets at Reduced Pressures"

The authors reproduced the previously observed self-detaching of freezing sessile droplets from superhydrophobic surfaces in this work. Importantly, the authors proposed a much more reliable "vaporization momentum" mechanism than the previous "overpressure" theory behind the droplet self-detaching phenomenon. In contrast to the fitting error by the previous "overpressure" theory, the new "vaporization momentum" mechanism can quantitatively and precisely predict the self-

jumping velocity of the freezing droplets. Particularly, the analysis of droplet self-jumping direction further enhanced the significance of the “vaporization momentum” mechanism, which was missing in the previous studies.

There is however an important point which needs further discussion from the authors in order to help the readers understand the difference between the previous “overpressure” theory and the new “vaporization momentum” mechanism. The “overpressure” theory was based on the analysis of trampolining of droplets (not freezing ones) under low pressure (ref. 12 in the main text), and further used to explain the self-jumping of freezing droplets on superhydrophobic surfaces. As such, it is possible that “overpressure” could dominating self-jumping of non-freezing droplets, while “vaporization momentum” play the determining role in self-jumping of freezing droplets. The authors are highly recommended to provide extended comments on this regard before the publication of the work.

Overall, the results of presented in the manuscript are clear and convincing. The manuscript is well structured with good readability. I would suggest minor revision to further address the differences between “vaporization momentum” and “overpressure” before accepting the current manuscript.

Zhiliang Zhang

Response to Reviewers

Manuscript ID: NCOMMS-23-25686

Article Title: Unraveling the Role of Vaporization Momentum in Self-Jumping Dynamics of Freezing Supercooled Droplets at Reduced Pressures

We appreciate the helpful comments from the editorial office and reviewers. We believe that we have addressed all the comments, through which we have substantially improved the quality of our manuscript.

To address the reviewers' comments, we have performed additional experiments and analyses. Specifically, we have: 1) significantly broadened the droplet size range and variation of substrate surfaces to confirm the generalized understanding and scaling relationship of droplet jumping dynamics; 2) statistically confirmed the jumping direction as a function of angular position of ice nucleation; 3) experimentally demonstrated the effects of surface pinning/adhesion on droplet jumping velocity and direction; 4) clarified the relationship between detachment time, freezing time, and capillary time; 5) added detailed discussion of our findings in the context of previous important studies, including self-propulsion of freezing airborne droplets, recalescence induced wetting transition, and overpressure induced droplet trampolining and detaching; 6) identified size-defined regime of vaporization-momentum governed self-jumping in contrast to other droplet transport phenomena; 7) clarified other important aspects.

Particularly, to **enhance the novelty** of our work, we have added discussion along with additional work to highlight our contributions in the context of previous studies (*Nature*, 527, 82-85, 2015; *Nature Physics*, 1-7, 2023; *Phys. Review Fluids*, 8, L021601). In addition to developing a generalized understanding of vaporization momentum-driven self-jumping dynamics of supercooled water droplets on superhydrophobic surfaces, which includes a scaling analysis of the jumping velocity, we have presented new findings of droplet-size defined regimes (**a regime map**) to better identify self-jumping in contrast to other low-pressure-induced transport phenomena of sessile water droplets.

Below we have provided point-to-point responses (**in black**) to reviewers' comments (**in blue**). Revised text in the manuscript or Supporting Information is highlighted **in yellow**. We are looking forward to feedback from the reviewers.

Response to Reviewer # 1

Comments: The authors investigate experimentally the effect of vaporization on droplet motion from a freezing supercooled droplet in a low-pressure environment. They attribute the momentum transfer from the vapor to the droplet due to recalescence-intensified vaporization at the droplet free surface. Such vaporization can be strong enough to deform the droplet and promote self-propelled jumping from non-wetting superhydrophobic surfaces. They investigate the effects of vaporization, freezing recalescence, and liquid-solid interaction on vaporization momentum and jumping velocity. The experiments are clear—top and side view high speed imaging with optical and infrared cameras—and they compare the detachment time scale to the capillary time and they analyze the effect of droplet radii on the jumping droplet velocity and find a square root dependence. They also look at the effect of nucleation freezing location on the jumping angle of the droplet.

For context, which the authors note in their introduction, there has been previous work done on sessile droplets and supercooled freezing droplets that self-jump on superhydrophobic surfaces in low pressure environments, which is driven by vaporization, and an analysis on the effects of surface texture on the overpressure build up beneath the droplet (DOI: 10.1038/nature15738). In the present paper under review here, the authors state that the mechanism they observe is distinct from that which was previously reported. Specifically, the authors show that vaporization alone is strong enough to deform the droplet interface and also overcome surface tension and gravitational forces. Very recent experimental work, which the authors also note, has been published where they analyzed the effect of supercooling on the pressure change near the interface of a supercooled freezing droplet in a low pressure environment (~100 Pa to 1000 Pa; DOI: 10.1038/s41567-023-01946-3), modelled it, and proposed superhydrophobic surface designs to prevent such strong vaporization from driving wetting transitions on non-wetting surfaces and also identify surface textures that promote what is termed here as “self-jumping”.

I believe that the paper could benefit from more clearly stating what was novel—in my opinion the analysis on jumping height and direction, which to the best of my knowledge has been observed but never explored with such depth—along with some different surface geometries on their influence on jump angle would lead to more universal understanding, as is typical for this journal. My comments below reflect this and are hopefully helpful.

Response: We appreciate the reviewer for the comments. They are very constructive and helpful, by addressing which we have developed a more universal understanding of vaporization-mediated droplet dynamics and better shaped the novelty of our work in the context of previous studies. Here, we summarize the **enhanced novelty of our revised work:**

1) Complementary insights into the vaporization momentum-driven self-jumping dynamics of supercooled water droplets on superhydrophobic surfaces. In previous studies, the droplet detachment was either attributed to localized overpressure built underneath the droplet (*Nature*, 527, 82-85, 2015), or to the recalescence force that typically develops progressively from the droplet's free surface to the substrate (*Nature Physics*, 1-7, 2023). However, neither the overpressure nor the recalescence force theories could well explain our observation of consistent self-jumping from various structured surfaces, as the bottom-up overpressure is highly dependent on surface structures, while the top-down recalescence force induces liquid impalement into structures and acts against droplet jumping. We revealed that the intricate droplet-substrate interactions during droplet freezing could re-direct the vaporization momentum to propel droplet self-jumping, regardless of the surface structures and the top-down recalescence forces. Our study clarifies the role of vaporization momentum and overpressure and presents new insights into droplet dynamics at reduced pressure.

2) Scaling analysis of droplet jumping dynamics. Despite their significance in advancing the comprehension of mass and energy transport under extreme conditions, the dynamics of self-jumping phenomena, i.e., jumping velocity and direction, *remain unexplored* in previous studies. We resolved the jumping dynamics and their interplay with droplet sizes and freezing processes, and developed a scaling relationship for a largely extended range of droplet sizes (ranging from 0.6 to 2.8 mm) and a significant variation of surfaces (including **11 differing surfaces** ranging from nano-, micro-, micro/nano-, macro-structured, and re-entrant surfaces, see our responses to Comment # 2 below). This strengthened the universality of our reported vaporization-momentum-governed transport phenomena.

3) New findings of droplet-size defined regions where the vaporization momentum governs droplet dynamics. Through additional efforts, we have developed a **regime map** (see our responses to Comment # 4 below) to identify the vaporization momentum-driven self-jumping of supercooled droplets in contrast to other low-pressure-induced transport phenomena of sessile

water droplets (i.e., evaporative drying of small droplets, overpressure-initiated Leidenfrost effects of large droplets). The regime map helps better identify the vaporization-momentum-induced self-jumping of freezing water droplets and offers a complete picture of droplet transport dynamics under reduced pressure.

Below we provide point-to-point responses to the specific comments of the reviewer.

1) Clarification: The modeling in the abovementioned paper (DOI: 10.1038/s41567-023-01946-3) wasn't limited to an overpressure analysis, but was rather based on evaporation into a vacuum environment containing non-condensable gases accounting for both kinetic and diffusive terms. I think it would be important to emphasize the main findings here in the abstract/introduction, namely, that an analysis was done especially on the dynamics of droplet self-jumping, i.e., angle, velocity, and not on the magnitude of the pressure build up (overpressure) or the magnitude of momentum transfer from vaporization of a supercooled freezing droplet.

Response: We greatly appreciate the reviewer's comments and suggestions. We have carefully revised the manuscript to clarify the focus and contribution of our work in contrast to previous important studies (*Nature Physics*, 1-7, 2023; *Nature*, 527, 82-85, 2015; *Phys. Review Fluids*, 8, L021601). Our work is further differentiated from previous studies through additional work and new findings about jumping dynamics and size-defined regions where vaporization-momentum governs droplet dynamics, as will be detailed in our responses to the subsequent comments.

Thanks to the reviewer's suggestion, we have revised the title of our manuscript to highlight our focus on droplet jumping dynamics. The **revised title** is: *Unraveling the Role of Vaporization Momentum in Self-Jumping **Dynamics** of Freezing Supercooled Droplets at Reduced Pressures*.

Furthermore, we have **revised the abstract** as follows to focus on vaporization-momentum-governed droplet jumping dynamics. We also added our new findings of the size-defined regime map to further enhance the novelty of our work. The **revised text in the abstract** is **highlighted in yellow** as follows:

Complex phase change and transport phenomena arise when sessile water droplets are exposed to low pressure. These intriguing processes have attracted growing interest due to their

fundamental significance in various industries. Here, we investigate the freezing and self-jumping dynamics of supercooled water droplets on superhydrophobic surfaces, induced by a remarkable vaporization momentum, in a low-pressure environment (e.g., ~100 Pa). The vaporization momentum arises from the progressive recalescent freezing that intensifies interfacial vaporization over the droplet's free surface, subsequently induces droplet compression, collision, and finally self-jumping. By incorporating the physics of liquid-gas-solid phase changes involving vaporization, freezing recalescence, and liquid-solid interactions, we resolve the vaporization momentum and droplet dynamics, revealing a size-scaled jumping velocity and a nucleation-governed jumping direction with minimal reliance on tested surface structures, in contrast to previously identified overpressure-mediated droplet launching. Moreover, we establish a droplet size-defined regime map that distinguishes the vaporization momentum-dominated self-jumping from other depressurization-induced transport phenomena of sessile supercooled water droplets, including the evaporative drying of small droplets and the overpressure-initiated levitation of large droplets. Our findings underscore the previously uncharted size dependence of supercooled droplet dynamics and may facilitate the design of environment-controlled water-surface interaction systems for passive anti-icing, advanced cooling, cloud freezing and climate physics, particularly in the context of aerospace and deep-space exploration.

Lastly, we have **revised the introduction** to discuss more on the droplet jumping dynamics and to **highlight our focus** is to unveil the fundamental role played by vaporization momentum in shaping the self-jumping dynamics. We also highlighted **our key contributions** in contrast to previous studies are not only the quantified jumping dynamics but also the fate of droplets with respect to droplet sizes, with less dependence on surface structures (pages 3-4 of the revised manuscript):

The unexplored driving force other than overpressure for droplet self-launching has prompted researchers to examine the role of vaporization during progressive recalescence.¹ Recalescence has been demonstrated to elevate the surface temperature, introduce intensive vaporization, and exert forces on the droplet.² Recent endeavor has identified the recalescence-induced reaction force, referred to as the propulsion force³ or recalescence force,¹ which acts on the freezing droplets and plays a crucial role in shaping their behaviors. This is exemplified by the remarkable Cassie-Baxter-to-Wenzel wetting transition of freezing droplets on textured superhydrophobic

surfaces,¹ as well as the intriguing acceleration exhibited by airborne droplets.^{3,4} However, for a sessile droplet, the recalescence force develops in a progressive top-down manner, from the droplet's free surface to the substrate, and tends to induce droplet impalement into substrate structures.¹ This opposes the localized overpressure that typically separates the droplet from the substrate.⁵ The paradoxical outcomes (impalement or jumping) of these two mechanisms highlight the lack of a comprehensive understanding of supercooled droplet freezing and the challenges in predicting the intricate dynamics associated with droplet jumping.

In this study, we investigate the self-jumping dynamics of supercooled, freezing droplets on superhydrophobic surfaces at low pressures (~100 Pa). We reveal a significant counteractive momentum, termed vaporization momentum, resulting from the progressive recalescence of the droplets. We demonstrate that this vaporization momentum surpasses the overpressure generated underneath the droplet and leads to compressive deformation followed by jumping. In contrast to the previous work on self-launching,⁵ wetting transition,¹ and self-propulsion of freezing droplets,^{3,4} the focus of this study is to unveil the fundamental role played by vaporization momentum in shaping the self-jumping dynamics, considering their intricate interplay with droplet size, freezing dynamics, droplet deformation, and substrate-droplet interactions. Particularly, a regime map is developed to demonstrate the findings of droplet-size defined regimes of depressurization-induced transport phenomena observed in supercooled droplets.

We have also **added a discussion** to page 16 of the revised manuscript to summarize our findings:

While previous studies^{1,5} have delicately elucidated the effects of surface structures in droplet dynamics (self-trampolining, suffusion, wetting transition, and expulsion),⁵ our current work focuses on droplet self-jumping from impalement-resisting superhydrophobic surfaces, and highlighted the crucial role of droplet size rather than surfaces structures in jumping velocity and the size-defined fate of droplets at reduced pressures (evaporative drying, self-jumping, and overpressure-initiated levitation).

2) Figure 4: Was the effect of nucleation location (θ_n) and jumping angle (θ_j) only done for one surface type? Is this universal to different surface types? The surface in panel (c) also doesn't look like a superhydrophobic surface.

Response: We thank the reviewer for the helpful comments. The experiments regarding nucleation and jumping directionality were done on surfaces we presented in Figure 3c, including the grooved and pillared surfaces. Based on the statistics of our extensive experiments, we believe the mechanisms governing nucleation and jumping direction are universal in our cases.

To further confirm this observation, we conducted **additional experiments** using a greater variety of surfaces having differing surface structures. The length scales of surface structures span a wider range from nanoscale (~ 100 nm) and microscale (~ 10 μm to ~ 100 μm) to macroscale (\sim mm). The morphology of the structures varies from randomly distributed nanoblades, micropillars, microgrooves, re-entrant microgrids, microhills with nanowires, microlattice, to macrogrooves (**Figure R1** and **Table R1**). All surfaces have nanostructures except the re-entrant surfaces. The additional data consistently demonstrate that the jumping angle is strongly correlated with the nucleation site angular position (**Figure R2a**) and the nucleation site distribution follows the Gaussian distribution (**Figure R2b**). Based on the statistics of our extensive experiments using 8 additional surfaces, we believe the mechanisms governing nucleation and jumping direction are universal in our cases.

We would like to emphasize that the superhydrophobic surfaces we used here have uniform wettability such that the vaporization momentum dominates the droplet dynamics. In particular cases, the local high-energy spots or defects could introduce pinning forces to the droplet, and thus can alter the jumping direction through the pivot effects.⁶ This is similar to the case of coalescence-induced droplet jumping on bi-philic surfaces,⁶ where the local pinning causes a deviation in the jumping direction, shifting it from an out-of-plane trajectory to an in-plane one. Furthermore, large surface structures (e.g., the ratio of the droplet size to the length scale of the structures is greater than 5) may result in non-uniform counteractive forces that interfere with the droplet bouncing dynamics,⁶ thus violating the linear dependence of jumping direction on nucleation angular position. We have demonstrated the effects of surface pinning on jumping direction in the detailed response to Comment # 4 from the reviewer. In our current study, we eliminated the effects of local pinning and heterogeneous structures to focus on the vaporization momentum-dominated droplet jumping, and the surfaces used for the experiments have uniform wettability and structures.

To address these important points, we have **updated Figure 4 with Figure R2** with data for more substrates. The details on the new surfaces (Figure R1 with Table R1) were added to Section S2

of the revised Supporting Information. We have also added a discussion to page 16 of the revised manuscript, with reference to a new section added to the revised Supporting Information (Section S12):

Furthermore, while our current study demonstrates that the jumping dynamics are primarily governed by freezing dynamics and show minimal dependence on the tested surface structures, surfaces with heterogeneous adhesion and structures may interfere with droplet-substrate interaction and impact the jumping dynamics (see Section S12, Supporting Information).

As for Figure 4c, we attribute the seemingly small contact angle of the droplet to the front lighting (i.e., the LED and camera were on the same side, Figure 1a) that failed to illuminate the contact lines. While in the standard contact angle measurement (Biolin Scientific Theta Lite), back lighting was used and optimized to ensure precise contact angle measurement (**Figure R3**). Furthermore, the droplet was shrinking due to evaporation and the contact angle was close to the receding contact angle.

To clarify this point, we have replaced Figure 4c with sharper-contrast photographs (**Figure R4**). We have also added a sentence to page 4 of the revised manuscript: *The superhydrophobic substrate has an apparent advancing contact angle $\theta_a^{app} \approx 160^\circ$ and an apparent receding contact angle $\theta_r^{app} \approx 156^\circ$ (Biolin Scientific Theta Lite with back lighting).*

Figure R1. SEM of surfaces with differing surface structures used for additional droplet freezing and jumping experiments. (a) Nanostructured Al surface, labeled surface Na. The nanostructures have a characteristic size of 100-200 nm. (b) Macro-grooved Al surface, labeled surface Ma. The triangle grooves have a pitch of 1 mm and a depth of 1 mm. (c) Micro-latticed Al surface, labeled La. The spacing and depth of the square lattices are 700 μm and $\approx 100 \mu\text{m}$, respectively. (d) Hierarchical CuO nanowire surface, labeled Hi, consisting of microhills covered by CuO nanowires. The pitch and height of the microhills are 40 μm and $\approx 60 \mu\text{m}$, respectively. The CuO nanowires have a diameter of $\approx 100 \text{nm}$. (e) Re-entrant Si

surface, labeled Re. The side-to-side width and height of the hexagon lattices are $200 \mu\text{m}$ and $\approx 23 \mu\text{m}$, respectively. The wall of the lattice has a T-shape with a width of $\approx 5 \mu\text{m}$ at the top and $\approx 3 \mu\text{m}$ at the bottom. See Table R1 for the contact angles of the surfaces. See our previous studies for more details on the fabrication and characterization of the superhydrophobic Al surfaces,⁷ CuO nanowire surfaces,^{6,8} and re-entrant Si surfaces.⁹

Table R1. Apparent advancing contact angle (θ_a^{app}) and apparent receding contact angle (θ_r^{app}) of the superhydrophobic surfaces. Samples S1-S3 are characterized in the manuscript.

Surfaces	θ_a^{app} ($^\circ$)	θ_r^{app} ($^\circ$)
Nanostructured Al (Na)	163 ± 1.1	160 ± 1.5
Macrogrooved Al (Ma)	158 ± 1.8	155 ± 2.2
Microlatticed Al (La)	161 ± 1.6	159 ± 1.8
Hierarchical CuO (Hi)	163 ± 1.5	158 ± 2.4
Re-entrant Si (Re)	161 ± 2.0	144 ± 4.5

Figure R2. (a) Jumping angle (θ_j) as a function of the angular position of the ice nucleation site (θ_n), showing a linear trend, i.e., $\theta_j = -\theta_n$. (b) The relative frequency (Ψ) of the jumping angle (θ_j), fitted using a Gaussian function (dotted curve), $\phi = \frac{1}{43.3\sqrt{2\pi}} \exp[-0.5(\theta_j/43)^2]$, with a mean of $\approx 0^\circ$ and a standard deviation of $\approx 43^\circ$. The errors of θ_j associated with imaging processing are estimated to be $< 5^\circ$, and the errors of θ_n associated with measurement are estimated to be $< 2^\circ$. The total sample number for evaluating Ψ is 101.

Figure R3. Measurement of the apparent static contact angle of a water droplet (6 μL) on the superhydrophobic micro-grooved surfaces.

Figure R4. An example demonstrating the relationship of jumping angle (θ_j) and the angular position of the nucleation site (θ_n). The nucleation site is marked by a hexagon.

3) Comparing the internal capillary time scale to the freezing time scale implies that droplet oscillation, or conversion of kinetic to potential (surface) energy is important in the jumping process. The problem I see with this though is that when the droplet partially solidifies, you do not rectify kinetic energy back from the surface deformation. I can't follow that analysis. Can the authors clarify through further theoretical analysis or experiments the mechanism they envision here? I can imagine a clear trend in detachment time and "capillary time", which depends only the droplet size and has nothing to do with capillary forces.

Response: We thank the reviewer for the helpful and insightful comments. We agree with the reviewer that the surface energy-to-kinetic energy conversion may be altered due to the confinement of the liquid core by the solidified ice shell. **How the liquid hydrodynamics are**

affected by the growing ice shell is definitely crucial for understanding droplet jumping dynamics. As the droplet is only partially solidified, presumably, the confined ice-shell has limited effects on the droplet jumping dynamics, whereas the liquid hydrodynamics still play an important role in the droplet detachment and jumping. To justify our assumption as well as to further elucidate the freezing and detaching dynamics, we would like to first clarify the important aspects below, i.e., the relationship between detachment time, capillary time, and freezing time.

1) Relationship between detachment time and capillary time.

As the reviewer noticed, we have introduced the **detachment time** Δt_d to characterize the time scale during which the droplet *effectively* interacts with the substrate to accumulate and redirect the vaporization momentum. Δt_d is defined as the time duration starting from ice nucleation to droplet-substrate separation.

Previously, we compared the detachment time Δt_d (*rather than freezing time*) with the **capillary time** τ_c , and found a linear relationship ($\Delta t_d/\tau_c \approx 2.2$, or $\Delta t_d \sim R_d^{1.5}$) for the tested droplet sizes ($0.6 \leq R_d \leq 2.8$ mm, see **Figure R5**) and experimental conditions ($P_e \sim 100$ Pa). This comparison is based on the experimental observation that the freezing droplet experiences a compressive deformation under the recalescence-induced propulsion force, reminiscent of the classic droplet impact and bouncing dynamics on superhydrophobic surfaces under inertial and capillary forces. To confirm the scaling of detachment time with capillary time, we have performed **additional experiments** using water droplets having a wider range of radii (0.6-2.8 mm) and substrates having a wider range of structure geometries and length scales (see Figure R1 in our responses to Comment # 1), and demonstrated consistent scaling law ($\Delta t_d \sim R_d^{1.5}$) as seen in **Figure R5**.

We would like to point out, when considering droplet detachment, the dynamic effect is mainly dominated by the hydrodynamics of the liquid core of the droplet. We assumed that, during droplet deformation, the ice shell remains a thin layer of ice-water slurry mixture with a large portion of liquid water during the short-time freezing, thus the hydrodynamics of the liquid core still plays an important role in droplet deformation and jumping. This assumption was justified by our calculations of the thickness and water mass fraction of the ice-water slurry shell at the detaching moment (please see Figure S11b, Section S5 of Supporting Information), showing that for a 16 μ l water droplet around the detachment moment ($t=17$ s), the average thickness of the ice slurry shell

reaches $\approx 47\%$ of the droplet radius, and the averaged ice mass fraction within the ice-water slurry is $< 10.2\%$, which is adequately low such that the ice slurry shell remains a very loose ice-water mixture. To some extent, the hydrodynamics of ice-shelled droplets in the early stage of freezing are similar to those of particle-laden droplets or droplet marbles, which behave as a liquid with a low particle concentration when impacting solid surfaces.¹⁰ To further confirm the assumption, we have performed **additional experiments to capture the freezing dynamics at a higher capture rate** (up to 20000 fps) using a high-speed camera (Photron FASTCAM NOVA S12) with a 12X micro lens. The enhanced temporal resolution allowed us to clearly differentiate the ice and water phases and capture the interaction between the liquid portion with the substrate (**Figure R6**). It was seen that before detaching, the liquid part of the droplet was pushed to deform and then contracted when interacting with the substrate ($t \in [8.2 \text{ ms}, 12.4 \text{ ms}]$, Figure R6). Even at the detaching moment, part of the liquid bottom remains unfrozen ($t < 14.4 \text{ ms}$, Figure R6), as further shown by slight oscillation of the liquid after detaching ($t > 14.4 \text{ ms}$, Figure R6). This clearly indicates the hydrodynamics of the droplet during the short-time deformation process.

As we have highlighted in the manuscript, we must note that the fluid dynamics during freezing recalescence are rather complicated involving the dynamic confinement of supercooled liquid by the fast-spreading ice shell. A more fundamental study is needed to rigorously verify the relationship between detachment time and freezing time by incorporating the crystallization kinetics, ice propagation, and internal flow within the liquid core. We feel this is an interesting and multi-physics topic that is worth exploring in the future. Particle image velocimetry (PIV) might be a useful tool to explore the hydrodynamics within the liquid core during droplet freezing and deforming, though our trials were *not successful* due to short-time physics and limited time window for imaging (Lavisson, 2D-PIV, **Figure R7**). In our current work, the demonstrated scaling of detachment time indicates that the droplet hydrodynamics at least partially govern the compressive deformation and detaching dynamics.

Figure R5. Droplet detachment time Δt_d as a function of the power of droplet radius $R_d^{1.5}$. Surfaces with differing structures (samples S1, S2, S3, Re, Ma, Na, La, Hi) were used as the substrates. Independent experimental data from literature (Lambley, H. et al. *Nat. Phys.*, 2023) which used different structured surfaces (samples C2, D4, D6, star symbols) also follow the same scaling law. The dashed line represents a linear fitting. See Figure R1 and Table R1 for the detailed characterization of the surfaces.

Figure R6. High-speed optical imaging of the droplet freezing, deforming, and self-jumping dynamics. The enhanced temporal resolution allows us to differentiate the ice and water phases and capture the interaction between the liquid portion with the substrate. The initiation of ice nucleation was marked by the yellow diamond, the vaporization repulsion force (F_t) acting on the ice shell was indicated by the yellow

arrow, the ice front was traced by yellow curves, the liquid spreading and retraction during droplet compression was marked by blue arrows, and the jumping velocity (v_j) was marked by the red arrow. A superhydrophobic hierarchical CuO nanowire surface with 40- μ m-spaced microhills⁶ (labeled as surface Hi, see Figure R1) was used as the substrate.

Figure R7. Particle image velocimetry (PIV) of the freezing and jumping droplet. A 450-nm blue laser with a lens were used to generate a light sheet to illuminate the cross-section (perpendicular to the page) of the droplet. Silver-coated glass spheres with a diameter of 1-10 μ m, diluted with DI water at a particle/water mass ratio of 1:2000, were used as the tracing particles. Particle trajectory was shown but with a very limited resolution (marked in the dotted frame) due to the low capture rate (500 fps) and low illumination blocked by the ice shell.

2) Relationship between freezing time and detachment time.

Having demonstrated the detachment time governed by the hydrodynamics of the liquid core, we then clarify the relationship between the freezing time and detachment time. We have observed that the **freezing time** Δt_f , the time needed for the ice crystal to spread over the whole water surface area, which should be even slightly longer than the detachment Δt_d in our experiments. This was supported by the IR and optical time-lapse images shown in Figures 2a and 2c of the manuscript as well as Figure R6, where the ice front had nearly enclosed the droplet at the detachment moment, while a small portion at the droplet bottom remained unfrozen upon jumping.

The freezing time Δt_f can be alternatively estimated by the crystallization kinetics, as we have demonstrated in the Supporting Information (Eq. S16), $\Delta t_f = \pi R_d / v_g$, where v_g is the growth velocity of ice dendrite depending on the supercooling.¹² However, this calculation may not be accurate since the calculation of v_g assumes isotropy in liquid water and one-dimensional ice propagation. Furthermore, v_g is highly dependent on supercooling. In real conditions, the compressive deformation and the resultant internal flow within the liquid core enhance convective heat transfer, and the ice s does not necessarily spread uniformly over the surface (see the irregular

shape of the ice shell defined by the ice front, **Figure R8** or Figure 2c in the manuscript). Consequently, the freezing time could not be well captured by Eq. S16.

The comparable time scale of ice spreading (with a time scale of Δt_f) and compressive deformation (or detachment, with a time scale of Δt_d) of the droplet is likely due to the synchronized freezing and deforming processes. As ice nucleation initiates on the droplet surface, a propulsion force due to intensive vaporization is exerted onto the freezing surface. The net vaporization force can be modeled by $F_t = \pi R_d^2 J_i v_r \sin^2 \phi$ as we have elucidated in Eq. S17 of the Supporting Information, where ϕ is the angular position of the freezing front with respect to the initial nucleation site. As seen in **Figure R9a**, F_t first increases as the ice spreads ($\phi < 90^\circ$), reaching a maximum when half of the droplet surface is covered by ice ($\phi = 90^\circ$), and then decreases as the ice proceeds to enclose the droplet ($\phi > 90^\circ$). Corresponding to the development of net vaporization force applied to the droplet, the droplet experiences fast compressive deformation in the beginning until reaching the maximum deformation, and then the liquid portion is partially relaxed as the net vaporization force starts to release ($\phi > 90^\circ$, **Figure R9b**, corresponding to the freezing dynamics in Figure R6). Therefore, the crystallization development determines the evolution of the applied net propulsion force, thus the deforming dynamics.

Given the comparable time scales of freezing and detachment, in our modeling of the vaporization momentum (Eq. 1 of the manuscript), we used the detachment time to characterize the timescale for momentum transport. In our current study, the out-of-plane jumping is generated due to the redirection of the vaporization momentum by the substrate during the droplet detachment time, therefore, we quantify the jumping velocity at the detaching moment (note that the velocity may change after detachment due to air drag, gravity, asymmetry vaporization, etc) and it is during the detachment time that the substrate interacts with the droplet to re-direct the vaporization momentum for out-of-plane jumping.

Given the comparable time scales of freezing and detachment, in our modeling of the vaporization momentum (Eq. 1 of the manuscript), we used the detachment time to characterize the timescale for momentum transport. In our current study, the out-of-plane jumping is generated due to the redirection of the vaporization momentum by the substrate during the droplet detachment time, therefore, we quantify the jumping velocity at the detaching moment (note that the velocity may

change after detachment due to air drag, gravity, asymmetry vaporization, etc) and it is during the detachment time that the substrate interacts with the droplet to re-direct the vaporization momentum for out-of-plane jumping.

In our current study, we quantify the jumping velocity at the detaching moment (note that the velocity may change after detachment due to air drag, gravity, asymmetry vaporization, etc.), and it is during the detachment time that the substrate interacts with the droplet to re-direct the vaporization momentum for out-of-plane jumping. To simplify the modeling, we assume that the recalescence completes upon detachment given the comparable time scales of freezing and detachment. Therefore, in our modeling of the vaporization momentum (Eq. 1 of the manuscript), we used the detachment time to characterize the timescale for momentum transport.

To address these important points, we have **updated Figure 3d** with additional experimental data shown in Figure R5. The updated figure further confirms the scaling of detachment time in a larger range of droplet radii. To avoid confusion between the capillary time and detachment time, we have changed the axis of Figure 3d from the capillary time to the power function of the droplet radius. Correspondingly, we have **added a discussion** to page 11 of the revised manuscript:

Regardless of surface microstructures (Figure 3c), Δt_d is shown to scale with $R_d^{1.5}$ (Figure 3d), where R_d is droplet's spherical radius immediately before freezing. The prefactor of the Δt_d scaling is consistent with that of the characteristic timescale of an inertia/capillary-dominated oscillating droplet ($\Delta t_d/\tau_c \approx 2.2$),^{39,40} and is slightly shorter than that of the contact time of a low-deformation impact droplet on superhydrophobic surfaces ($\Delta t_d/\tau_c \approx 2.6$),^{40,41} suggesting that the fluid dynamics of the droplet are likely to play an important role in droplet deformation and detachment. Here, $\tau_c = \sqrt{\rho_w R_d^3 / \sigma}$, ρ_w is the water density, and σ is the surface tension of water.

See Section S11, Supporting Information for the timescale analysis.

The detachment time Δt_d is used to characterize the timescale for momentum transport in Eq. 1, during which the substrate interacts with the droplet to re-direct the vaporization momentum for out-of-plane jumping. Recalescence is assumed to complete upon droplet detachment given the comparable time scales of freezing and detachment (see Section S11, Supporting Information for a detailed discussion on the time scales of detachment and freezing).

Furthermore, we have **added a new section** (Section S11) to the revised Supporting Information to discuss the detachment, freezing, and capillary time scales.

Figure R8. IR images showing the irregular shape of the ice shell (light blue) which spreads over the droplet’s free surface (dark blue) as the freezing front progresses. The initial droplet radius is 1 mm.

Figure R9. (a) Calculated net vaporization force F_t as a function of the angular position of the ice front ϕ . A spherical droplet is assumed in the calculation. The maximum vaporization force is reached at $\phi=90^\circ$, as marked by the dashed blue line. (b) Droplet deformation, characterized by the aspect ratio (width to height) of the droplet K_d , as a function of time t . The nucleation and detaching moments are marked by arrows. The maximum deformation is reached before detachment, as marked by the dashed blue line.

4) I would expect that the jumping velocity increases initially with droplet size (radius R), the driving force being proportional to area (R^2), the resisting term being proportional to the contact line length (R), however, at a certain point, the droplet weight should prevent further increase in velocity as the gravitational force scales as R^3 . Can the author clarify where the cut-off should happen? I don't believe that v_j should scale like $R^{0.5}$ forever. I'm also curious to know if a "sticker" the superhydrophobic surface plays a role in preventing such ballistic transport, for

example, by local dissipation of energy through contact line/meniscus deformation. I don't see why such a narrow range of Bond numbers was investigated.

Response: We thank the reviewer for the insightful comments. In our current study, we focused on a certain range of droplet sizes where capillary forces dominate gravitational forces. This facilitated our study of the vaporization momentum and resultant self-jumping dynamics of partially freezing droplets due to the elimination of gravitational forces. We agree with the reviewer that when increasing the droplet size, the scaling relationship will be altered. We appreciate the reviewer's comments, which inspired us to further explore the boundaries (or cut-off sizes of droplets) of the current scaling relationship.

To better identify the regions within which the vaporization momentum and scaling analysis remain valid, we have performed **additional experiments and analyses**. Below we summarize these efforts and results.

1) Validation of the self-jumping dynamics in a wider range of droplet sizes.

We have performed additional experiments with a wider range of droplet sizes (R_d ranging from 0.6 to 2.8 mm, **Figure R10**). We note that the **Bo number has been largely extended** from $Bo=0.08-0.18$ to $Bo=0.05-0.945$. The droplet freezing experiments were conducted following the same procedures demonstrated in the Methods of the manuscript. Furthermore, to examine the effects of surface structures on the velocity scaling, additional surfaces were used as the substrates, which were fabricated by chemical etching, thermal oxidation, laser ablation, or lithography⁶⁻⁹ to extend the variation of the surface structure's length scale (~100 nm to ~1 mm) and topology (nanoblades, microgrids, macro/microgroove, microhills, and T-shaped re-entrance, Figure R1 and Table R1). Our extended experiments further confirm the detachment time scaling (**Figure R10a**), jumping velocity scaling (**Figure R10b**), as well as jumping direction distribution (Figure R2), thus strengthening the proposed self-jumping mechanisms and scaling relationship.

Figure R11 shows the freezing and jumping of a relatively large droplet (~70 μ L). Regardless of the large Bo number (0.945), the measured jumping velocity agrees with the scaling. This indicates that for $Bo < 1$, vaporization repulsion and capillary forces dominate over the gravitational forces, and the velocity scaling remains valid. However, beyond the droplet size range in Figure R10, differing regimes dominate the droplet dynamics. In such cases, the droplet jumping velocity

reduces to ≈ 0 m/s (for significantly large droplets) or the droplet dries before freezing (for significantly small droplets), as we will discuss in the following section.

To strengthen the scaling analysis, we have **updated Figures 3d and 3e** with an extended range of droplet sizes. Furthermore, we have **added a discussion** to page 12 of the revised manuscript:

To validate the scaling relationship, we conducted additional experiments using a greater variety of surfaces having differing surface structures and materials. The length scales of surface structures span a wider range from nanoscale (~ 100 nm) and microscale (~ 10 μ m to ~ 100 μ m) to macroscale (\sim mm). The morphology of the structures varies from randomly distributed boehmite nanoblades (labeled Na), Al microlattice with nanostructures (labeled La), Al macrogrooves with nanostructures (labeled Ma), hierarchical CuO nanowires (labeled Hi), to re-entrant Si microgrids (labeled Re) (see Figure S4 and Table S1, Supporting Information for detailed characterization). In addition, we analyzed recently reported droplet jumping on polymer surfaces with well-defined microstructures (labeled C2, D4, and D6, respectively).¹ All results were in good agreement with the scaling regardless of the surface structures.

Figure R10. (a) Detachment time Δt_d as a function of the power of the droplet radius $R_d^{0.5}$ for substrates with differing microstructures (samples S1, S2, S3, Re, Ma, Na, La, Hi). The dashed line represents the best linear fitting. (b) Droplet jumping velocity v_j as a function of droplet spherical radius R_d for different substrates. The solid line and gray band represent the prediction by the vaporization momentum ($v_j \sim R_d^{0.5}$) and overpressure theory ($v_j \sim R_d^{2.5}$), respectively. The predicted velocity by the overpressure theory reflects the variation of microstructures of samples S1-S3. See Section S10, Supporting Information for detailed

modeling of overpressure. Independent experimental data from literature (Lambley, H. et al. *Nat. Phys.*, 2023)¹ which used different structured surfaces (samples C2, D4, D6, star symbols) also follow the same scaling law. R_d was measured immediately before freezing.

Figure R11. High-speed optical imaging of the droplet freezing, deforming, and self-jumping dynamics for a $\sim 70 \mu\text{L}$ droplet. The initiation of ice nucleation was marked by the yellow diamond, the vaporization repulsion force (F_t) acting on the ice shell was indicated by the yellow arrow, the ice front was traced by the yellow curve, and the jumping velocity (v_j) was marked by the red arrow. A superhydrophobic macro-grooved surface (surface Ma) was used as the substrate. Scale bar: 1.8 mm.

2) Identification of the droplet-size defined region where the vaporization momentum governs droplet dynamics.

To further explore the size effects of droplet self-jumping, we have further pushed the limits of the droplet sizes in our experiments and developed a **size-defined regime map** of depressurization-induced phenomena of sessile water droplets. The identified regimes include: 1) evaporative drying for small droplets, 2) vaporization-momentum-dominated self-jumping for medium-sized droplets, and 3) overpressure-initiated Leidenfrost effects for large droplets (**Figure R12a**). Notably, there exists a lower ($R_{d,l}$) and upper ($R_{d,u}$) boundaries in terms of equivalent droplet radius (R_d) that delineate the regime of droplet freezing and self-jumping. For $R_d < R_{d,l}$, the droplet dries up before freezing (**Figure R12b**). The value of $R_{d,l}$ is determined when the evaporation lifespan (t_{eva}) of a droplet with an initial R_d is shorter than the ice incubation time (t_{inc}) (**Figure R12c**). While for $R_d > R_{d,u}$, the droplet experiences a marginal and dynamic levitation rather than jumping, facilitated by a vapor layer underneath, as well as remarkable fluctuations at the interface

(**Figure R12d**), both of which indicate the characteristics of the Leidenfrost effect.^{11,12} The Leidenfrost effect significantly delays freezing due to enhanced fluid convection caused by the remarkable droplet fluctuations, counteracting the decrease in interface temperature. $R_{d,u}$ is reached when the overpressure force F_{op} overcomes the resistance forces (gravitational force F_g and adhesion force F_a), causing droplet-substrate separation (**Figure R12e**). The theoretical boundaries estimated by force analysis are in good agreement with our experiments.

We believe the new findings of the size-defined regimes better identify the characteristics of the vaporization-momentum-dominated self-jumping, which significantly enhances the novelty of our work. We have **added a new figure** (Figure R12 as Figure 5) with a detailed discussion to the revised manuscript and **an additional section** (Section S13) to the revised Supporting Information. We have presented detailed analyses of the regime boundaries at the end of the response to this comment. Below we listed the changes of our manuscript:

Abstract: ...*Moreover, we establish a droplet size-defined regime map that distinguishes the vaporization momentum-dominated self-jumping from other depressurization-induced transport phenomena of sessile supercooled water droplets, including the evaporative drying of small droplets and the overpressure-initiated levitation of large droplets. Our findings underscore the previously uncharted size dependence of supercooled droplet dynamics and may facilitate the design of environment-controlled water-surface interaction systems for passive anti-icing, advanced cooling, cloud freezing and climate physics, particularly in the context of aerospace and deep-space exploration.*

Main text on pages 15-16 of the revised manuscript: *The resolved dynamics of jumping droplets highlight the significant role of vaporization momentum, which, however, may be diminished or inhibited for significantly small or large droplets. To determine the regime where vaporization momentum governs droplet dynamics, we extensively expanded the range of droplet sizes in our experiments. This led to the identification of distinct regimes, defined by droplet size, in the depressurization-induced phenomena of sessile water droplets. These regimes include evaporative drying for small droplets, vaporization-momentum-dominated droplet freezing and self-jumping for medium-sized droplets, and overpressure-initiated Leidenfrost effects for large droplets (Figure 5a). Specifically, there exist a lower ($R_{a,l}$) and upper ($R_{a,u}$) boundaries in terms of equivalent droplet radius (R_d) that delineate the regime of droplet freezing and self-jumping. For*

$R_d < R_{d,l}$, the droplet dries up before freezing (Figure 5b). The value of $R_{d,l}$ is determined when the evaporation lifespan (t_{eva}) of a droplet with an initial R_d is shorter than the ice incubation time (t_{inc}) (Figure 5c). While for $R_d > R_{d,w}$, the droplet experiences a marginal and dynamic levitation rather than jumping, facilitated by a vapor layer underneath, as well as remarkable fluctuations at the interface (Figure R5d), both of which indicate the characteristics of the Leidenfrost effect.^{11,12} The Leidenfrost effect significantly delays freezing due to enhanced fluid convection caused by the remarkable droplet fluctuations, counteracting the decrease in interface temperature. $R_{d,u}$ is reached when the overpressure force F_{op} overcomes the resistance forces (gravitational force F_g and adhesion force F_a), causing droplet-substrate separation (Figure 5e). Here, gravitational and adhesion forces are incorporated as the droplet has a significantly large size ($Bo > 1$) and a non-spherical shape with increased droplet-surface contact area. The theoretical boundaries are in good agreement with our experiments (see Section S13, Supporting Information for detailed analyses of the regime boundaries).

Figure R12. (a) Regime map illustrating droplet transport phenomena on superhydrophobic substrates in a low-pressure (~ 100 Pa) environment, depicting the relationship between droplet jumping velocity (v_j) and droplet size (R_d). The vaporization-momentum-governed self-jumping regime (jumping) is delimited by the evaporative drying regime (drying) for small droplets and the overpressure-initiated Leidenfrost effect regime (Leidenfrost) for large droplets. Theoretical boundaries are represented by dashed lines. Note, for the evaporative drying regime, R_d is defined as the initial droplet radius immediately before pumping down. For the self-jumping regime, R_d is measured as the droplet radius immediately before freezing. For the Leidenfrost regime, R_d is defined as the equivalent droplet radius of a spherical droplet having an identical volume immediately before droplet levitation. (b) Optical images showing the evaporative drying regime. $t=0$ s corresponds to the start of pumping down. See Video S3. (c) Theoretical determination of the lower boundary ($R_{d,l}$) by comparing the evaporation lifespan (t_{eva}) and ice incubation time (t_{inc}) as a function of the initial droplet radius (R_d). $t_{eva} < t_{inc}$ represents the droplet dries up before freezing. (d) Optical images showing the Leidenfrost effect regime. The overpressure-induced droplet-substrate separation is highlighted and zoomed in in the frame. $t_0 \approx 30$ s represents the pumping-down time before the initiation of droplet levitation. See Video S4. (e) Theoretical determination of the upper boundary ($R_{d,u}$) by comparing the overpressure force (F_{op}) and the sum of gravitational force and adhesion force ($F_g + F_a$) as a function of

equivalent droplet radius (R_d). $F_{op} > F_g + F_a$ represents that the overpressure overcomes the resistance forces and induces droplet-substrate separation. Here, gravitational and adhesion forces are incorporated since the droplet has a significantly large size ($Bo > 1$) and a non-spherical shape with an increased droplet-surface contact area. Sample S1 with a microstructure height of 15 μm was used in the experiments and modeling.

The following details on the development of the regime map have been **added as a new section** to the revised Supporting Information (Section S13):

To further differentiate the overpressure and vaporization momentum, we have significantly extended the range of droplet sizes (initial R_d ranging from 0.2 to 3 mm, equivalent to a volume ranging from 0.0335 to 115 μL) in our experiments and developed a regime map (Figure R12) to identify differing regimes including: (1) evaporative drying for small droplet sizes, (2) vaporization-momentum dominated droplet freezing and self-jumping for medium droplet sizes, and (3) Leidenfrost effects for large droplet sizes (Figure R12a). Specifically, there exists a lower boundary $R_{d,l}$ and an upper boundary $R_{d,u}$ in terms of droplet radius for the droplet freezing and self-jumping regime. For $R_d < R_{d,l}$, the droplet stays on the surface and dries up due to evaporation before freezing (Figure R12b), while for $R_d > R_{d,u}$, the droplet experiences a marginal elevation (but not jumping) by a vapor layer underneath the droplet and remarkable fluctuations at the interface (Figure R12c), characteristic of the Leidenfrost effect.^{11,12} The freezing is largely delayed by the Leidenfrost effect due to the remarkable fluctuations of the droplet that enhance fluid convection and counteract the decrease of interface temperature. To obtain the lower boundary $R_{d,l}$ that separates the evaporative drying and self-jumping regimes, we compared the lifespan t_{eva} and the incubation time for ice nucleation t_{inc} of an evaporating droplet during pumping down. For a given initial droplet radius, evaporative drying occurs when $t_{eva} < t_{inc}$. t_{eva} can be obtained by considering the mass conservation during evaporation. For droplet evaporation in a depinning mode with a constant contact angle θ , which is approximately the case for our superhydrophobic surfaces having a high receding contact angle, the decrease in the droplet volume results from the vaporization over the droplet surface,^{13,14} and thus:

$$\rho dV_d = -JA_d dt, \quad (R1)$$

where $V_d = \pi R_d^3 (1 - \cos \theta)^2 (2 + \cos \theta) / 3$ is the droplet volume as a sphere cap, $A_d = 2\pi R_d^2 (1 - \cos \theta)$ is the area of the droplet's free surface, and $J = \frac{2\beta}{2-\beta} \sqrt{\frac{M}{2\pi R}} \left(\frac{P_d}{\sqrt{T_d}} - \frac{P_e}{\sqrt{T_e}} \right)$ is the vaporization mass flux given by the Hertz-Knudsen-Schrage equation^{15,16} (Eq. S7, Supporting Information) for kinetic-dominated evaporation. Integrating Eq. R1 with a known initial droplet radius R_d gives:

$$\int_0^{t_{eva}} J dt = \frac{\rho R_d (1 - \cos \theta) (2 + \cos \theta)}{2}, \quad (R2)$$

Note, J is a function of the vapor pressure at the droplet surface (P_d) and vapor pressure in the environment (P_e), all of which are time-dependent. Assuming a time-averaged vaporization flux \bar{J} , the droplet lifetime can be solved:

$$t_{eva} = \frac{\rho R_d (1 - \cos \theta) (2 + \cos \theta)}{2\bar{J}}. \quad (R3)$$

To estimate \bar{J} , the transient temperature field within the droplet was simulated via COMSOL 6.0 (Heat Transfer in Fluid and Laminar Flow Physics) incorporating evaporative cooling, achieved by applying a temperature-dependent vaporization heat flux to the droplet's free surface. The modeling exploited an axisymmetric 2D geometry setup as shown in **Figure R13a**. A substrate temperature $T_{sub} = 298.15$ K in equilibrium with the environmental temperature T_e was assumed. The transient environmental pressure P_e (Figure S9, Supporting Information) was inputted as the boundary condition. Simulations were performed for an overall simulation time of 30 s with a time step of 0.1 s. This simulation time covers the time window for our depressurization experiments (Figure 1b of the manuscript). Droplets with varied radius $R_d = 0.2 - 3$ mm and a constant contact angle $\theta = 160^\circ$ were simulated. **Figure R13b** presents the simulated temperature profile within a typically sized droplet, showing $T_d \approx 254.1$ K ($P_d \approx 112$ Pa) at $t = 30$ s after the initiation of evaporative cooling. Simulations for differing droplet sizes ($R_d = 0.2 - 3$ mm) showed a modest variation (2-3 K) of the temperature at the droplet's free surface at $t = 30$ s. The simulated droplet surface temperature is a reasonable estimate when compared to the experimental measurement of the temperature inside the droplet with a similar size (-12°C , Figure 1b of the manuscript) and the derived temperature of the free surface at the freezing moment (-27°C , Eqns. S1-S3, Supporting

Information). \bar{J} is then estimated by $\bar{J} = \frac{1}{\Delta t} \int_0^{\Delta t} J(T_d, P_d, T_e, P_e) dt$ with T_d (and thus P_d) simulated as a function of time t (**Figure R13c**) and Δt approximated by 30 s. Note that an evaporation coefficient^{17,18} $\beta \approx 0.759$ is adopted when evaluating J (see Section S3, Supporting Information).

It should be noted that the simulations did not incorporate changes in droplet size during evaporation and assumed negligible effects of droplet mass depletion on the droplet temperature profile. This assumption is based on the fact that the droplet size does not change significantly during the rapid pumping-down process. The simulations were only used to estimate the droplet surface temperature for the analytical modeling (Eq. R3) and were not performed to obtain the lifetime of evaporating droplets, which is feasible but numerically challenging to include the phase change (mass depletion) at the interface and internal circulation within the droplet. Instead of dealing with the numerical convergence issues, we used the simulations to estimate the surface temperature and to develop a semi-analytical model of the lifetime of evaporating droplets.

Figure R13. (a) COMSOL multiphysics model setup of the droplet evaporation, where the free surface of the droplet is subjected to the evaporation cooling flux $J(T_d)$ as a function of surface temperature T_d . The transient environmental pressure P_e (Figure S9, Supporting Information) was inputted as the boundary condition. Simulation was performed with a contact angle $\theta=160^\circ$ and a time step of 0.1 s. (b) Simulated temperature distribution at time $t=30$ s for a droplet having a radius $R_d=1.5$ mm. Evaporative cooling starts at $t=0$ s. (c) The averaged droplet surface temperature $T_{d,ave}$ as a function of time t . Minimal differences in $T_{d,ave}$ were observed for R_d ranging from 0.2 to 3 mm.

The ice nucleation incubation time t_{inc} corresponds to the moment when the very first nucleus is formed in the droplet. The total number of nuclei is a function of the droplet volume and the ice nucleation rate. The latter is dependent on the temperature field inside the droplet. Here, the

transient temperature field of droplets with different sizes was obtained from the numerical simulation performed above (Figure R13). With a known time-dependent temperature field, the local nucleation rate per unit volume is given by the classical nucleation theory:¹⁹

$$J_n = \frac{A_m}{3} n^{7/3} \sqrt{\frac{A_m \sigma}{\pi k_B T}} D_w \exp \left\{ -\frac{4 \left(\frac{A_m \sigma}{k_B T} \right)^3}{27 \left[\ln \left(\frac{P_w}{P_i} \right) \right]^2} \right\}, \quad (R4)$$

where $A_m = 4\pi r_m^2$ is the surface area of a water molecule with a radius of r_m , $n = \frac{\rho N}{M}$ is the number density of the water, ρ is water density, Avogadro constant N , and molar mass M , $D_w = 1.5 \times 10^{-6} \exp \left(-\frac{2045.0}{T} \right)$ is the self-diffusivity of supercooled water, $\sigma = 0.0291 + 10^{-4}(T - 273.15)$ is the water-ice interfacial tension, P_w is the vapor pressure of supercooled water at temperature T , and P_i is the vapor pressure of ice at an equilibrium freezing temperature of 273.15 K. The condition of ice nucleation is defined when the total number of ice nuclei exceeds unity, given by integrating the local ice nucleation rate J_n over the droplet volume during a period of t_{inc} .¹⁹⁻²¹

$$\int_0^{t_{inc}} \int_{V_d} J_n dV_d dt = 1, \quad (R5)$$

where $V_{d,t}$ is the droplet volume at time t and can be solved by Eq. R1. For a given initial droplet volume V_d , t_{inc} can be found by numerically solving the integration problem above with the input of the time-dependent droplet temperature profile. Given that the droplet temperature is lower than the substrate (Figure R13), we only considered the homogeneous nucleation and neglected the heterogeneous nucleation at the droplet-substrate interface. Figure R12c shows t_{inc} as a function of the initial droplet radius R_d , revealing that a smaller droplet has a longer incubation time for ice nucleation. This is because a smaller surface area or volume leads to a reduced number of ice nuclei. Moreover, the small droplet has a lower temperature that is closer to the substrate temperature due to its lower thermal conduction resistance.

By comparing the ice incubation time and the lifetime of the evaporating droplet, a critical droplet size $R_{d,l}$, below which ice nucleation hardly occurs, can be solved by:

$$t_{eva} < t_{inc}. \quad (R6)$$

The calculated $R_{d,l}$ is shown to be consistent with the experiments (Figures R24a and R24c).

To obtain the upper droplet size boundary $R_{d,u}$ that separates the Leidenfrost effect and the self-jumping regimes, we consider the criteria for the initiation of the Leidenfrost effect, which starts when the overpressure force F_{op} underneath the droplet overcomes the gravity and adhesion of the droplet. Here, gravitational force (F_g) and adhesion force (F_a) are incorporated as the droplet has a significantly large size ($Bo > 1$) and a non-spherical shape with increased droplet-surface contact area. The overpressure force F_{op} , defined consistently with that in our analyses of overpressure (Figure 3e in the manuscript and Eqns. S26-S29 in the Supporting Information), can be modeled by:⁵

$$F_{op} \approx \frac{3\mu\pi R_b^4 J}{h^3 F \rho_v}, \quad (R7)$$

where μ is the dynamic viscosity of vapor, R_b is the basal radius of the droplet, J is the evaporation flux of the droplet, h is the height of the surface structure, $F = 1 + Kn \left(\frac{2}{\omega} + 1 \right)$ is the correction factor for the slip flow with respect to the Knudsen number Kn and the tangential momentum accommodation factor $\omega = 0.87$,⁵ and ρ_v is the vapor density corresponding to the droplet-substrate interfacial temperature.

The evaporation flux (J) can be calculated by Eq. S7 of the Supporting Information. Due to the absence of freezing, J is evaluated by the temperature of supercooled water at the droplet-substrate interface rather than the freezing temperature. The droplet-substrate interfacial temperature can be approximated by the average of the substrate temperature and the droplet's free surface temperature. The latter was obtained from the numerical simulations (Figure R13), which demonstrated a free surface temperature of ≈ 253.4 K at $t = 30$ s, with minimal dependence on the droplet radius (0.6-3 mm). We note that $t = 30$ s is adequately long for the environmental pressure to stabilize and for the droplet to reach a relatively stable temperature, thus the simulated

droplet surface temperature could be a reasonable estimate of the temperature at the moment of droplet levitation.

The adhesion force F_a can be calculated by considering the contact angle hysteresis of the surface:^{6,22,23}

$$F_a = 2\pi\sigma R_b(1 + \cos \theta_r), \quad (R8)$$

where σ is the surface tension of the water, R_b is the basal radius of the droplet at the moment of levitation, θ_r is the receding contact angle that is assumed to be close to the apparent receding contact angle θ_r^{app} .

The gravitational force of the droplet F_g is obtained by:

$$F_g = \rho V_d g, \quad (R9)$$

where ρ is the density of water, g is the gravitational acceleration, V_d is the droplet volume at the levitation moment and is a function of the droplet basal radius R_b . For Bond numbers of droplets close to 1, the droplet can no longer hold a spherical cap shape. In this case, the volume of the droplet as a function of the basal radius R_b can be derived by using a polynomial to interpolate the volume-radius relationship given by Ref.24:

$$V_d [\mu L] = 16.164R_b^2 + 4.6477R_b + 4.7753 [R_b \text{ in mm}], \quad (R10)$$

which is interpolated for a droplet with a contact angle of 162° . A critical droplet volume V_d is then solved by Eq. R11 below and converted to a critical droplet radius $R_{d,u}$ (of a sphere with an equivalent volume of V_d). The calculation is consistent with our experiments (Figures R24a and R24e).

$$F_{op} > F_a + F_g. \quad (R11)$$

Although a large and stationary droplet necessitates a shorter timescale to incubate ice nucleation (due to a large surface area), it has a higher volume-averaged temperature as we have demonstrated in **Figure R14** via numerical simulations (COMSOL 6.0 Heat Transfer in Fluid and Laminar Flow Physics under an axisymmetric 2D geometry). Here, the average temperature of a

droplet was defined by the average of the maximum and minimum temperatures within the droplet. Furthermore, the temperature profile within the droplet can be significantly flattened due to the remarkable fluctuations of the Leidenfrost droplet that enhances fluid convection. Therefore, freezing is delayed by the Leidenfrost effect.

Once the Leidenfrost effect is initiated via overpressure, a vapor layer is formed to provide shear pressure that elevates the droplet.¹¹ The Leidenfrost phenomenon is characterized by a vapor film underneath the droplet that elevates the droplet from the substrate, which shares similarities to the overpressure effects that drive droplet trampolining.⁵ However, the difference between overpressure and the shear stress driving the Leidenfrost effect is that the Leidenfrost droplet detaches locally and slightly from the surfaces and has reduced dependence on surface structures, while the overpressure theory considers the situation when the droplet contacts closely with the structured surface and the limited drainage of the vapor through the structures strengthens the local pressure. Furthermore, the thickness of the vapor layer for a Leidenfrost droplet is highly correlated to the weight of the droplet, and thus the local pressure decreases rapidly with the vapor layer thickness.¹¹

Figure R14. Volume-averaged droplet temperature T_{avg} as a function of droplet size R_d ranging from 0.6–3 mm for a water droplet exposed to a low-pressure environment. Simulations were performed via COMSOL 6.0 (Heat Transfer in Fluid and Laminar Flow Physics under an axisymmetric 2D geometry). The transient depressurizing pressure as a function of time (Figure S9, Supporting Information), a constant substrate temperature of 298.15 K, a fixed droplet shape ($\theta=160^\circ$), and a no-slip boundary condition on the droplet’s free surface for simplicity were applied. The simulation was carried out in a time-dependent model, where the time span of the simulation was 30 s, adequately long for the stabilization of the environmental pressure. The average temperature was extracted from the droplet at 30 s.

3) Effects of substrate adhesion on droplet dynamics.

As the reviewer has pointed out, the substrate adhesion can resist droplet jumping from the surfaces. In our experiments, the superhydrophobic surfaces used for experiments have excellent water repellency ($\theta_a^{\text{app}} \approx 160^\circ$, $\theta_r^{\text{app}} > 150^\circ$), thus allowing us to quantify the vaporization momentum without involving substrate adhesion.

To broaden the range of substrate wettability, we have performed additional experiments using a re-entrant Si surface with a lower level of hydrophobicity.⁹ The re-entrant surface is selected because it has a high apparent advancing contact angle ($\theta_a^{\text{app}} \approx 160^\circ$) with a low apparent receding contact angle ($\theta_r^{\text{app}} \approx 144^\circ$). As shown in **Figure R15**, similar freezing, deformation, and jumping dynamics were observed on the re-entrant surface, though the captured pinning and stretching of the liquid filament at the detachment moment suggested local adhesion to the surface structures. For an identical initial droplet size ($R_d = 2.06$ mm), the measured jumping velocity ($v_j = 0.39$ m/s) was slightly lower (<7%) than that of a freezing droplet on superhydrophobic surfaces ($v_j = 0.42$ m/s, obtained from the scaling relationship).

Figure R15. High-speed optical imaging of droplet freezing and self-jumping on the re-entrant surface (see Figure R1 and Table R1 for the detailed characterization of sample Re). The initiation of ice nucleation was marked by the yellow diamond, the vaporization repulsion force (F_t) acting on the ice shell was indicated by the yellow arrow, and the jumping velocity (v_j) was marked by the red arrow. The pinning and stretching of the liquid filament at the detachment moment is marked in the dotted frame. Scale bar: 1 mm.

To understand the effect of surface adhesion on droplet jumping, we quantified the work of adhesion for the surfaces by: $W_a = \sigma(1 + \cos \theta)A_b$, where A_b is the basal area of the droplet and

θ is the Young's contact angle of the surface. By taking $A_b = \pi R_b^2$ without considering the solid fraction of the surface and θ to be the receding contact angle θ_r^{app} , one can estimate the maximum work of adhesion. Here, $R_b = R_d \sin \theta_a^{\text{app}}$ is the radius of the droplet's basal area. Compared with the kinetic energy of the detaching droplet, i.e., $E_k = 0.5\rho V_d v_j^2$, where ρ is the liquid density and V_d is the volume of the jumping droplet, the work of adhesion for the surfaces and the range of droplet sizes in our experiments is relatively small ($W_a/E_k < 5\%$). Thus, we did not include the work of adhesion in the scaling analysis.

The local high-energy spots or defects (“sticker”) could introduce pinning forces to the droplet, and thus can alter the jumping direction through the pivot effects.⁴ To demonstrate the **effects of pinning (“sticker”)**, we introduced an artificial defect (with a length of $\sim 500 \mu\text{m}$ and a width of $\sim 50 \mu\text{m}$) on the superhydrophobic substrate using a sharp knife tip (with a width of $\sim 50 \mu\text{m}$), deposited a droplet to the defect, and performed low-pressure freezing experiments. It was shown in **Figure R16** that the freezing droplet detached obliquely from the surface, and the jumping direction was not correlated to the nucleation site angular position (e.g., $\theta_n \approx 56^\circ$ and $\theta_j \approx 53^\circ$). This is similar to the case of coalescence induced droplet jumping on superhydrophobic surfaces with high-surface-energy pinning sites,⁶ where the local pinning causes a deviation in the jumping direction, making it possible to shift from an out-of-plane trajectory to an in-plane one. Furthermore, the jumping velocity of the pinned droplet was reduced to 0.14 m/s, in contrast to 0.24 m/s calculated from the velocity scaling for an identically sized droplet on a pinning-free surface. In our current study, the surfaces used for the experiments have low-adhesion and uniform wettability/structures with extremely low adhesion, and thus the effect of local pinning has been eliminated.

To clarify these important points, we have **added a discussion** to page 18 of the revised manuscript. Correspondingly, **a new section** (Section S12) has been added to the revised Supporting Information: *Furthermore, while our current study demonstrates that the jumping dynamics are primarily governed by freezing dynamics and show minimal dependence on the tested surface structures, surfaces with heterogeneous adhesion and structures may interfere with droplet-substrate interaction and impact the jumping dynamics (see Section S12, Supporting Information).*

Figure R16. High-speed optical imaging of droplet freezing and self-jumping on a superhydrophobic surface with a local pinning site. High-speed optical imaging of droplet freezing and self-jumping on the micro-pillared surface (sample S1, see Figure 3c of the manuscript). The initiation of ice nucleation was marked by the yellow diamond, the vaporization repulsion force (F_t) acting on the ice shell was indicated by the yellow arrow, the ice front was marked by the yellow curve, and the jumping velocity (v_j) was marked by the red arrow. The pinning, stretching, and breakup of the liquid filament during droplet detachment are marked in the blue frame. A residual liquid filament is highlighted by the blue frame after the droplet detachment.

5) Other work has investigated the role of the inherently asymmetric recalescence freezing process—nucleation only happens at one point and freezing progresses towards supercooled water—itsself on momentum transfer without a substrate (DOI: 10.1103/PhysRevFluids.8.L021601). I would recommend discussing this work in the context of the present manuscript.

Response: We thank the reviewer for bringing forth the inspiring work, which reported the recalescence-mediated propulsion of airborne microscale droplets in the absence of a substrate. The gained momentum by the airborne droplet results directly from the progressive recalescence, sharing similarity with the self-jumping of freezing droplets on superhydrophobic substrates. The latter is complicated by the droplet-substrate interaction, with the momentum transfer mediated by the droplet deformation and substrate counteraction. The reported self-propulsion of airborne droplets together with the self-jumping of sessile droplets highlight the generality and importance of the vaporization momentum of freezing droplets at reduced pressures.

To address this point, we have added a discussion to pages 3-4 and 14 of the revised manuscript with references to the aforementioned literature³ (*Phys Rev Fluids* 8, L021601, 2023) as well as a recent related work¹ (*Nature* 620, 557–561, 2023). The reference to these papers is in bold.

*The unexplored driving force other than overpressure for droplet self-launching has prompted researchers to **examine the role of vaporization during progressive recalescence**.^{1,25} Recalescence has been demonstrated to elevate the surface temperature, introduce intensive vaporization, and exert forces on the droplet.² Recent endeavor has identified the recalescence-induced reaction force, **referred to as the propulsion force³ or recalescence force,¹** which acts on the freezing droplets and plays a crucial role in shaping their behaviors. This is exemplified by the remarkable Cassie-Baxter-to-Wenzel wetting transition of freezing droplets on textured superhydrophobic surfaces,¹ as well as **the intriguing acceleration exhibited by airborne droplets.**^{3,41,25211}*

*In contrast to the previous work on self-launching,⁵ wetting transition,¹ and **self-propulsion of freezing droplets,**^{3,4} the focus of this study is to unveil the fundamental role played by vaporization momentum in shaping the self-jumping dynamics, considering their intricate interplay with droplet size, freezing dynamics, droplet deformation, and substrate-droplet interactions.*

*Our study provides insights into the phase change of supercooled water droplets, which will facilitate the design of environment-controlled water-surface interaction systems for advanced cooling, passive anti-icing,¹ as well as **cloud freezing and climate physics.**³*

6) I had a bit of trouble following the work as the abstract states that the present finding was: “This momentum is attributed to the counteraction of recalescence-intensified interfacial vaporization over the droplet's free surface, which is distinct from the previously identified overpressure underneath the droplet.” While in the introduction they write, “While the recalescence force is a significant factor, the mechanism by which the recalescence force gives rise to the droplet's out-of-plane jumping remains elusive.” To me this implies either that the novelty is focusing on either interfacial vaporization without a substrate nearby (overpressure is not important) or overpressure is important. I would recommend trying to write this a bit clearer for the readers.

Response: We thank the reviewer for the helpful suggestion, which helped us to clarify our novelty, namely the vaporization-momentum-governed self-jumping dynamics. We are sorry for the confusing sentences. We would like to highlight that our novelty is the vaporization-momentum-governed self-jumping dynamics, rather than the overpressure. To clarify the confusion, we have substantially revised the abstract and introduction to further concentrate on our findings, and the confusing sentences mentioned by the reviewer have been re-written:

*Here, we investigate the freezing and **jumping dynamics** of supercooled water droplets on superhydrophobic surfaces, induced by the remarkable vaporization momentum, in a low-pressure environment (e.g., ~100 Pa). The vaporization momentum arises from the droplet's free surface vaporization, progressed and intensified by recalescence, subsequently induces droplet compression, collision, and finally self-jumping. By incorporating the physics of liquid-gas-solid phase changes involving vaporization, freezing recalescence, and liquid-solid interactions, we resolve the vaporization momentum and droplet dynamics, revealing a size-scaled jumping velocity and a nucleation-governed jumping direction with minimal reliance on tested surface structures, **in contrast to the overpressure directed droplet jumping.***

*The unexplored driving force other than overpressure for droplet self-launching has prompted researchers to examine the role of vaporization during progressive recalescence.^{1,25} Recalescence has been demonstrated to elevate the surface temperature, introduce intensive vaporization, and exert forces on the droplet.² Recent endeavor has identified the recalescence-induced reaction force, referred to as the propulsion force³ or recalescence force,¹ which acts on the freezing droplets and plays a crucial role in shaping their behaviors. This is exemplified by the remarkable Cassie-Baxter-to-Wenzel wetting transition of freezing droplets on textured superhydrophobic surfaces,¹ as well as the intriguing acceleration exhibited by airborne droplets.^{3,4} **However, for a sessile droplet, the recalescence force develops in a progressive top-down manner, from the droplet's free surface to the substrate, and tends to induce droplet impalement into substrate structures.¹ This opposes the localized overpressure that typically separates the droplet from the substrate,⁵ The paradoxical outcomes (impalement or jumping) of these two mechanisms highlight the lack of a comprehensive understanding of supercooled droplet freezing and the challenges in predicting the intricate dynamics associated with droplet jumping.***

*In this study, we investigate the self-jumping dynamics of supercooled, freezing droplets on superhydrophobic surfaces at low pressures (~100 Pa). We reveal a significant counteractive momentum, termed vaporization momentum, resulting from the progressive recalescence of the droplets. We demonstrate that this vaporization momentum surpasses the overpressure generated underneath the droplet and leads to compressive deformation followed by jumping. **In contrast to the previous work on self-launching,^{1,5} wetting transition,¹ and self-propulsion^{3,4} of freezing droplets, the focus of this study is to unveil the fundamental role played by vaporization momentum in shaping the self-jumping dynamics, considering their intricate interplay with droplet size, freezing dynamics, droplet deformation, and substrate-droplet interactions. Particularly, a regime map is developed to demonstrate the findings of droplet-size defined regimes of depressurization-induced transport phenomena observed in supercooled droplets.***

7) Minor comments: Reference 34 is missing a journal name.

Response: Thanks for the reviewer's capture. We have updated the references.

Response to Reviewer # 2

Comments: The authors show that a mismatch in vaporization momentum during the recalescent (i.e., stage one) freezing of a droplet on a superhydrophobic surface can result in spontaneous jumping when the system is within a vacuum chamber pumped down to ~100 Pa. While previous works have shown the spontaneous jumping of droplets from superhydrophobic surfaces in vapor chambers, this was a separate “overpressure” mechanism where the vapor pressure was building up in the underlying surface structure due to evaporation. In contrast, here the vaporization force is top-down and is due to the recalesced fraction of the droplet being warmer (and thus evaporating faster); further, the direction of jumping depends on the direction of the recalescent freeze front.

I especially appreciated the clever control experiments to ensure that the jumping mechanism was the recalescence force, rather than overpressure. Namely, showing that the droplets dramatically deform and slide even on a liquid-impregnated surface (which removes the air pockets causing overpressure), and that the jumping kinetics did not change with varying the underlying surface structure. Showing that the data matches well with the scaling for the recalescence force ($v \sim R^{0.5}$) and poorly for the overpressure force ($v \sim R^{.25}$) is also a nice touch.

However, all this being said, I cannot recommend this report for consideration in Nature Communications. For the sole reason that another report, published only months ago in Nature Physics by another group, has already discovered, measured, and theoretically analyzed this recalescence force for droplets on a SHPB surface in a vacuum chamber (Ref. 33). Further, this existing report also reported on the possibility of two other types of wetting transitions during recalescent freezing (impalement and suffusion) depending on the surface structure, which this present manuscript does not focus on. While I do think there are some creative approaches to looking at the recalescent force in this present manuscript, especially the aforementioned control cases to rule out overpressure, scaling analysis of jumping speed, and focus on jumping directionality, these are incremental and not revolutionary given that the majority of the paper is covering similar ground to Ref. 33. Given that Ref. 33 was published only months ago, it is obvious that the research projects on discovering the recalescence force were prepared in parallel, and it is bad luck that Ref. 33 got there first. Given the fresh perspective of the present work on the recalescence force, it would still be suitable for publication but in a lower-impact journal.

Response: We appreciate the reviewer for the helpful comments and suggestions. To address the concern regarding the novelty in contrast to previous studies (Ref. 33 of our original manuscript, Ref. 32 of our revised manuscript, *Nature Physics*, 1-7, 2023) we would like to summarize the **novelty of our work** here, **strengthened by our additional work**:

1) Complementary insights into the vaporization momentum-driven self-jumping dynamics of supercooled water droplets on superhydrophobic surfaces. In previous studies, the droplet detachment was either attributed to localized overpressure built underneath the droplet (*Nature*, 527, 82-85, 2015), or to the recalescence force that typically develops progressively from the droplet's free surface to the substrate (*Nature Physics*, 1-7, 2023). However, neither the overpressure nor the recalescence force theories could well explain our observation of consistent self-jumping from various structured surfaces, as the bottom-up overpressure is highly dependent on surface structures, while the top-down recalescence force induces liquid impalement into structures and acts against droplet jumping. We revealed that the intricate droplet-substrate interactions during droplet freezing could re-direct the vaporization momentum to propel droplet self-jumping, regardless of the surface structures and the top-down recalescence forces. Our study clarifies the role of vaporization momentum and overpressure and presents new insights into droplet dynamics at reduced pressure.

2) Scaling analysis of droplet jumping dynamics. Despite their significance in advancing the comprehension of mass and energy transport under extreme conditions, the dynamics of self-jumping phenomena, i.e., jumping velocity and direction, *remain unexplored* in previous studies. We resolved the jumping dynamics and their interplay with droplet sizes and freezing processes, and developed a scaling relationship for a largely extended range of droplet sizes (ranging from 0.6 to 2.8 mm) and a significant variation of surfaces (including **11 differing surfaces** ranging from nano-, micro-, micro/nano-, macro-structured, and re-entrant surfaces, Figure R20 and our responses to Comment # 4 below). This strengthened the universality of our reported vaporization-momentum-governed transport phenomena.

3) New findings of droplet-size defined regions where the vaporization momentum governs droplet dynamics. Through additional efforts, we have developed a **regime map** (Figure R17, see our responses to Comment # 3 below) to identify the vaporization momentum-driven self-jumping of supercooled droplets in contrast to other low-pressure-induced transport phenomena of sessile

water droplets (i.e., evaporative drying of small droplets, overpressure-initiated Leidenfrost effects of large droplets). The regime map helps better identify the vaporization-momentum-induced self-jumping of freezing water droplets and offers a complete picture of droplet transport dynamics under reduced pressure.

Furthermore, we have added a detailed discussion of our findings in the context of previous studies (*Nature Physics*, 1-7, 2023; *Nature*, 527, 82-85, 2015).

Below, we provide a point-to-point response to the reviewer to address the specific concerns raised by the reviewer.

1) I don't understand how the high depressurization rate is related to better removing NCGs. To my knowledge, the amount of NCGs in the chamber is purely related to the final pressure, not the rate of depressurization. Please elaborate.

Response: We thank the reviewer for the comments. We agree with the reviewer that the total amount of NCGs in the chamber is determined by the final pressure if there is no release of vapor from an evaporating droplet during pumping down. We believe our cases are different from a regular pumping process, as the droplet keeps evaporating and releasing vapor into the chamber. As a result, during the pumping process, the amount of NCGs keeps decreasing while the droplet vaporization flux keeps increasing, and the relative content of NCGs (mass or molar fraction) keeps decreasing. Based on this understanding of the pumping down process, we qualitatively analyzed the effects of depressurization rate (pumping speed) on the final relative content (mass/molar fraction) of NCGs.

For the closed chamber with a volume of V_c and initial pressure $p_{c,0}$, the pump-down time Δt_{pump} that is needed to reach a target pressure $p_{c,1}$ within the chamber is given by:²⁶

$$\Delta t_{\text{pump}} \approx \frac{V_c}{S_p} \ln \frac{p_{c,0}}{p_{c,1}}. \quad (\text{R12})$$

where S_p is the pumping speed (depressurization rate, determined by the pump capacity subtracted by the droplet's vaporization rate). For a given vacuum system (known V_c , $p_{c,0}$, and a relatively

slow release of vapor from the droplet compared to the pumping speed), it can be seen that a higher pumping speed S_p leads to a lower pressure $p_{c,1}$ after pumping for a fixed pump-down time Δt_{pump} . As the initial amount of NCGs is fixed in the chamber for a given $p_{c,0}$ and relative humidity RH_0 , a higher pumping speed would cause a larger amount of removed gasses (including NCGs) and thus a smaller amount of NCGs remaining in the chamber. Furthermore, the droplet vaporization flux increases with decreasing environmental pressure, and a lower final pressure (corresponding to a higher pumping speed for a fixed pump-down time) would result in a higher vapor release rate and subsequently a relatively low content of NCGs.

To further understand the effects of pumping speed, we consider two extreme cases. For an extremely large pumping speed, the gasses in the chamber are immediately eliminated at a time scale much shorter than the temperature-decreasing time scale (or heat transfer time scale), and then the droplet would evaporate in an environment in the absence of NCGs. For another extreme case with an extremely slow pumping rate, the vapor and NCGs in the chamber are slowly and proportionally removed from the chamber. In this case, during most of the lifespan of the evaporating droplet, the droplet would be exposed to an environment with NCGs of an almost constant content.

Therefore, a high pumping speed is preferred to reduce the relative content of NCGs to minimize the effects of NCGs on mass transfer at the moment of freezing.

To address this point, we have revised the sentence on page 5 of the manuscript: *The high depressurization rate (~ 10 kPa/s) was employed to rapidly evacuate the chamber and minimize the amount of non-condensable gasses (NCGs) during droplet freezing.*

2) The scaling analysis is interesting and should have stand-alone equations that are more clearly developed. It was hard to compare the graphs to the scaling theory without this.

Response: Thanks for the reviewer's helpful comments. We have revised the text to leave the equation stand-alone (page 11).

3) When submitted to a different journal, the authors should devote at least an entire paragraph of discussion contrasting their contributions to Ref. 33. Here, they do so very briefly in a couple of

sentences in the introduction, but it's not sufficient. Perhaps a discussion paragraph after the results have been presented would be the most fruitful way to compare and contrast the two similar works.

Response: Thanks for the reviewer's helpful comments. We appreciate the Nature Physics paper (Ref. 33 of our original manuscript, Ref. 32 of our revised manuscript, *Nature Physics*, 1-7, 2023) published recently, which has delicately elucidated the outcomes of freezing droplets on superhydrophobic surfaces, i.e., wetting transition or droplet expulsion, depending on the balance between recalescence forces and resisting forces (capillary forces or adhesion forces). **Our study aims to address a different and important aspect** of droplet freezing at reduced pressures, i.e., self-jumping dynamics (velocity and direction) with respect to droplet sizes and surface structures, which remains unexplored. **The novelty of our work is further strengthened** by our new findings of the size-defined regimes of droplet dynamics at reduced pressures, which distinguish the vaporization-momentum-governed self-jumping from evaporative drying and overpressure-initiated levitation. While previous studies (*Nature Physics*, 1-7, 2023; *Nature*, 527, 82-85, 2015) focused on the effects of surface structures in droplet dynamics (self-trampolining, wetting transition, and expulsion), **our current work focuses on the crucial role of droplet size in droplet self-jumping dynamics** (velocity and direction). We believe **our key contributions** in contrast to previous studies are not only the quantified jumping dynamics but also the fate of droplets with respect to droplet sizes, with less dependence on surface structures.

More importantly, to further strengthen our novelty in the context of previous studies (Ref. 33 of our original manuscript, Ref. 32 of our revised manuscript, *Nature Physics*, 1-7, 2023), we have significantly extended the droplet size range in our experiments and developed **a size-defined regime map** of depressurization-induced phenomena of sessile water droplets. The identified regimes include: 1) evaporative drying for small droplets, 2) vaporization-momentum-dominated self-jumping for medium-sized droplets, and 3) overpressure-initiated Leidenfrost effects for large droplets (**Figure R17a**). Notably, there exist a lower ($R_{d,l}$) and upper ($R_{d,u}$) boundaries in terms of equivalent droplet radius (R_d) that delineate the regime of droplet freezing and self-jumping. For $R_d < R_{d,l}$, the droplet dries up before freezing (**Figure R17b**). The value of $R_{d,l}$ is determined when the evaporation lifespan (t_{eva}) of a droplet with an initial R_d is shorter than the ice incubation time (t_{inc}) (**Figure R17c**). While for $R_d > R_{d,u}$, the droplet experiences a marginal and dynamic levitation rather than jumping, facilitated by a vapor layer underneath, as well as remarkable

fluctuations at the interface (**Figure R17d**), both of which indicate the characteristics of the Leidenfrost effect.^{3,4} The Leidenfrost effect significantly delays freezing due to enhanced fluid convection caused by the remarkable droplet fluctuations, counteracting the decrease in interface temperature. $R_{d,u}$ is reached when the overpressure force F_{op} overcomes the resistance forces (gravitational force F_g and adhesion force F_a), causing droplet-substrate separation (**Figure R17e**). Here, gravitational and adhesion forces are incorporated as the droplet has a significantly large size ($Bo > 1$) and a non-spherical shape with increased droplet-surface contact area. The theoretical boundaries are in good agreement with our experiments.

We believe the new findings of the size-defined regimes help better identify the vaporization-momentum-dominated self-jumping and add to the novelty of our work. We presented detailed analyses of the regime boundaries at the end of the response to this comment. We have **added a new figure (Figure 5)** with discussion to the revised manuscript and **an additional section (Section S13)** to the revised Supporting Information.

Figure R17. (a) Regime map illustrating droplet transport phenomena on superhydrophobic substrates in a low-pressure (~ 100 Pa) environment, depicting the relationship between droplet jumping velocity (v_j) and droplet size (R_d). The vaporization-momentum-governed self-jumping regime (jumping) is delimited by the evaporative drying regime (drying) for small droplets and the overpressure-initiated Leidenfrost effect regime (Leidenfrost) for large droplets. Theoretical boundaries are represented by dashed lines. Note, for the evaporative drying regime, R_d is defined as the initial droplet radius immediately before pumping down. For the self-jumping regime, R_d is measured as the droplet radius immediately before freezing. For the Leidenfrost regime, R_d is defined as the equivalent droplet radius of a spherical droplet having an identical volume immediately before droplet levitation. (b) Optical images showing the evaporative drying regime. $t=0$ s corresponds to the start of pumping down. See Video S3. (c) Theoretical determination of the lower boundary ($R_{d,l}$) by comparing the evaporation lifespan (t_{eva}) and ice incubation time (t_{inc}) as a function of the initial droplet radius (R_d). $t_{eva} < t_{inc}$ represents the droplet dries up before freezing. (d) Optical images showing the Leidenfrost effect regime. The overpressure-induced droplet-substrate separation is highlighted and zoomed in in the frame. $t_0 \approx 30$ s represents the pumping-down time before the initiation of droplet levitation. See Video S4. (e) Theoretical determination of the upper boundary ($R_{d,u}$) by comparing the overpressure force (F_{op}) and the sum of gravitational force and adhesion force ($F_g + F_a$) as a function of equivalent droplet radius (R_d). $F_{op} > F_g + F_a$ represents that the overpressure overcomes the resistance forces and induces droplet-substrate separation. Here, gravitational and adhesion forces are incorporated since the droplet has a significantly large size ($Bo > 1$) and a non-spherical shape with an increased droplet-surface contact area. Sample S1 with microstructure height of $15 \mu\text{m}$ was used in the experiments and modeling.

Lastly, we have discussed the inspiring work (Ref. 33 of our original manuscript, Ref. 32 of our revised manuscript, *Nature Physics*, 1-7, 2023) multiple times throughout our manuscript. We listed our related discussion below, and the additional discussions are underlined:

1) In the introduction (pages 3-4 in the revised manuscript):

The instant freezing subsequently leads to droplet self-launching from superhydrophobic surfaces.^{12, 33}

In fact, freezing droplets have been observed to jump off superhydrophobic surfaces that possess large-scale structures within this range.³³

The unexplored driving force other than overpressure for droplet self-launching has prompted researchers to examine the role of vaporization during progressive recalescence.^{33,34}

Recent endeavor has identified the recalescence-induced reaction force, referred to as the propulsion force or recalescence force,³³ which acts on the freezing droplets and plays a crucial role in shaping their behaviors. This is exemplified by the remarkable Cassie-Baxter-to-Wenzel wetting transition of freezing droplets on textured superhydrophobic surfaces,³³ as well as the intriguing acceleration exhibited by airborne droplets.³⁴

In contrast to the previous work on self-launching,^{12,33} wetting transition,³³ and self-propulsion³⁴ of freezing droplets, the focus of this study is to unveil the fundamental role played by vaporization momentum in shaping the self-jumping dynamics, considering their intricate interplay with droplet size, freezing dynamics, droplet deformation, and substrate-droplet interactions.

2) In the main text (pages 7-9 in the revised manuscript):

To further validate the scaling relationship, we analyzed recently reported experiments³³ and the results together with our experiments were in good agreement with the scaling regardless of the surface microstructures.

Independent experimental data from Ref.33 are included as star symbols in (d) and (e).

Furthermore, the vaporization momentum is consistent with the recalescence force identified in a recent study on freezing droplets.³³

3) In the discussion (pages 16-17 in the revised manuscript):

While previous studies^{1,5} have delicately elucidated the effects of surface structures in droplet dynamics (self-trampolining, suffusion, wetting transition, and expulsion),⁵¹ our current work focuses on droplet self-jumping from impalement-resisting superhydrophobic surfaces, and highlighted the crucial role of droplet size rather than surface structures in jumping velocity and the size-defined fate of droplets at reduced pressures (evaporative drying, self-jumping, and overpressure-initiated levitation).

Moreover, the absence of wetting transition,³³ an opposite phenomenon induced by recalescence force, suggests the robustness of self-jumping of freezing droplets on a variety of superhydrophobic surfaces. However, the criteria of wetting transition³³ need to be incorporated into the rational design of surface structures, especially single-tier structures, to regulate vaporization-mediated droplet dynamics.

Below we present detailed analyses of the size-defined regimes and their boundaries, which have been added as a new section to Supporting Information (Section 13):

*To further differentiate the overpressure and vaporization momentum, we have significantly extended the range of droplet sizes (initial R_d ranging from 0.2 to 3 mm, equivalent to a volume ranging from 0.0335 to 115 μL) in our experiments and developed a **regime map (Figure R17)** to identify differing regimes including: (1) evaporative drying for small droplet sizes, (2) vaporization-momentum dominated droplet freezing and self-jumping for medium droplet sizes, and (3) Leidenfrost effects for large droplet sizes (**Figure R17a**). Specifically, there exists a lower boundary $R_{d,l}$ and an upper boundary $R_{d,u}$ in terms of droplet radius for the droplet freezing and self-jumping regime. For $R_d < R_{d,l}$, the droplet stays on the surface and dries up due to evaporation before freezing (**Figure R17b**), while for $R_d > R_{d,u}$, the droplet experiences a marginal elevation (but not jumping) by a vapor layer underneath the droplet and remarkable fluctuations at the interface (**Figure R17c**), characteristic of the Leidenfrost effect.^{11,12} The freezing is largely delayed by the Leidenfrost effect due to the remarkable fluctuations of the droplet that enhance fluid convection and counteract the decrease of interface temperature.*

To obtain the lower boundary $R_{d,l}$ that separates the evaporative drying and self-jumping regimes, we compared the lifespan t_{eva} and the incubation time for ice nucleation t_{inc} of an evaporating

droplet during pumping down. For a given initial droplet radius, evaporative drying occurs when $t_{eva} < t_{inc}$. t_{eva} can be obtained by considering the mass conservation during evaporation. For droplet evaporation in a depinning mode with a constant contact angle θ , which is approximately the case for our superhydrophobic surfaces having a high receding contact angle, the decrease in the droplet volume results from the vaporization over the droplet surface,^{13,14} and thus:

$$\rho dV_d = -JA_d dt, \quad (R13)$$

where $V_d = \pi R_d^3 (1 - \cos \theta)^2 (2 + \cos \theta) / 3$ is the droplet volume as a sphere cap, $A_d = 2\pi R_d^2 (1 - \cos \theta)$ is the area of the droplet's free surface, and $J = \frac{2\beta}{2-\beta} \sqrt{\frac{M}{2\pi R}} \left(\frac{P_d}{\sqrt{T_d}} - \frac{P_e}{\sqrt{T_e}} \right)$ is the vaporization mass flux given by the Hertz-Knudsen-Schrage equation^{15,16} (Eq. S7, Supporting Information) for kinetic-dominated evaporation. Integrating Eq. R13 with a known initial droplet radius R_d gives:

$$\int_0^{t_{eva}} J dt = \frac{\rho R_d (1 - \cos \theta) (2 + \cos \theta)}{2}, \quad (R14)$$

Note, J is a function of the vapor pressure at the droplet surface (P_d) and vapor pressure in the environment (P_e), all of which are time-dependent. Assuming a time-averaged vaporization flux \bar{J} , the droplet lifetime can be solved:

$$t_{eva} = \frac{\rho R_d (1 - \cos \theta) (2 + \cos \theta)}{2\bar{J}}. \quad (R15)$$

To estimate \bar{J} , the transient temperature field within the droplet was simulated via COMSOL 6.0 (Heat Transfer in Fluid and Laminar Flow Physics) incorporating evaporative cooling, achieved by applying a temperature-dependent vaporization heat flux to the droplet's free surface. The modeling exploited an axisymmetric 2D geometry setup as shown in **Figure R18a**. A substrate temperature $T_{sub} = 298.15$ K in equilibrium with the environmental temperature T_e was assumed. The transient environmental pressure P_e (Figure S9, Supporting Information) was inputted as the boundary condition. Simulations were performed for an overall simulation time of 30 s with a time step of 0.1 s. This simulation time covers the time window for our depressurization experiments (Figure 1b of the manuscript). Droplets with varied radius $R_d = 0.2 - 3$ mm and a constant contact

angle $\theta=160^\circ$ were simulated. **Figure R18b** presents the simulated temperature profile within a typically sized droplet, showing $T_d \approx 254.1$ K ($P_d \approx 112$ Pa) at $t=30$ s after the initiation of evaporative cooling. Simulations for differing droplet sizes ($R_d=0.2-3$ mm) showed a modest variation (2-3 K) of the temperature at the droplet's free surface at $t=30$ s. The simulated droplet surface temperature is a reasonable estimate when compared to the experimental measurement of the temperature inside the droplet with a similar size (-12°C , Figure 1b of the manuscript) and the derived temperature of the free surface at the freezing moment (-27°C , Eqns. S1-S3, Supporting Information). \bar{J} is then estimated by $\bar{J} = \frac{1}{\Delta t} \int_0^{\Delta t} J(T_d, P_d, T_e, P_e) dt$ with T_d (and thus P_d) simulated as a function of time t (**Figure R18c**) and Δt approximated by 30 s. Note that an evaporation coefficient^{17,18} $\beta \approx 0.759$ is adopted when evaluating J (see Section S3, Supporting Information).

It should be noted that the simulations did not incorporate changes in droplet size during evaporation and assumed negligible effects of droplet mass depletion on the droplet temperature profile. This assumption is based on the fact that the droplet size does not change significantly during the rapid pumping-down process. The simulations were only used to estimate the droplet surface temperature for the analytical modeling (Eq. R15) and were not performed to obtain the lifetime of evaporating droplets, which is feasible but numerically challenging to include the phase change (mass depletion) at the interface and internal circulation within the droplet. Instead of dealing with the numerical convergence issues, we used the simulations to estimate the surface temperature and to develop a semi-analytical model of the lifetime of evaporating droplets.

Figure R18. (a) COMSOL multiphysics model setup of the droplet evaporation, where the free surface of the droplet is subjected to the evaporation cooling flux $J(T_d)$ as a function of surface temperature T_d . The transient environmental pressure P_e (Figure S9, Supporting Information) was inputted as the boundary

condition. Simulation was performed with a contact angle $\theta=160^\circ$ and a time step of 0.1 s. (b) Simulated temperature distribution at time $t=30$ s for a droplet having a radius $R_d=1.5$ mm. Evaporative cooling starts at $t=0$ s. (c) The averaged droplet surface temperature $T_{d,ave}$ as a function of time t . Minimal differences in $T_{d,ave}$ were observed for R_d ranging from 0.2 to 3 mm.

The ice nucleation incubation time t_{inc} corresponds to the moment when the very first nucleus is formed in the droplet. The total number of nuclei is a function of the droplet volume and the ice nucleation rate. The latter is dependent on the temperature field inside the droplet. Here, the transient temperature field of droplets with different sizes was obtained from the numerical simulation performed above (Figure R18). With a known time-dependent temperature field, the local nucleation rate per unit volume is given by the classical nucleation theory:¹⁹

$$J_n = \frac{A_m}{3} n^{7/3} \sqrt{\frac{A_m \sigma}{\pi k_B T}} D_w \exp \left\{ - \frac{4 \left(\frac{A_m \sigma}{k_B T} \right)^3}{27 \left[\ln \left(\frac{P_w}{P_i} \right) \right]^2} \right\}, \quad (R16)$$

where $A_m = 4\pi r_m^2$ is the surface area of a water molecule with a radius of r_m , $n = \frac{\rho N}{M}$ is the number density of the water, ρ is water density, Avogadro constant N , and molar mass M , $D_w = 1.5 \times 10^{-6} \exp \left(- \frac{2045.0}{T} \right)$ is the self-diffusivity of supercooled water, $\sigma = 0.0291 + 10^{-4}(T - 273.15)$ is the water-ice interfacial tension, P_w is the vapor pressure of supercooled water at temperature T , and P_i is the vapor pressure of ice at an equilibrium freezing temperature of 273.15 K. The condition of ice nucleation is defined when the total number of ice nuclei exceeds unity, given by integrating the local ice nucleation rate J_n over the droplet volume during a period of t_{inc} :¹⁹⁻²¹

$$\int_0^{t_{inc}} \int_{V_d} J_n dV_d dt = 1, \quad (R17)$$

where $V_{d,t}$ is the droplet volume at time t and can be solved by Eq. R13. For a given initial droplet volume V_d , t_{inc} can be found by numerically solving the integration problem above with the input of the time-dependent droplet temperature profile. Given that the droplet temperature is lower than the substrate (Figure R18), we only considered the homogeneous nucleation and neglected

the heterogeneous nucleation at the droplet-substrate interface. Figure R17c shows t_{inc} as a function of the initial droplet radius R_d , revealing that a smaller droplet has a longer incubation time for ice nucleation. This is because a smaller surface area or volume leads to a reduced number of ice nuclei. Moreover, the small droplet has a lower temperature that is closer to the substrate temperature due to its lower thermal conduction resistance.

By comparing the ice incubation time and the lifetime of the evaporating droplet, a critical droplet size $R_{d,l}$, below which ice nucleation hardly occurs, can be solved by:

$$t_{eva} < t_{inc}. \quad (R18)$$

The calculated $R_{d,l}$ is shown to be consistent with the experiments (Figures R24a and R24c).

To obtain the upper droplet size boundary $R_{d,u}$ that separates the Leidenfrost effect and the self-jumping regimes, we consider the criteria for the initiation of the Leidenfrost effect, which starts when the overpressure force F_{op} underneath the droplet overcomes the gravity and adhesion of the droplet. Here, gravitational force (F_g) and adhesion force (F_a) are incorporated as the droplet has a significantly large size ($Bo > 1$) and a non-spherical shape with increased droplet-surface contact area. The overpressure force F_{op} , defined consistently with that in our analyses of overpressure (Figure 3e in the manuscript and Eqns. S26-S29 in the Supporting Information), can be modeled by:⁵

$$F_{op} \approx \frac{3\mu\pi R_b^4 J}{h^3 F \rho_v}, \quad (R19)$$

where μ is the dynamic viscosity of vapor, R_b is the basal radius of the droplet, J is the evaporation flux of the droplet, h is the height of the surface structure, $F = 1 + Kn \left(\frac{2}{\omega} + 1 \right)$ is the correction factor for the slip flow with respect to the Knudsen number Kn and the tangential momentum accommodation factor $\omega = 0.87$,⁵ and ρ_v is the vapor density corresponding to the droplet-substrate interfacial temperature.

The evaporation flux (J) can be calculated by Eq. S7 of the Supporting Information. Due to the absence of freezing, J is evaluated by the temperature of supercooled water at the droplet-substrate interface rather than the freezing temperature. The droplet-substrate interfacial

temperature can be approximated by the average of the substrate temperature and the droplet's free surface temperature. The latter was obtained from the numerical simulations (Figure R18), which demonstrated a free surface temperature of ≈ 253.4 K at $t=30$ s, with minimal dependence on the droplet radius (0.6-3 mm). We note that $t=30$ s is adequately long for the environmental pressure to stabilize and for the droplet to reach a relatively stable temperature, thus the simulated droplet surface temperature could be a reasonable estimate of the temperature at the moment of droplet levitation.

The adhesion force F_a can be calculated by considering the contact angle hysteresis of the surface:^{6,22,23}

$$F_a = 2\pi\sigma R_b(1 + \cos \theta_r), \quad (R20)$$

where σ is the surface tension of the water, R_b is the basal radius of the droplet at the moment of levitation, θ_r is the receding contact angle that is assumed to be close to the apparent receding contact angle θ_r^{app} .

The gravitational force of the droplet F_g is obtained by:

$$F_g = \rho V_d g, \quad (R21)$$

where ρ is the density of water, g is the gravitational acceleration, V_d is the droplet volume at the levitation moment and is a function of the droplet basal radius R_b . For Bond numbers of droplets close to 1, the droplet can no longer hold a spherical cap shape. In this case, the volume of the droplet as a function of the basal radius R_b can be derived by using a polynomial to interpolate the volume-radius relationship given by Ref.24:

$$V_d [\mu L] = 16.164R_b^2 + 4.6477R_b + 4.7753 [R_b \text{ in mm}], \quad (R22)$$

which is interpolated for a droplet with a contact angle of 162° . A critical droplet volume V_d is then solved by Eq. R23 below and converted to a critical droplet radius $R_{d,c}$ (of a sphere with an equivalent volume of V_d). The calculation is consistent with our experiments (Figures R24a and R24e).

$$F_{op} > F_a + F_g. \quad (R23)$$

Although a large and stationary droplet necessitates a shorter timescale to incubate ice nucleation (due to a large surface area), it has a higher volume-averaged temperature as we have demonstrated in **Figure R19** via numerical simulations (COMSOL 6.0 Heat Transfer in Fluid and Laminar Flow Physics under an axisymmetric 2D geometry). Furthermore, the temperature profile within the droplet can be significantly flattened due to the remarkable fluctuations of the Leidenfrost droplet that enhances fluid convection. Therefore, freezing is delayed by the Leidenfrost effect.

Once the Leidenfrost effect is initiated via overpressure, a vapor layer is formed to provide shear pressure that elevates the droplet.¹¹ The Leidenfrost phenomenon is characterized by a vapor film underneath the droplet that elevates the droplet from the substrate, which shares similarities to the overpressure effects that drive droplet trampolining.⁵ However, the difference between overpressure and the shear stress driving the Leidenfrost effect is that the Leidenfrost droplet detaches locally and slightly from the surfaces and has reduced dependence on surface structures, while the overpressure theory considers the situation when the droplet contacts closely with the structured surface and the limited drainage of the vapor through the structures strengthens the local pressure. Furthermore, the thickness of the vapor layer for a Leidenfrost droplet is highly correlated to the weight of the droplet, and thus the local pressure decreases rapidly with the vapor layer thickness.¹¹

Figure R19. Volume-averaged droplet temperature T_{avg} as a function of droplet size R_d ranging from 0.6–3 mm for a water droplet exposed to a low-pressure environment. Simulations were performed via COMSOL 6.0 (Heat Transfer in Fluid and Laminar Flow Physics under an axisymmetric 2D geometry). The transient depressurizing pressure as a function of time (Figure S9, Supporting Information), a constant substrate temperature of 298.15 K, a fixed droplet shape ($\theta=160^\circ$), and a no-slip boundary condition on the droplet’s free surface for simplicity were applied. The simulation was carried out in a time-dependent model, where the time span of the simulation was 30 s, adequately long for the stabilization of the environmental pressure. The average temperature was extracted from the droplet at 30 s.

4) The authors vary the surface structure to show it has no effect on the jumping speed, but according to Ref. 33, aren’t there certain choices of surface structure that would result in impalement upon recalescence rather than jumping? Did the authors ever see this?

Response: Thanks for the reviewer’s comments. We have shown the jumping velocity-size scaling with insignificant dependence on surface structures, and the wetting transition was not observed for all the surfaces tested in our experiments. Rather, robust self-jumping was consistently demonstrated by our **additional experiments** performed using a greater variety of surfaces having differing surface structures. The length scales of surface structures span a wider range from nanoscale (~ 100 nm) and microscale (~ 10 μm to ~ 100 μm) to macroscale (\sim mm). The morphology of the structures varies from randomly distributed nanoblades, micropillars, microgrooves, re-entrant microgrids, microhills with nanowires, microlattice, to macrogrooves (**Figure R20** and **Table R2**). All surfaces have nanostructures except the re-entrant surfaces. These surfaces represent the commonly used superhydrophobic surfaces with robust water repellency against liquid impalement, condensation, frosting, and ice accretion.^{5,27–30} The robust superhydrophobicity of our surfaces eliminates the potential interference by surface adhesion and droplet impalement, thus allowing us to focus on the droplet jumping dynamics and to develop a general framework of vaporization momentum-dominated droplet dynamics.

The velocity scaling of droplet jumping ($v_j \sim R_d^{0.5}$) was confirmed for all the tested surfaces, even on the re-entrant surfaces with apparent superhydrophobicity and intrinsic hydrophilicity (**Figure R21**). The absence of wetting transition in our experiments can be attributed to the high capillary pressure p_{cap} that is dependent on the surface structures and intrinsic wettability:¹

$$p_{cap} = -\frac{2\varphi_n}{1-\varphi_n} \frac{2\sigma}{s_n} \cos\theta, \quad (\text{R24})$$

where φ_n is the solid fraction, θ is the intrinsic contact angle of the surface, σ is the surface tension of water, and s_n is the spacing between the surface structures. For hydrophobic nanostructured Al surfaces functionalized with a self-assembled monolayer of silane (FAS-17), p_{cap} is calculated to be ≈ 113.5 kPa when assuming $\varphi_n=0.108$ (Eq. S4, see Section S2, Supporting Information) and $s_n=100$ nm.³¹ While for microstructures with $\varphi_n=0.072$ and $s_n=33$ μm , as is with the case of the surfaces with the densest structures in Ref. ¹, p_{cap} is calculated to be ≈ 0.23 kPa, $>450\text{X}$ lower than that for nanostructures. Therefore, the presence of nanostructures significantly enhances the resistance to wetting transition, as was also confirmed by Ref. 33 (of our original manuscript, now Ref. 32 in our revised manuscript) where the addition of nanostructures prevents liquid impalement.¹

To clarify these important points, we have added a discussion to page 11 of the revised manuscript: *These surfaces represent the commonly used superhydrophobic surfaces with robust water repellency against liquid impalement, condensation, frosting, and ice accretion.*^{6,7,9,32,33} *The robust superhydrophobicity of our surfaces eliminates the potential interference by surface adhesion and droplet impalement,¹ thus allowing us to focus on the droplet jumping dynamics and to develop a general framework of vaporization momentum-dominated droplet dynamics.*

In addition, we have added a discussion to page 17 of the revised manuscript to further **highlight the implications of the previously identified wetting transition** (Ref.1), which provides important insights into the rational design of surfaces to regulate self-jumping dynamics of the freezing droplets: *The absence of wetting transition,¹ an opposite phenomenon induced by recalescence force, suggests the robustness of self-jumping of freezing droplets on a variety of superhydrophobic surfaces. However, the criteria of wetting transition¹ need to be incorporated into the rational design of surface structures, especially single-tier structures, to regulate vaporization-mediated droplet dynamics.*

Figure R20. SEM of surfaces with differing surface structures used for additional droplet freezing and jumping experiments. (a) Nanostructured Al surface, labeled surface Na. The nanostructures have a characteristic size of 100-200 nm. (b) Macro-grooved Al surface, labeled surface Ma. The triangle grooves have a pitch of 1 mm and a depth of 1 mm. (c) Micro-latticed Al surface, labeled La. The spacing and depth of the square lattices are $700 \mu\text{m}$ and $\approx 100 \mu\text{m}$, respectively. (d) Hierarchical CuO nanowire surface, labeled Hi, consisting of microhills covered by CuO nanowires. The pitch and height of the microhills are $40 \mu\text{m}$ and $\approx 60 \mu\text{m}$, respectively. The CuO nanowires have a diameter of $\approx 100 \text{nm}$. (e) Re-entrant Si

surface, labeled Re. The side-to-side width and height of the hexagon lattices are $200 \mu\text{m}$ and $\approx 23 \mu\text{m}$, respectively. The wall of the lattice has a T-shape with a width of $\approx 5 \mu\text{m}$ at the top and $\approx 3 \mu\text{m}$ at the bottom. See Table R1 for the contact angles of the surfaces. See our previous studies for more details on the fabrication and characterization of the superhydrophobic Al surfaces,⁷ CuO nanowire surfaces,^{6,8} and re-entrant Si surfaces.⁹

Table R2. Apparent advancing contact angle (θ_a^{app}) and apparent receding contact angle (θ_r^{app}) of the superhydrophobic surfaces. Samples S1-S3 are characterized in the manuscript.

Surfaces	θ_a^{app} ($^\circ$)	θ_r^{app} ($^\circ$)
Nanostructured Al (Na)	163 ± 1.1	160 ± 1.5
Macrogrooved Al (Ma)	158 ± 1.8	155 ± 2.2
Microlatticed Al (La)	161 ± 1.6	159 ± 1.8
Hierarchical CuO (Hi)	163 ± 1.5	158 ± 2.4
Re-entrant Si (Re)	161 ± 2.0	144 ± 4.5

Figure R21. Droplet jumping velocity v_j as a function of droplet spherical radius R_d (measured immediately before freezing) for substrates with differing structures (labeled samples S1, S2, S3, Re, Ma, Na, La, Hi). See Figures S3 and S4 in Supporting Information and Figure R20 and Table R2 for the detailed characterization of the substrate surfaces. The solid line and gray band represent the prediction by the vaporization momentum ($v_j \sim R_d^{0.5}$) and overpressure theory ($v_j \sim R_d^{2.5}$), respectively. The predicted velocity by the overpressure theory reflects the variation of microstructures of samples S1-S3. See Section S10, Supporting Information for detailed modeling of overpressure. Independent experimental data from literature (Lambley, H. *et al. Nat. Phys.*, 2023)¹ which used different structured surfaces (samples C2, D4, D6, star symbols) also follow the same scaling law.

Response to Reviewer # 3

Comments: The authors reproduced the previously observed self-detaching of freezing sessile droplets from superhydrophobic surfaces in this work. Importantly, the authors proposed a much more reliable “vaporization momentum” mechanism than the previous “overpressure” theory behind the droplet self-detaching phenomenon. In contrast to the fitting error by the previous “overpressure” theory, the new “vaporization momentum” mechanism can quantitatively and precisely predict the self-jumping velocity of the freezing droplets. Particularly, the analysis of droplet self-jumping direction further enhanced the significance of the “vaporization momentum” mechanism, which was missing in the previous studies.

There is however an important point which needs further discussion from the authors in order to help the readers understand the difference between the previous “overpressure” theory and the new “vaporization momentum” mechanism. The “overpressure” theory was based on the analysis of trampolining of droplets (not freezing ones) under low pressure (ref. 12 in the main text), and further used to explain the self-jumping of freezing droplets on superhydrophobic surfaces. As such, it is possible that “overpressure” could dominating self-jumping of non-freezing droplets, while “vaporization momentum” play the determining role in self-jumping of freezing droplets. The authors are highly recommended to provide extended comments on this regard before the publication of the work.

Overall, the results of presented in the manuscript are clear and convincing. The manuscript is well structured with good readability. I would suggest minor revision to further address the differences between “vaporization momentum” and “overpressure” before accepting the current manuscript.

Response: We thank the reviewer for bringing forth this important point. We would like to address this important point by further analyzing the effect of the surface structures and droplet sizes on the droplet dynamics to differentiate the vaporization momentum and overpressure contributions on the. We agree with the reviewer that the overpressure mechanism could dominate droplet dynamics in the non-freezing cases, as demonstrated by the self-trampolining of droplets exposed to reduced pressure.⁵ The vaporization flux of supercooled water is significantly (~10X) lower compared to that of the freezing surface (Figure 3b of the manuscript). By contrast, in the presence of freezing and recalescence, the overpressure may be dominated by the vaporization momentum,

as demonstrated by our experiments. However, the overpressure force/momentum can be enhanced by surface structures (preferably with shorter features¹⁷) and droplet-substrate contact area (or droplet size) and plays a more important role in some conditions.

An important fact with respect to the overpressure force/momentum worth noting is the droplet size effect. A larger droplet radius is preferred to achieve a higher overpressure force.¹⁷ This is reflected in our analysis of the jumping velocity driven by overpressure, through which we show that for various surfaces tested (typical superhydrophobic hierarchical surfaces, S1-S3), the jumping velocity solely derived from the overpressure increases with droplet radius within the range of 0.6-2 mm ($v_j \sim R_d^{2.5}$), though it is lower than that resulted from vaporization momentum (Figure 3e of the manuscript). For a larger droplet size (e.g., $R_d > 2.5$ mm for the superhydrophobic surfaces S1-S3, Figure 3e), which is beyond the range of droplet sizes ($R_d < 1.8$ mm) tested in our previous experiments, the jumping velocity by overpressure is likely to exceed that from the vaporization-induced jumping. We note that when evaluating the overpressure in comparison with the vaporization momentum, the vaporization flux at the freezing surface was assumed for both cases (Eq. S24, Supporting Information).

Given the dependence of overpressure force on droplet sizes, we broadened the droplet size range in our experiments to further differentiate the droplet dynamics governed by vaporization momentum from those governed by overpressure. The maximum droplet volume was extended from $< 25 \mu\text{L}$ ($R_d < 1.8$ mm) to $\approx 80 \mu\text{L}$ (with an equivalent radius $R_d \approx 2.6$ mm assuming a sphere shape). It was shown within the extended droplet radius range, self-jumping of freezing droplet still happens and the jumping velocity consistently follows the scaling $v_j \sim R_d^{0.5}$, almost independent of the substrates (see **Figure R22** for the scaling relationship *in an extended range of droplet size*) with differing structures ranging from nano to micro and macro-structures (*11 differing surfaces* in total, including 8 surfaces in our test and 3 from literature, see **Figure R23** and **Table R3** for the *additional surfaces* tested). The confirmed **size dependence and structural independence** of droplet jumping velocity further highlight that the jumping dynamics are dominated by the vaporization momentum.

More interestingly, when further increasing the initial droplet volume to $\sim 100 \mu\text{L}$, a different phenomenon was observed, termed here **overpressure-initiated Leidenfrost effect**, where the droplet is levitated slightly and dynamically from the surface in the absence of freezing (**Figure**

R24d). We observed dramatic interfacial fluctuations and a layer of vapor cushion between the droplet base and the substrate, characteristics of Leidenfrost effects.^{3,4} It is worth noting that although the droplet can locally detach from the surface, it gains negligible net jumping velocity ($v_j \approx 0$) due to the gravitational forces. Once the vapor cushion layer is formed, the shear pressure in the thin vapor layer acts to balance the gravity, and the thickness changes dynamically to adapt to the gravity of the droplet. The elevation and fluctuation of the Leidenfrost droplet significantly delay the droplet freezing due to enhanced mixing of the fluid, counteracting the decrease of the temperature at the droplet's free surface. Therefore, overpressure dominates the regime for large droplets, leading to the initiation of droplet levitation and delaying freezing (no freezing was observed during our visualization for >50 s). This overpressure-initiated Leidenfrost effect is defined by a critical droplet size $R_{d,u}$ that can be determined by balancing the overpressure force and resisting forces (gravitational force and adhesion force, **Figures R24a and R24e**). To provide a more complete picture, we also identified another regime for small droplets (initial radii $R_{d,0} < 0.6$ mm), where neither the vaporization momentum nor the overpressure dominates (**Figure R24b**). Rather, the evaporative drying limits the lifespan of the droplets, during which freezing is not preferred and overpressure is insignificant. Correspondingly, a critical droplet radius $R_{d,l}$ exists to divide the regimes of evaporative drying and self-jumping, which can be obtained by comparing the ice nucleation incubation time with the lifespan of the droplet (**Figure R24c**). We have developed a **regime map** with respect to droplet sizes, obtained the boundaries defining the regimes, and showed consistency between experiments and theories (**Figure R24**), as demonstrated in detail below. As an echo to the reviewer's comments, we demonstrate that there exists a regime (overpressure-initiated Leidenfrost effect) for large droplets (>100 μL) where overpressure dominates the droplet dynamics.

Figure R22. Droplet jumping velocity v_j as a function of droplet radius R_d (measured immediately before freezing) for substrates with differing structures (labeled samples S1, S2, S3, Re, Ma, Na, La, Hi). See Figures S3 and S4 in Supporting Information and Figure R23 and Table R3 for the detailed characterization of the substrate surfaces. The solid line and gray band represent the prediction by the vaporization momentum ($v_j \sim R_d^{0.5}$) and overpressure theory ($v_j \sim R_d^{2.5}$), respectively. The predicted velocity by the overpressure theory reflects the variation of microstructures of samples S1-S3. See Section S10, Supporting Information for detailed modeling of overpressure. Independent experimental data from literature (Lambley, *H. et al. Nat. Phys.*, 2023)¹ which used different structured surfaces (samples C2, D4, D6, star symbols) also follow the same scaling law.

Figure R23. SEM of surfaces with differing surface structures used for additional droplet freezing and jumping experiments. (a) Nanostructured Al surface, labeled surface Na. The nanostructures have a characteristic size of 100-200 nm. (b) Macro-grooved Al surface, labeled surface Ma. The triangle grooves have a pitch of 1 mm and a depth of 1 mm. (c) Micro-latticed Al surface, labeled La. The spacing and depth of the square lattices are 700 μm and $\approx 100 \mu\text{m}$, respectively. (d) Hierarchical CuO nanowire surface, labeled Hi, consisting of microhills covered by CuO nanowires. The pitch and height of the microhills are 40 μm and $\approx 60 \mu\text{m}$, respectively. The CuO nanowires have a diameter of $\approx 100 \text{nm}$. (e) Re-entrant Si

surface, labeled Re. The side-to-side width and height of the hexagon lattices are 200 μm and $\approx 23 \mu\text{m}$, respectively. The wall of the lattice has a T-shape with a width of $\approx 5 \mu\text{m}$ at the top and $\approx 3 \mu\text{m}$ at the bottom. See Table R1 for the contact angles of the surfaces. See our previous studies for more details on the fabrication and characterization of the superhydrophobic Al surfaces,⁷ CuO nanowire surfaces,^{6,33} and re-entrant Si surfaces.⁹

Table R3. Apparent advancing contact angle (θ_a^{app}) and apparent receding contact angle (θ_r^{app}) of the superhydrophobic surfaces. Samples S1-S3 are characterized in the manuscript.

Surfaces	θ_a^{app} ($^\circ$)	θ_r^{app} ($^\circ$)
Nanostructured Al (Na)	163 ± 1.1	160 ± 1.5
Macrogrooved Al (Ma)	158 ± 1.8	155 ± 2.2
Microlatticed Al (La)	161 ± 1.6	159 ± 1.8
Hierarchical CuO (Hi)	163 ± 1.5	158 ± 2.4
Re-entrant Si (Re)	161 ± 2.0	144 ± 4.5

Figure R24. (a) Regime map illustrating droplet transport phenomena on superhydrophobic substrates in a low-pressure (~ 100 Pa) environment, depicting the relationship between droplet jumping velocity (v_j) and droplet size (R_d). The vaporization-momentum-governed self-jumping regime (jumping) is delimited by the evaporative drying regime (drying) for small droplets and the overpressure-initiated Leidenfrost effect regime (Leidenfrost) for large droplets. Theoretical boundaries are represented by dashed lines. Note, for the evaporative drying regime, R_d is defined as the initial droplet radius immediately before pumping down. For the self-jumping regime, R_d is measured as the droplet radius immediately before freezing. For the Leidenfrost regime, R_d is defined as the equivalent droplet radius of a spherical droplet having an identical volume immediately before droplet levitation. (b) Optical images showing the evaporative drying regime. $t=0$ s corresponds to the start of pumping down. See Video S3. (c) Theoretical determination of the lower boundary ($R_{d,l}$) by comparing the evaporation lifespan (t_{eva}) and ice incubation time (t_{inc}) as a function of the initial droplet radius (R_d). $t_{eva} < t_{inc}$ represents the droplet dries up before freezing. (d) Optical images showing the Leidenfrost effect regime. The overpressure-induced droplet-substrate separation is highlighted and zoomed in in the frame. $t_0 \approx 30$ s represents the pumping-down time before the initiation of droplet levitation. See Video S4. (e) Theoretical determination of the upper boundary ($R_{d,u}$) by comparing the overpressure force (F_{op}) and the sum of gravitational force and adhesion force ($F_g + F_a$) as a function of equivalent droplet radius (R_d). $F_{op} > F_g + F_a$ represents that the overpressure overcomes the resistance forces and induces droplet-substrate separation. Here, gravitational and adhesion forces are incorporated since the droplet has a significantly large size ($Bo > 1$) and a non-spherical shape with an increased droplet-surface contact area. Sample S1 with a microstructure height of 15 μm was used in the experiments and modeling.

To address these important points, we have revised the abstract as follows, highlighting the structure dependence of these two mechanisms and adding our new findings regarding the regime map:

Complex phase change and transport phenomena arise when sessile water droplets are exposed to low pressure. These intriguing processes have attracted growing interest due to their fundamental significance in various industries. Here, we investigate the freezing and self-jumping dynamics of supercooled water droplets on superhydrophobic surfaces, induced by a remarkable vaporization momentum, in a low-pressure environment (e.g., ~100 Pa). The vaporization momentum arises from the vaporization at droplet's free surface, progressed and intensified by recalescence, subsequently inducing droplet compression, collision, and finally self-jumping. By incorporating the physics of liquid-gas-solid phase changes involving vaporization, freezing recalescence, and liquid-solid interactions, we resolve the vaporization momentum and droplet dynamics, revealing a size-scaled jumping velocity and a nucleation-governed jumping direction with minimal reliance on tested surface structures, in contrast to previously identified overpressure directed droplet launching. Moreover, we establish a droplet size-defined regime map that distinguishes the vaporization momentum-dominated self-jumping from other depressurization-induced transport phenomena of sessile supercooled water droplets, including the evaporative drying of small droplets and the overpressure-initiated levitation of large droplets. Our findings underscore the previously uncharted size dependence of supercooled droplet dynamics and may facilitate the design of environment-controlled water-surface interaction systems for passive anti-icing, advanced cooling, cloud freezing and climate physics, particularly in the context of aerospace and deep-space exploration.

Correspondingly, to highlight the minimal dependence of vaporization momentum on surface structures, we have updated Figures 3 and 4 with additional data for more variations of the surface structures. The characterization of surfaces has been added to Section S2 of the revised Supporting Information.

Furthermore, we have strengthened the discussion on the vaporization momentum-governed jumping dynamics in contrast to overpressure-dominated ones (pages 12-13 of the revised manuscript):

...the overpressure momentum yields a jumping velocity $v_j \sim R_d^{2.5}$ (see Section S10, Supporting Information for derivation). We found that the best-fitted power index to our experimental data is 0.53 rather than 2.5 (Figure S15, see Section S10, Supporting Information). Moreover, the overpressure theory significantly underestimates the droplet jumping velocity **in the range of droplet size tested** (Figure 3e, see also Eq. S30, Supporting Information). More importantly, the local overpressure is anticipated to be sensitive to substrate structure length scales,⁵ while we observed little dependence of jumping velocity on surface patterns (**nanosheets, pillars, grooves, lattice, and re-entrance**) and solid fractions (ϕ_m ranging from 0.223 to 0.447, ϕ_n ranging from **0.108 to 0.196**) (Figure 3e, see also Figure S16, Section 10, Supporting Information). The discrepancy between the overpressure theory and our experiments suggests that droplet jumping is not dominated by the local overpressure in our experiments. However, at higher environmental pressure with a significant amount of NCGs, we anticipate the vaporization momentum to decrease due to diffusion-limited interfacial vaporization (see Figure S17, Section S10 in the Supporting Information). In such conditions, the local overpressure mechanism mediated by small-scale surface structures may play a more important role in droplet behavior.⁵ **Overpressure effects can be also amplified by the droplet sizes as the overpressure force (as well as the resulting velocity, Figure 3e) increases rapidly with droplet radius. As such, droplet levitation in the absence of freezing can be observed for extremely large droplets (~100 μ L) as we will demonstrate later.**

More importantly, we **have discussed the new findings in the context of a regime map**, developed to better identify the vaporization momentum-induced self-jumping of freezing water droplets in contrast to other depressurization-induced phenomena of sessile water droplets at the low-pressure environment and to provide a more comprehensive understanding of the reported phenomena. We have added the regime map (**Figure R24** as Figure 5) along with a detailed discussion to pages 14-15 of the revised manuscript:

The resolved dynamics of jumping droplets highlight the significant role of vaporization momentum, which, however, may be diminished or inhibited for significantly small or large droplets. To determine the regime where vaporization momentum governs droplet dynamics, we extensively expanded the range of droplet sizes in our experiments. This led to the identification of distinct regimes, defined by droplet size, in the depressurization-induced phenomena of sessile water droplets. These regimes include evaporative drying for small droplets, vaporization-

momentum-dominated droplet freezing and self-jumping for medium-sized droplets, and overpressure-initiated Leidenfrost effects for large droplets (Figure 5a). Specifically, there exist a lower ($R_{d,l}$) and upper ($R_{d,u}$) boundaries in terms of equivalent droplet radius (R_d) that delineate the regime of droplet freezing and self-jumping. For $R_d < R_{d,l}$, the droplet dries up before freezing (Figure 5b). The value of $R_{d,l}$ is determined when the evaporation lifespan (t_{eva}) of a droplet with an initial R_d is shorter than the ice incubation time (t_{inc}) (Figure 5c). While for $R_d > R_{d,u}$, the droplet experiences a marginal and dynamic levitation rather than jumping, facilitated by a vapor layer underneath, as well as remarkable fluctuations at the interface (Figure 5d), both of which indicate the characteristics of the Leidenfrost effect.¹¹ The Leidenfrost effect significantly delays freezing due to enhanced fluid convection caused by the remarkable droplet fluctuations, counteracting the decrease in interface temperature. $R_{d,u}$ is reached when the overpressure force F_{op} overcomes the resistance forces (gravitational force F_g and adhesion force F_a), causing droplet-substrate separation (Figure 5e). The theoretical boundaries are in good agreement with our experiments (see Section S13, Supporting Information for detailed analyses of the regime boundaries).

The following details on the development of the regime map have been added as a new section to the revised Supporting Information (Section S13):

To further differentiate the overpressure and vaporization momentum, we have significantly extended the range of droplet sizes (initial R_d ranging from 0.2 to 3 mm, equivalent to a volume ranging from 0.0335 to 115 μL) in our experiments and developed **a regime map (Figure R24)** to identify differing regimes including: (1) evaporative drying for small droplet sizes, (2) vaporization-momentum dominated droplet freezing and self-jumping for medium droplet sizes, and (3) Leidenfrost effects for large droplet sizes (**Figure R24a**). Specifically, there exists a lower boundary $R_{d,l}$ and an upper boundary $R_{d,u}$ in terms of droplet radius for the droplet freezing and self-jumping regime. For $R_d < R_{d,l}$, the droplet stays on the surface and dries up due to evaporation before freezing (**Figures R24b, c**), while for $R_d > R_{d,u}$, the droplet experiences a marginal elevation (but not jumping) by a vapor layer underneath the droplet and remarkable fluctuations at the interface (**Figures R24d, e**), characteristic of the Leidenfrost effect.^{11,12} The freezing is largely delayed by the Leidenfrost effect due to the remarkable fluctuations of the droplet that enhance fluid convection and counteract the decrease of interface temperature. To obtain the

lower boundary $R_{d,l}$ that separates the evaporative drying and self-jumping regimes, we compared the lifespan t_{eva} and the incubation time for ice nucleation t_{inc} of an evaporating droplet during pumping down. For a given initial droplet radius, evaporative drying occurs when $t_{eva} < t_{inc}$. t_{eva} can be obtained by considering the mass conservation during evaporation. For droplet evaporation in a depinning mode with a constant contact angle θ , which is approximately the case for our superhydrophobic surfaces having a high receding contact angle, the decrease in the droplet volume results from the vaporization over the droplet surface,^{13,14} and thus:

$$\rho dV_d = -JA_d dt, \quad (R25)$$

where $V_d = \pi R_d^3 (1 - \cos \theta)^2 (2 + \cos \theta) / 3$ is the droplet volume as a sphere cap, $A_d = 2\pi R_d^2 (1 - \cos \theta)$ is the area of the droplet's free surface, and $J = \frac{2\beta}{2-\beta} \sqrt{\frac{M}{2\pi R}} \left(\frac{P_d}{\sqrt{T_d}} - \frac{P_e}{\sqrt{T_e}} \right)$ is the vaporization mass flux given by the Hertz-Knudsen-Schrage equation^{15,16} (Eq. S7, Supporting Information) for kinetic-dominated evaporation. Integrating Eq. R25 with a known initial droplet radius R_d gives:

$$\int_0^{t_{eva}} J dt = \frac{\rho(1 - \cos \theta)(2 + \cos \theta)R_d}{2}, \quad (R26)$$

Note, J is a function of the vapor pressure at the droplet surface (P_d) and vapor pressure in the environment (P_e), all of which are time dependent. Assuming a time-averaged vaporization flux \bar{J} , the droplet lifetime can be solved:

$$t_{eva} = \frac{\rho R_d (1 - \cos \theta)(2 + \cos \theta)}{2\bar{J}}. \quad (R27)$$

To estimate \bar{J} , the transient temperature field within the droplet was simulated via COMSOL 6.0 (Heat Transfer in Fluid and Laminar Flow Physics) incorporating evaporative cooling, achieved by applying a temperature-dependent vaporization heat flux to the droplet's free surface. The modeling exploited an axisymmetric 2D geometry setup as shown in **Figure R25a**. A substrate temperature $T_{sub} = 298.15$ K in equilibrium with the environmental temperature T_e was assumed. The transient environmental pressure P_e (Figure S9, Supporting Information) was inputted as the boundary condition. Simulations were performed for an overall simulation time of 30 s with a time

step of 0.1 s. This simulation time covers the time window for our depressurization experiments (Figure 1b of the manuscript). Droplets with varied radius $R_d=0.2-3$ mm and a constant contact angle $\theta=160^\circ$ were simulated. **Figure R25b** presents the simulated temperature profile within a typically sized droplet, showing $T_d \approx 254.1$ K ($P_d \approx 112$ Pa) at $t=30$ s after the initiation of evaporative cooling. Simulations for differing droplet sizes ($R_d=0.2-3$ mm) showed a modest variation (2-3 K) of the temperature at the droplet's free surface at $t=30$ s. The simulated droplet surface temperature is a reasonable estimate when compared to the experimental measurement of the temperature inside the droplet with a similar size (-12°C , Figure 1b of the manuscript) and the derived temperature of the free surface at the freezing moment (-27°C , Eqns. S1-S3, Supporting Information). \bar{J} is then estimated by $\bar{J} = \frac{1}{\Delta t} \int_0^{\Delta t} J(T_d, P_d, T_e, P_e) dt$ with T_d (and thus P_d) simulated as a function of time t (**Figure R25c**) and Δt approximated by 30 s. Note that an evaporation coefficient^{17,18} $\beta \approx 0.759$ is adopted when evaluating J (see Section S3, Supporting Information).

It should be noted that the simulations did not incorporate changes in droplet size during evaporation and assumed negligible effects of droplet mass depletion on the droplet temperature profile. This assumption is based on the fact that the droplet size does not change significantly during the rapid pumping-down process. The simulations were only used to estimate the droplet surface temperature for the analytical modeling (Eq. R27) and were not performed to obtain the lifetime of evaporating droplets, which is feasible but numerically challenging to include the phase change (mass depletion) at the interface and internal circulation within the droplet. Instead of dealing with the numerical convergence issues, we used the simulations to estimate the surface temperature and to develop a semi-analytical model of the lifetime of evaporating droplets.

Figure R25. (a) COMSOL multiphysics model setup of the droplet evaporation, where the free surface of the droplet is subjected to the evaporation cooling flux $J(T_d)$ as a function of surface temperature T_d . The transient environmental pressure P_e (Figure S9, Supporting Information) was inputted as the boundary condition. Simulation was performed with a contact angle $\theta=160^\circ$ and a time step of 0.1 s. (b) Simulated temperature distribution at time $t=30$ s for a droplet having a radius $R_d=1.5$ mm. Evaporative cooling starts at $t=0$ s. (c) The averaged droplet surface temperature $T_{d,ave}$ as a function of time t . Minimal differences in $T_{d,ave}$ were observed for R_d ranging from 0.2 to 3 mm.

The ice nucleation incubation time t_{inc} corresponds to the moment when the very first nucleus is formed in the droplet. The total number of nuclei is a function of the droplet volume and the ice nucleation rate. The latter is dependent on the temperature field inside the droplet. Here, the transient temperature field of droplets with different sizes was obtained from the numerical simulation performed above (Figure R25). With a known time-dependent temperature field, the local nucleation rate per unit volume is given by the classical nucleation theory:¹⁹

$$J_n = \frac{A_m}{3} n^{7/3} \sqrt{\frac{A_m \sigma}{\pi k_B T}} D_w \exp \left\{ - \frac{4 \left(\frac{A_m \sigma}{k_B T} \right)^3}{27 \left[\ln \left(\frac{P_w}{P_i} \right) \right]^2} \right\}, \quad (R28)$$

where $A_m = 4\pi r_m^2$ is the surface area of a water molecule with a radius of r_m , $n = \frac{\rho N}{M}$ is the number density of the water, ρ is water density, Avogadro constant N , and molar mass M , $D_w = 1.5 \times 10^{-6} \exp \left(- \frac{2045.0}{T} \right)$ is the self-diffusivity of supercooled water, $\sigma = 0.0291 + 10^{-4}(T - 273.15)$ is the water-ice interfacial tension, P_w is the vapor pressure of supercooled water at temperature T , and P_i is the vapor pressure of ice at an equilibrium freezing temperature of 273.15

K. The condition of ice nucleation is defined when the total number of ice nuclei exceeds unity, given by integrating the local ice nucleation rate J_n over the droplet volume during a period of t_{inc} :^{19–21}

$$\int_0^{t_{inc}} \int_{V_d}^{V_{d,t}} J_n dV_d dt = 1, \quad (R29)$$

where $V_{d,t}$ is the droplet volume at time t and can be solved by Eq. R25. For a given initial droplet volume V_d , t_{inc} can be found by numerically solving the integration problem above with the input of the time-dependent droplet temperature profile. Given that the droplet temperature is lower than the substrate (Figure R25), we only considered the homogeneous nucleation and neglected the heterogeneous nucleation at the droplet-substrate interface. Figure R24c shows t_{inc} as a function of the initial droplet radius R_d , revealing that a smaller droplet has a longer incubation time for ice nucleation. This is because a smaller surface area or volume leads to a reduced number of ice nuclei. Moreover, the small droplet has a lower temperature that is closer to the substrate temperature due to its lower thermal conduction resistance.

By comparing the ice incubation time and the lifetime of the evaporating droplet, a critical droplet size $R_{d,l}$, below which ice nucleation hardly occurs, can be solved by:

$$t_{eva} < t_{inc}. \quad (R30)$$

The calculated $R_{d,l}$ is shown to be consistent with the experiments (Figures R24a and R24c).

To obtain the upper droplet size boundary $R_{d,u}$ that separates the Leidenfrost effect and the self-jumping regimes, we consider the criteria for the initiation of the Leidenfrost effect, which starts when the overpressure force F_{op} underneath the droplet overcomes the gravity and adhesion of the droplet. Here, gravitational force (F_g) and adhesion force (F_a) are incorporated as the droplet has a significantly large size ($Bo > 1$) and a non-spherical shape with increased droplet-surface contact area. The overpressure force F_{op} , defined consistently with that in our analyses of overpressure (Figure 3e in the manuscript and Eqns. S26-S29 in the Supporting Information), can be modeled by:⁵

$$F_{op} \approx \frac{3\mu\pi R_b^4 J}{h^3 F \rho_v}, \quad (R31)$$

where μ is the dynamic viscosity of vapor, R_b is the basal radius of the droplet, J is the evaporation flux of the droplet, h is the height of the surface structure, $F = 1 + Kn \left(\frac{2}{\omega} + 1 \right)$ is the correction factor for the slip flow with respect to the Knudsen number Kn and the tangential momentum accommodation factor $\omega=0.87$,⁵ and ρ_v is the vapor density corresponding to the droplet-substrate interfacial temperature.

The evaporation flux (J) can be calculated by Eq. S7 of the Supporting Information. Due to the absence of freezing, J is evaluated by the temperature of supercooled water at the droplet-substrate interface rather than the freezing temperature. The droplet-substrate interfacial temperature can be approximated by the average of the substrate temperature and the droplet's free surface temperature. The latter was obtained from the numerical simulations (Figure R25), which demonstrated a free surface temperature of ≈ 253.4 K at $t=30$ s, with minimal dependence on the droplet radius (0.6-3 mm). We note that $t=30$ s is adequately long for the environmental pressure to stabilize and for the droplet to reach a relatively stable temperature, thus the simulated droplet surface temperature could be a reasonable estimate of the temperature at the moment of droplet levitation.

The adhesion force F_a can be calculated by considering the contact angle hysteresis of the surface:^{6,22,23}

$$F_a = 2\pi\sigma R_b(1 + \cos \theta_r), \quad (R32)$$

where σ is the surface tension of the water, R_b is the basal radius of the droplet at the moment of levitation, θ_r is the receding contact angle that is assumed to be close to the apparent receding contact angle θ_r^{app} .

The gravitational force of the droplet F_g is obtained by:

$$F_g = \rho V_d g, \quad (R33)$$

where ρ is the density of water, g is the gravitational acceleration, V_d is the droplet volume at the levitation moment and is a function of the droplet basal radius R_b . For Bond numbers of droplets close to 1, the droplet can no longer hold a spherical cap shape. In this case, the volume of the droplet as a function of the basal radius R_b can be derived by using a polynomial to interpolate the volume-radius relationship given by Ref.24:

$$V_d [\mu\text{L}] = 16.164R_b^2 + 4.6477R_b + 4.7753 [R_b \text{ in mm}], \quad (\text{R34})$$

which is interpolated for a droplet with a contact angle of 162° . A critical droplet volume V_d is then solved by Eq. R35 below and converted to a critical droplet radius $R_{d,u}$ (of a sphere with an equivalent volume of V_d). The calculation is consistent with our experiments (Figures R24a and R24e).

$$F_{op} > F_a + F_g. \quad (\text{R35})$$

Although a large and stationary droplet necessitates a shorter timescale to incubate ice nucleation (due to a large surface area), it has a higher volume-averaged temperature as we have demonstrated in **Figure R26** via numerical simulations (COMSOL 6.0 Heat Transfer in Fluid and Laminar Flow Physics under an axisymmetric 2D geometry). Here, the average temperature of a droplet was defined by the average of the maximum and minimum temperatures within the droplet. Furthermore, the temperature profile within the droplet can be significantly flattened due to the remarkable fluctuations of the Leidenfrost droplet that enhances fluid convection. Therefore, freezing is delayed by the Leidenfrost effect.

Once the Leidenfrost effect is initiated via overpressure, a vapor layer is formed to provide shear pressure that elevates the droplet.¹¹ The Leidenfrost phenomenon is characterized by a vapor film underneath the droplet that elevates the droplet from the substrate, which shares similarities to the overpressure effects that drive droplet trampolining.⁵ However, the difference between overpressure and the shear stress driving the Leidenfrost effect is that the Leidenfrost droplet detaches locally and slightly from the surfaces and has reduced dependence on surface structures, while the overpressure theory considers the situation when the droplet contacts closely with the structured surface and the limited drainage of the vapor through the structures strengthens the local pressure. Furthermore, the thickness of the vapor layer for a Leidenfrost droplet is highly

correlated to the weight of the droplet, and thus the local pressure decreases rapidly with the vapor layer thickness.¹¹

Figure R26. Volume-averaged droplet temperature T_{avg} as a function of droplet size R_d ranging from 0.6–3 mm for a water droplet exposed to a low-pressure environment. Simulations were performed via COMSOL 6.0 (Heat Transfer in Fluid and Laminar Flow Physics) under an axisymmetric 2D geometry. The transient depressurizing pressure as a function of time (Figure S9, Supporting Information), a constant substrate temperature of 298.15 K, a fixed droplet shape ($\theta=160^\circ$), and a no-slip boundary condition on the droplet’s free surface for simplicity were applied. The simulation was carried out in a time-dependent model, where the time span of the simulation was 30 s, adequately long for the stabilization of the environmental pressure. The average temperature was extracted from the droplet at 30 s.

References:

1. Lambley, H. *et al.* Freezing-induced wetting transitions on superhydrophobic surfaces. *Nat Phys* (2023) doi:10.1038/s41567-023-01946-3.
2. Graeber, G., Dolder, V., Schutzius, T. M. & Poulikakos, D. Cascade Freezing of Supercooled Water Droplet Collectives. *ACS Nano* **12**, 11274–11281 (2018).
3. Stan, C. A. *et al.* Rocket drops: The self-propulsion of supercooled freezing drops. *Phys Rev Fluids* **8**, L021601 (2023).
4. Kalita, A. *et al.* Microstructure and crystal order during freezing of supercooled water drops. *Nature* **620**, 557–561 (2023).
5. Schutzius, T. M. *et al.* Spontaneous droplet trampolining on rigid superhydrophobic surfaces. *Nature* **527**, 82–85 (2015).
6. Yan, X. *et al.* Droplet Jumping: Effects of Droplet Size, Surface Structure, Pinning, and Liquid Properties. *ACS Nano* acsnano.8b06677 (2019) doi:10.1021/acsnano.8b06677.
7. Ma, W. *et al.* Solar-assisted icephobicity down to -60°C with superhydrophobic selective surfaces. *Cell Rep Phys Sci* **2**, 100384 (2021).
8. Yan, X. *et al.* Atmosphere-Mediated Scalable and Durable Biphilicity on Rationally Designed Structured Surfaces. *Adv Mater Interfaces* **7**, (2020).
9. Liao, D., Yang, Y. & Qiu, H. Droplet impact dynamics and heat transfer on nanostructured doubly reentrant cavity under freezing temperature. *Physics of Fluids* **33**, (2021).
10. Lathia, R., Modak, C. D. & Sen, P. Two modes of contact-time reduction in the impact of particle-coated droplets on superhydrophobic surfaces. *Droplet* **2**, (2023).
11. Shi, M., Das, R., Arunachalam, S. & Mishra, H. Suppression of Leidenfrost effect on superhydrophobic surfaces. *Physics of Fluids* **33**, (2021).
12. Yang, J. *et al.* A standing Leidenfrost drop with Sufi whirling. *Proceedings of the National Academy of Sciences* **120**, (2023).
13. Kang, F., Shen, Y., Cheng, Y. & Li, N. Lifetime Prediction of Sessile Droplet Evaporation with Coupled Fields. *Ind Eng Chem Res* **60**, 15782–15792 (2021).
14. Nguyen, T. A. H. & Nguyen, A. V. On the Lifetime of Evaporating Sessile Droplets. *Langmuir* **28**, 1924–1930 (2012).
15. Gerasimov, D. N. & Yurin, E. I. *Kinetics of Evaporation*. vol. 68 (Springer International Publishing, 2018).
16. Vaartstra, G., Lu, Z., Lienhard, J. H. & Wang, E. N. Revisiting the Schrage Equation for Kinetically Limited Evaporation and Condensation. *J Heat Transfer* **144**, (2022).

17. Narusawa, U. & Springer, G. S. Measurements of evaporation rates of water. *J Colloid Interface Sci* **50**, 392–395 (1975).
18. Eames, I. W., Marr, N. J. & Sabir, H. The evaporation coefficient of water: a review. *Int J Heat Mass Transf* **40**, 2963–2973 (1997).
19. Meng, Z. & Zhang, P. Dynamic propagation of ice-water phase front in a supercooled water droplet. *Int J Heat Mass Transf* **152**, 119468 (2020).
20. Meng, Z. & Zhang, P. Freezing dynamics of supercooled micro-sized water droplets. *Int J Heat Mass Transf* **193**, 122955 (2022).
21. Castillo, J. E., Huang, Y., Pan, Z. & Weibel, J. A. Quantifying the Pathways of Latent Heat Dissipation during Droplet Freezing on Cooled Substrates. *Int J Heat Mass Transf* **164**, 120608 (2021).
22. Yan, X. *et al.* Particulate–Droplet Coalescence and Self-Transport on Superhydrophobic Surfaces. *ACS Nano* **16**, 12910–12921 (2022).
23. Konstantinidis, K., Göhl, J., Mark, A. & Sasic, S. Coalescence-induced jumping of droplets from superhydrophobic surfaces—The effect of contact-angle hysteresis. *Physics of Fluids* **34**, (2022).
24. Yu, Y., Lv, C., Wang, L. & Li, P. The Shape of Heavy Droplets on Superhydrophobic Surfaces. *ACS Omega* **5**, 26732–26737 (2020).
25. Au, S. C. Y. *et al.* Supercooled Droplet Icing and Self-Jumping on Micro/nanostructured Surfaces: Role of Vaporization Momentum. (2022).
26. Chauhari, V. D. & Desai, A. D. Performance Evaluation of Vacuum System: Pump-down Time. *Int J Sci Eng Res* **2**, (2011).
27. Jung, S., Tiwari, M. K. & Poulikakos, D. Frost halos from supercooled water droplets. *Proceedings of the National Academy of Sciences* **109**, 16073–16078 (2012).
28. Jung, S., Tiwari, M. K., Doan, N. V. & Poulikakos, D. Mechanism of supercooled droplet freezing on surfaces. *Nat Commun* **3**, 615 (2012).
29. Nguyen, V.-H. *et al.* Anti-icing performance on aluminum surfaces and proposed model for freezing time calculation. *Sci Rep* **11**, 3641 (2021).
30. Hao, T., Zhu, Z., Yang, H., He, Z. & Wang, J. All-Day Anti-Icing/Deicing Film Based on Combined Photo-Electro-Thermal Conversion. *ACS Appl Mater Interfaces* **13**, 44948–44955 (2021).
31. Jafari, R. & Farzaneh, M. Fabrication of superhydrophobic nanostructured surface on aluminum alloy. *Applied Physics A* **102**, 195–199 (2011).
32. Boreyko, J. B. & Collier, C. P. Delayed Frost Growth on Jumping-Drop Superhydrophobic Surfaces. *ACS Nano* **7**, 1618–1627 (2013).

33. Yan, X. *et al.* Atmosphere-Mediated Superhydrophobicity of Rationally Designed Micro/Nanostructured Surfaces. *ACS Nano* **13**, 4160–4173 (2019).

REVIEWER COMMENTS

Reviewer #1 (Remarks to the Author):

The authors have accounted for most of my feedback, which I think has improved the manuscript. I'm still not fully convinced regarding the inertial-capillary time discussion---I don't believe the analogy between droplet impact and rectifying kinetic to surface to kinetic energy holds here as the droplet interface deformation is never rectified back to kinetic energy. The only coupling I see here is the fact that the deformation leads to a larger droplet-substrate contact area at the moment of departure. If this is what they are trying to argue then I would say that this didn't come clearly through for me. Regardless, they put their best idea forward and have experimental evidence.

Reviewer #2 (Remarks to the Author):

I thought the authors did a tremendous job revising their paper in response to the reviewers' comments. In particular, their expanded parameter space for validating their model for the speed and direction of vaporization momentum-induced jumping for a wide variety of droplets sizes and surface types was a nice touch. Especially where they show and rationally model failure to jump at the two extremes, of evaporative drying at small droplet sizes and overpressure-induced Leidenfrost effect for puddles. I also liked how they performed careful visualizations to confirm that the droplet was still ~90% liquid during droplet recoil, to validate how rebound and jumping is even possible during solidification. There is no doubt that this work offers a more mature and systematic understanding of vaporization momentum-induced jumping upon recalescent freezing.

I do want to push back somewhat on the repeated claim in the rebuttal that the Nature Physics paper could not offer any mechanism for jumping. In particular, the repeated statement that "the top-down recalescent force induces liquid impalement into structures and acts against droplet jumping," feels unfair given that the Nature Physics paper clearly posits that the top-down recalescent force is indeed responsible for the jumping behavior (which they call "self-expulsion"). While it is true they do not explicitly talk about the rebound in detail, it is clearly implied when they discuss how expulsion is possible when the downward recalescent force is not stronger than the impalement force. In other words, if a droplet's downward recoil force on a superhydrophobic surface is weaker than the impalement force but stronger than the adhesive force, it is clearly implied that rebound will then occur for the given context of a subsequent jumping event. Indeed, the "News & Views" piece related to this publication ("Freeze in or breeze out") even talks about this rebound for the self-expulsion regime. So I do feel that the authors should step away from these sorts of statements, and instead, focus on the fact that the vaporization-induced jumping is

characterized here in substantively greater detail, for example the virtues extolled in my first paragraph above and in my original review. But this present work is certainly not a brand new mechanism relative to the Nature Physics paper, as they seemingly try to assert here.

Regarding whether a substantively fleshed out model and systematic results for the recalescent-induced jumping effect is enough to justify publication in Nature Communications, when the mechanism itself is not new, is a difficult question to answer in my opinion. I am certainly amenable to publication after this substantive revision, if the editor and other reviewers feel it is warranted.

Reviewer #3 (Remarks to the Author):

The authors have made substantial efforts to address my concerns and revise the manuscript.

Response to Reviewers

Manuscript ID: NCOMMS-23-25686A

Article Title: Unraveling the Role of Vaporization Momentum in Self-Jumping Dynamics of Freezing Supercooled Droplets at Reduced Pressures

We greatly appreciate the valuable comments and suggestions provided by the reviewers. In response to their feedback, we have made the following revisions: 1) We have refined the discussion on the capillary time and detachment time to avoid any potential confusion. 2) We have added a discussion to further highlight the contributions of previous studies and to focus specifically on our findings in contrast to these studies.

Below we have provided point-to-point responses (**in black**) to reviewers' comments (**in blue**). Text in the manuscript and Supporting Information is in *italics*. Revised text in the manuscript or Supporting Information is highlighted **in yellow**. Once again, we would like to express our gratitude to the reviewers for their helpful input.

Response to Reviewer # 1

Comments: The authors have accounted for most of my feedback, which I think has improved the manuscript. I'm still not fully convinced regarding the inertial-capillary time discussion---I don't believe the analogy between droplet impact and rectifying kinetic to surface to kinetic energy holds here as the droplet interface deformation is never rectified back to kinetic energy. The only coupling I see here is the fact that the deformation leads to a larger droplet-substrate contact area at the moment of departure. If this is what they are trying to argue then I would say that this didn't come clearly through for me. Regardless, they put their best idea forward and have experimental evidence.

Response: We appreciate the reviewer for the rigorous comments regarding the discussion of inertial-capillary time. We agree that the analogy (droplet impact dynamics vs. droplet freezing, deformation, and jumping dynamics) is not rigorous due to the involvement of the ice shell solidification and liquid core fluid dynamics, which is not simply equivalent to the capillary-inertial process.

As we presented in our last round of responses to reviewers, we would like to highlight the observed deformation (**Figure R1**, indicated by blue arrows at $t=8.2$ ms) and contraction (**Figure R1**, indicated by blue arrows at $t=12.4$ ms) of the liquid bulb at the bottom of the droplet when interacting with the substrate. In this revision, we included a new **Video S3** corresponding to Figure S20 (**Figure R1**) in the Supporting Information, from which the dynamics of both liquid and solid phases were captured clearly (Figure R1, water-ice boundary marked by yellow lines). Although the liquid core is partially confined and the deformation is not fully recovered after jumping, the confinement reduces the oscillation of the droplet, and the hydrodynamics of the partially confined liquid shall be conducive to momentum transfer. Our most important finding here is the scaling of the departure time with droplet size, i.e., $\Delta t_d \sim R_d^{1.5}$, and we would rather focus on the experimental observation and results.

To make our discussion more rigorous, we have shifted our focus to the droplet dynamics and highlighted the findings of the departure time scaling. The text on pages 7-8 of the **manuscript** was rephrased as follows, with the changes highlighted in yellow:

Regardless of surface microstructures (Figure 3c), Δt_d is shown to scale with $R_d^{1.5}$ ($\Delta t_d \approx 260R_d^{1.5}$, Figure 3d), where R_d is droplet's spherical radius immediately before freezing. In addition to the ice shell formation governed by freezing kinetics, the fluid dynamics of the unfrozen portion of the droplet likely play a significant role in determining the timescale of droplet deformation and detachment. High temporal resolution visualization revealed a liquid bulb at the bottom of the droplet, which underwent deformation and contraction when interacting with the substrate (see Videos S3 and Figure S20, Supporting Information). Furthermore, energy and heat transfer analysis shows that the average ice mass fraction within the ice-water slurry is adequately low ($\approx 10\%$) and the liquid volume within the freezing droplet occupies a large portion at the droplet detaching moment ($t = \Delta t_d$) (see Figures S6-S10, Section S5, Supporting Information for the transient heat transfer modeling and the calculation of the ice-water distribution^{1,2} within the droplet upon the completion of recalescence). Even in the presence of an ice shell, Δt_d is found to be comparable with the characteristic timescale of a freely oscillating droplet³ and the contact time of a low-deformation impact droplet on superhydrophobic surfaces (see Section S11, Supporting Information for a discussion of timescale).^{3,4}

Furthermore, we have revised related discussions in Section 11 of the **Supporting Information**:

We have introduced the detachment time Δt_d to characterize the time scale during which the droplet effectively interacts with the substrate to accumulate and redirect the vaporization momentum. Δt_d is defined as the time duration starting from ice nucleation to droplet-substrate separation.

To understand Δt_d in comparison with the timescale of other extensively studied droplet dynamics (free oscillation and bouncing), we compared Δt_d with the capillary time τ_c , and found a linear scaling ($\Delta t_d/\tau_c \approx 2.2$, or $\Delta t_d \sim R_d^{1.5}$) for the tested droplet sizes ($0.6 \leq R_d \leq 2.8$ mm, see Figure S19) and experimental conditions ($P_e \sim 100$ Pa), regardless of the variation of the tested structure geometries and length scales (see Figure S4). Here, $\tau_c = \sqrt{\rho_w R_d^3 / \sigma}$, ρ_w is the water density, and σ is the surface tension of water. The scaling of Δt_d ($\approx 2.2\tau_c$) is close to the characteristic timescale of a freely oscillating droplet ($\approx 2.2\tau_c$) and is slightly lower than the contact time of a low-deformation impact droplet on superhydrophobic surfaces ($\approx 2.6\tau_c$).^{3,4} This

comparison was motivated by the experimental observation that the freezing droplet experiences a compressive deformation under the recalescence-induced propulsion force while the bottom of the droplet remains unfrozen before jumping, reminiscent of the process of droplet impact and bouncing on superhydrophobic surfaces. We note that the droplet freezing, deforming, and jumping dynamics are different from the classic droplet bouncing dynamics, where a droplet experiences spreading and retraction before bouncing under the capillary and inertia forces. Whereas in the freezing droplet case, the droplet deforms as the ice shell builds up, and the retraction of the liquid core is partially restricted by the ice-slurry shell. Nevertheless, the similar time scales indicate that the hydrodynamics of the liquid core within the droplet may play an important role in droplet deformation and jumping.

We assumed that during the short-time droplet deformation, the ice shell remains a thin layer of ice-water slurry mixture with a large fraction being liquid water. This assumption was justified by our calculations of the thickness and water mass fraction of the ice-water slurry shell at the detaching moment (see Figure S11b), showing that for a 16 μL water droplet around the detachment moment ($t=17$ s), the average thickness of the ice slurry shell reaches $\approx 47\%$ of the droplet radius, and the averaged ice mass fraction within the ice-water slurry is $<10.2\%$, which is adequately low such that the ice slurry shell remains a very loose ice-water mixture. To some extent, the hydrodynamics of ice-shelled droplets in the early stage of freezing are similar to those of particle-laden droplets or droplet marbles, which behave as a liquid when impacting solid surfaces at a low particle concentration.⁵ To further confirm the assumption, we have performed additional experiments to capture the freezing dynamics at a higher capture rate (up to 20000 fps) using a high-speed camera (Photron Fastcam Nova S12) with a 12X micro lens. The enhanced temporal resolution allowed us to clearly differentiate the ice and water phases and capture the interaction between the liquid portion with the substrate (Figure S20). It was seen that before detaching, the liquid part of the droplet was pushed to deform and then contracted when interacting with the substrate ($t \in [8.2$ ms, 12.4 ms], Figure S20, Video S3). Even at the detaching moment, part of the liquid bottom remains unfrozen ($t < 14.4$ ms, Figure S20), as further reflected by the slight oscillation and contraction of the liquid after detaching ($t > 14.4$ ms, Figure S20). This indicates the hydrodynamics of the droplet might be only partially confined by the ice shell during the short-time deformation process.

Figure R1. High-speed optical imaging of the droplet freezing, deforming, and self-jumping dynamics. The enhanced temporal image sequences captured the dynamics of both ice and water phases and the interaction between the liquid portion with the substrate. The initiation of ice nucleation was marked by the yellow diamond, the vaporization repulsion force (F_t) acting on the ice shell was indicated by the yellow arrow, the ice front was traced by yellow curves, the liquid spreading and retraction during droplet compression was marked by blue arrows, and the jumping velocity (v_j) was marked by the red arrow. A superhydrophobic hierarchical CuO nanowire surface with 40- μm -spaced microhills⁶ (surface Hi, see Figure S4) was used as the substrate. See Video S3.

Response to Reviewer # 2

Comments: I thought the authors did a tremendous job revising their paper in response to the reviewers' comments. In particular, their expanded parameter space for validating their model for the speed and direction of vaporization momentum-induced jumping for a wide variety of droplets sizes and surface types was a nice touch. Especially where they show and rationally model failure to jump at the two extremes, of evaporative drying at small droplet sizes and overpressure-induced Leidenfrost effect for puddles. I also liked how they performed careful visualizations to confirm that the droplet was still ~90% liquid during droplet recoil, to validate how rebound and jumping is even possible during solidification. There is no doubt that this work offers a more mature and systematic understanding of vaporization momentum-induced jumping upon recalescent freezing.

I do want to push back somewhat on the repeated claim in the rebuttal that the Nature Physics paper could not offer any mechanism for jumping. In particular, the repeated statement that “the top-down recalescent force induces liquid impalement into structures and acts against droplet jumping,” feels unfair given that the Nature Physics paper clearly posits that the top-down recalescent force is indeed responsible for the jumping behavior (which they call “self-expulsion”). While it is true they do not explicitly talk about the rebound in detail, it is clearly implied when they discuss how expulsion is possible when the downward recalescent force is not stronger than the impalement force. In other words, if a droplet's downward recoil force on a superhydrophobic surface is weaker than the impalement force but stronger than the adhesive force, it is clearly implied that rebound will then occur for the given context of a subsequent jumping event. Indeed, the “News & Views” piece related to this publication (“Freeze in or breeze out”) even talks about this rebound for the self-expulsion regime. So I do feel that the authors should step away from these sorts of statements, and instead, focus on the fact that the vaporization-induced jumping is characterized here in substantively greater detail, for example the virtues extolled in my first paragraph above and in my original review. But this present work is certainly not a brand new mechanism relative to the Nature Physics paper, as they seemingly try to assert here.

Regarding whether a substantively fleshed out model and systematic results for the recalescent-induced jumping effect is enough to justify publication in Nature Communications, when the

mechanism itself is not new, is a difficult question to answer in my opinion. **I am certainly amenable to publication after this substantive revision**, if the editor and other reviewers feel it is warranted.

Response: We thank the reviewer for affirming our revision and improvement. We also appreciate the reviewer for bringing forth the Nature Physics publication (“News and Views”, Ref. 8 below) which provided a complementary and insightful discussion on the rebound dynamics. As suggested by the reviewer, we have rephrased our statements to further highlight the contribution of previous studies (page 4 of the revised manuscript):

For a sessile droplet, the recalescence force typically develops in a progressive top-down manner, from the droplet's free surface to the substrate, and tends to induce droplet impalement into sparsely spaced substrate structures.^{7,8} This opposes the localized overpressure that typically separates the droplet from the substrate, leading to self-trampolining of vaporizing droplets.⁹ Rational substrate structuring promotes self-expulsion of freezing supercooled droplets,⁷ potentially due to rebound behavior.^{7,8} However, a more comprehensive understanding of the top-down recalescence force's role in rebound and self-expulsion dynamics is needed. The diverse outcomes (impalement, jumping, and trampolining) emphasize the necessity for a thorough comprehension of the intricate dynamics in multiple phase change processes, as well as the challenges in predicting the droplet freezing and jumping phenomena.

Given the clear discussion and illustration of droplet freezing and jumping physics, we have added references to the aforementioned publication (*Nature Physics*, 19, 609–610, 2023, Ref. 8) within the discussion of our revised manuscript (pages 4, 18, 19):

For a sessile droplet, the recalescence force typically develops in a progressive top-down manner, from the droplet's free surface to the substrate, and tends to induce droplet impalement into sparsely spaced substrate structures.^{7,8}

While previous studies have delicately elucidated the effects of surface structures in droplet dynamics (self-trampolining, suffusion, wetting transition, and expulsion),^{7,8} current work focuses on droplet self-jumping from impalement-resisting superhydrophobic surfaces, and highlights the crucial role of droplet size rather than surfaces structures in jumping velocity and

the size-defined fate of droplets at reduced pressures (evaporative drying, self-jumping, and overpressure-initiated levitation).

However, the criteria of wetting transition^{7,8} need to be incorporated into the rational design of surface structures, especially single-tier structures, to regulate vaporization-mediated droplet dynamics.

Regarding the reviewer's last concern regarding the droplet rebound mechanisms reported by other studies, we would like to emphasize the new findings of our current work, as the reviewer has affirmed: 1) Elucidation of the freezing and jumping dynamics, including the jumping velocity scaling; 2) A regime map revealing the droplet-size defined fate and dynamics of freezing droplets at reduced pressures. We believe these aspects were the most important results that enhance the novelty and differentiate our work from previous studies. Furthermore, while previous studies have delicately elucidated the effects of surface structures in droplet dynamics (self-trampolining, suffusion, wetting transition, and expulsion),⁷⁻⁹ our current work focuses on droplet self-jumping from impalement-resisting superhydrophobic surfaces. Our work highlights the crucial role of droplet sizes rather than surface structures in jumping velocity and the fate of droplets at reduced pressures (evaporative drying, self-jumping, and overpressure-initiated levitation).

Again, we thank the reviewer for the helpful comments that have inspired us to improve our work.

Response to Reviewer # 3

Comments: The authors have made **substantial efforts** to address my concerns and revise the manuscript.

Response: We appreciate the reviewer's favorable comments.

References:

1. Meng, Z. & Zhang, P. Freezing dynamics of supercooled micro-sized water droplets. *Int J Heat Mass Transf* **193**, 122955 (2022).
2. Castillo, J. E., Huang, Y., Pan, Z. & Weibel, J. A. Quantifying the Pathways of Latent Heat Dissipation during Droplet Freezing on Cooled Substrates. *Int J Heat Mass Transf* **164**, 120608 (2021).
3. Rayleigh, Lord. VI. On the capillary phenomena of jets. *Proceedings of the Royal Society of London* **29**, 71–97 (1879).
4. Richard, D., Clanet, C. & Quéré, D. Contact time of a bouncing drop. *Nature* **417**, 811–811 (2002).
5. Lathia, R., Modak, C. D. & Sen, P. Two modes of contact-time reduction in the impact of particle-coated droplets on superhydrophobic surfaces. *Droplet* **2**, (2023).
6. Yan, X. *et al.* Droplet Jumping: Effects of Droplet Size, Surface Structure, Pinning, and Liquid Properties. *ACS Nano* acsnano.8b06677 (2019) doi:10.1021/acsnano.8b06677.
7. Lambley, H. *et al.* Freezing-induced wetting transitions on superhydrophobic surfaces. *Nat Phys* (2023) doi:10.1038/s41567-023-01946-3.
8. Boreyko, J. B. Freeze in or breeze out. *Nat Phys* **19**, 609–610 (2023).
9. Schutzius, T. M. *et al.* Spontaneous droplet trampolining on rigid superhydrophobic surfaces. *Nature* **527**, 82–85 (2015).

REVIEWERS' COMMENTS

Reviewer #1 (Remarks to the Author):

My points have been sufficiently addressed.

Reviewer #2 (Remarks to the Author):

I feel this final revision has adequately resolved the reviewers' remaining concerns regarding the departure time-scale and clarifying the novelty claims. The other reviewers seem to be in agreement that the previous revision dramatically improved the impact of this work. I am therefore happy to recommend acceptance.